# RNA-binding proteins hnRNPM and ELAVL1 promote type-I interferon induction downstream of the nucleic acid sensors cGAS and RIG-I

Alexander Kirchhoff [ID][1][✉], Anna-Maria Herzner[1,14], Christian Urban [ID][2], Antonio Piras[2], Robert Düster [ID][3], Julia Mahlberg [ID][1], Agathe Grünewald[1], Thais M Schlee-Guimarães [ID][1], Katrin Ciupka[1], Petro Leka[4], Robert J Bootz [ID][1], Christina Wallerath [ID][1], Charlotte Hunkler[1], Ann Kristin de Regt[1], Beate M Kümmerer [ID][5,6], Maria Hønholt Christensen [ID][4], Florian I Schmidt [ID][4], Min Ae Lee-Kirsch[7,8], Claudia Günther [ID][9], Hiroki Kato [ID][10], Eva Bartok [ID][1,11,12], Gunther Hartmann[1], Matthias Geyer [ID][3], Andreas Pichlmair [ID][2,13] & Martin Schlee [ID][1][✉]

## Abstract

The cytosolic nucleic acid sensors RIG-I and cGAS induce type-I interferon (IFN)-mediated immune responses to RNA and DNA viruses, respectively. So far no connection between the two cytosolic pathways upstream of IKK-like kinase activation has been investigated. Here, we identify heterogeneous nuclear ribonucleoprotein M (hnRNPM) as a positive regulator of IRF3 phosphorylation and type-I IFN induction downstream of both cGAS and RIG-I. Combining interactome analysis with genome editing, we further uncover the RNA-binding protein ELAV-like protein 1 (ELAVL1; also known as human antigen R, HuR) as an hnRNPM interactor. Depletion of hnRNPM or ELAVL1 impairs type-I IFN induction by herpes simplex virus 1 or Sendai virus. In addition, we show that hnRNPM and ELAVL1 interact with TANK-binding kinase 1, IκB kinase ε, IκB kinase β, and NF-κB p65. Our confocal microscopy experiments demonstrate cytosolic and perinuclear interactions between hnRNPM, ELAVL1, and TBK1. Furthermore, pharmacological inhibition of ELAVL1 strongly reduces cytokine release from type-I interferonopathy patient fibroblasts. The RNA-binding proteins hnRNPM and ELAVL1 are the first non-redundant regulators to bridge the cGAS/STING and RIG-I/MAVS pathways. Overall, our study characterizes the hnRNPM-ELAVL1 complex as a novel system promoting antiviral defense, pointing to a potential therapeutic target to reduce auto-inflammation in patients with type-I interferonopathies.

**Keywords** hnRNPM; ELAVL1; cGAS Signaling; RIG-I Signaling; IRF3
**Subject Categories** Immunology; Microbiology, Virology & Host Pathogen

Interaction; Molecular Biology of Disease

## Introduction

Recognition of non-self nucleic acids is an essential defense mechanism of the innate immune system (Tan et al, 2018). As multiple pathogens exploit the cytosol as a replication niche, it is constantly monitored for the presence of pathogen-associated molecular patterns (PAMPs) by specialized pattern recognition receptors (PRRs). In the cytosol, cyclic GMP-AMP (cGAMP) synthase (cGAS) constitutes the principal type I interferon (IFN)-inducing receptor of double-stranded (ds) DNA, whereas retinoic acid inducible gene I (RIG-I) and melanoma differentiation associated gene 5 (MDA5) are the predominant sensors of cytosolic dsRNA species (Schlee and Hartmann, 2016; Hertzog and Rehwinkel, 2020). All three receptors signal via the formation of multimeric protein complexes leading to interferon regulatory factor 3 (IRF3)-dependent type I IFN induction. Detection of cytoplasmic dsDNA activates cGAS to produce 2′3′-cGAMP, a second messenger that binds to the endoplasmic reticulum (ER)-resident adaptor protein stimulator of interferon genes (STING) (Wu et al, 2013; Sun et al, 2013b). Upon activation, STING multimerizes and translocates from the ER to the Golgi compartment. STING clustering provides a suitable surface for the

[1]Institute of Clinical Chemistry and Clinical Pharmacology, University Hospital Bonn, Bonn, Germany. [2]Institute of Virology, Technical University of Munich, School of Medicine, Munich, Germany. [3]Institute of Structural Biology, University Hospital Bonn, Bonn, Germany. [4]Institute of Innate Immunity, University Hospital Bonn, Bonn, Germany. [5]Institute of Virology, University Hospital Bonn, Bonn, Germany. [6]German Center for Infection Research (DZIF), Partner Site Bonn-Cologne, 53127 Bonn, Germany. [7]Department of Pediatrics, Medizinische Fakultät Carl Gustav Carus, Technische Universität Dresden, Dresden, Germany. [8]German Center for Child and Adolescent Health (DZKJ), partner site Leipzig/Dresden, Dresden, Germany. [9]Department of Dermatology, Medizinische Fakultät Carl Gustav Carus, Technische Universität Dresden, Dresden, Germany. [10]Institute of Cardiovascular Immunology, University Hospital Bonn, Bonn, Germany. [11]Unit of Experimental Immunology, Department of Biomedical Sciences, Institute of Tropical Medicine, Antwerp, Belgium. [12]Institute of Experimental Haematology and Transfusion Medicine, University Hospital Bonn, Bonn, Germany. [13]German Center for Infection Research (DZIF), Partner Site Munich, 81675 Munich, Germany. [14]Present address: Department of Cancer Immunology and Immune Modulation, Boehringer Ingelheim Pharma GmbH & Co. KG, Biberach an der Riß, Germany. ✉E-mail: alkirchhoff@web.de; martin.schlee@uni-bonn.de

recruitment of multiple TANK-binding kinase 1 (TBK1) molecules (Shang et al, 2019; Zhang et al, 2019; Zhao et al, 2019). Local accumulation of TBK1 induces *trans*-autophosphorylation in the activation loop at Ser172, followed by TBK1-dependent phosphorylation of STING at Ser366 in a conserved amino acid sequence motif previously defined as *pLxIS* motif ("p" = ζ, hydrophilic residue; x, any residue according to IUPAC rules) (Zhao et al, 2016; Liu et al, 2015). Binding of tri-/di-phosphorylated dsRNA and long dsRNA species by RIG-I and MDA5, respectively, induces caspase recruitment domain (CARD)-driven multimerization and interaction with mitochondrial antiviral-signaling protein (MAVS) at the cytoplasmic portion of the outer mitochondrial membrane, where MAVS is phosphorylated in the pLxIS motif at Ser442 by TBK1, inhibitor of kappa B kinase-ε (IKKε), or IKKβ (Wu et al, 2014; Xu et al, 2015; Liu et al, 2015; Jiang et al, 2012; Zeng et al, 2010). Thus, the signaling pathways downstream of cGAS and RIG-I/MDA5 converge at the stage of TBK1/IKK. Following, IRF3 is recruited to the phosphorylated pLxIS motifs of STING or MAVS, leading to homodimerization and nuclear translocation of dimeric IRF3 (Liu et al, 2015; Fitzgerald et al, 2003; Panne et al, 2007). Additionally, STING and MAVS induce the activation of nuclear factor kappa-light-chain-enhancer of activated B cells (NF-κB) via IKKα/IKKβ (Ishikawa and Barber, 2008). In concert with other transcription factors, IRF3 and NF-κB drive the expression of type I IFNs and pro-inflammatory cytokines.

Similar to cGAS and RIG-I/MDA5, activation of Toll-like receptor (TLR) 3 or TLR4 leads to the TBK1-dependent phosphorylation of an adaptor protein (TIR-domain-containing adapter-inducing IFN-β (TRIF)) at a consensus pLxIS motif, which serves as a binding site for IRF3 and is thus essential for type I IFN induction. By contrast, stimulation of myeloid differentiation primary response 88 (MyD88)-dependent TLRs such as TLR1/2 also induces TBK1 phosphorylation but does not lead to an IRF3-dependent type I IFN response (Clark et al, 2011a, 2011b). Phosphorylation of TBK1 is therefore necessary but not sufficient to activate IRF3, as this further requires the engagement of appropriate adaptor proteins. Although ligand preferences and molecular functions of cGAS and RIG-I have now been described in detail (Bartok and Hartmann, 2020), it is incompletely understood how signaling proteins shared by unrelated pathways integrate different input signals and induce PRR-specific gene expression programs. Accumulating evidence suggests that central kinases such as the canonical (IKKα, IKKβ) and non-canonical IKKs (TBK1, IKKε) are targeted to distinct signaling complexes in a PRR-dependent manner, thereby enabling the cell to specify and spatiotemporally regulate the antiviral response.

Heterogeneous nuclear ribonucleoprotein M (hnRNPM) has been predominantly described in the context of pre-messenger RNA (pre-mRNA) splicing, cancer biology, or muscle differentiation (Datar et al, 1993; Huelga et al, 2012; Gattoni et al, 1996; Passacantilli et al, 2017). In addition, hnRNPM was described to be targeted by the 3C proteases of Coxsackievirus B3 (CVB3) and Poliovirus (PV) leading to an increased replication of these viruses (Jagdeo et al, 2015) and reported to suppress the expression of a cluster of immune-related genes in RAW 264.7 cells (West et al, 2019). Curiously, hnRNPM was shown to restrict growth of *Listeria monocytogenes* (*L. monocytogenes*) and certain alphaviruses (Semliki Forest virus (SFV), Chikungunya virus (CHIKV)) (Luo et al, 2012; Varjak et al, 2013). Recently, interactions between hnRNPM

and putative ORF3b protein (ORF3B), a potent type I IFN antagonist encoded by severe acute respiratory syndrome coronavirus 1 (SARS-CoV1), have been identified in high-throughput screens (Stukalov et al, 2021; Konno et al, 2020). However, the molecular role of hnRNPM in innate antiviral immunity remains elusive.

In this study, we found that hnRNPM promotes the phosphorylation of IRF3 and expression of type I IFNs induced by both cGAS and RIG-I in a non-redundant manner. Through a combined affinity purification followed by mass spectrometry (AP-MS)-based RNA interference (RNAi) screening approach, we identified ELAV-like protein 1 (ELAVL1; also known as HuR) as an interactor of hnRNPM crucial for IRF3 phosphorylation as well as subsequent induction of type I IFNs and activation of NF-κB. Moreover, we provide evidence that hnRNPM, ELAVL1, TBK1, IKKε, IKKβ, and NF-κB p65 form a multiprotein complex that fuels type I IFN induction by linking cGAS and RIG-I signaling at the stage of IRF3 activation. Intriguingly, pharmacologic inhibition of ELAVL1 potently reduced the constitutive secretion of pro-inflammatory cytokines in skin fibroblasts from patients with type I interferonopathies and dermatomyositis, providing a new approach to treat RIG-I/cGAS-dependent interferonopathies.

# Results

## hnRNPM promotes both the cGAS-STING- and RIG-I-dependent production of type I IFNs

The 78 kDa hnRNPM is a multidomain RNA-binding protein with binding preferences towards G/U-rich intron mRNA and consists of a non-classical basic proline-tyrosine (bPY) nuclear localization sequence, three RNA-recognition motifs (RRMs), a glycine/methionine-rich region (GMG), and a methionine/arginine repeat motif (MR) (Huelga et al, 2012). As hnRNPM is cleaved by certain viral proteases and since viruses often attempt to inactivate components of the nucleic acid-sensing pathways to evade immune surveillance, we set out to explore the role of hnRNPM in host defense. First, we analyzed degradation of hnRNPM by CVB3 and PV 3C proteases. As expected, co-expression of FLAG-tagged variants of these 3C proteases with GFP-tagged hnRNPM in HEK293FT cells potently reduced the levels of transfected hnRNPM (Fig. 1A). To elucidate whether hnRNPM is involved in nucleic acid-sensing, we used a human monocytic cell line (THP-1) with an integrated IFN-stimulated response element (ISRE) reporter as a model system. Usage of a well-characterized ISRE reporter largely avoids detection of post-transcriptional regulation (splicing, untranslated region (UTR)-dependent degradation) and therefore mirrors the activation of IRF transcription factors. Similar to others, we did not obtain homozygous knockout (KO) clones after targeting *HNRNPM* using clustered regularly interspaced short palindromic repeats (CRISPR)-Cas9 in THP-1, suggesting that full depletion of hnRNPM is not tolerated by the cell (West et al, 2019). Thus, we generated THP-1 hnRNPM knockdown (KD) cells using short hairpin RNAs (shRNAs). KD of hnRNPM was confirmed by qPCR and did not impair cellular viability (Fig. 1B,C). Of note, we observed strong inhibition of the ISRE reporter in hnRNPM KD cells after stimulation with agonists for RIG-I (5′-triphosphorylated-dsRNA (5′ppp-dsRNA)), cGAS (plasmid DNA:

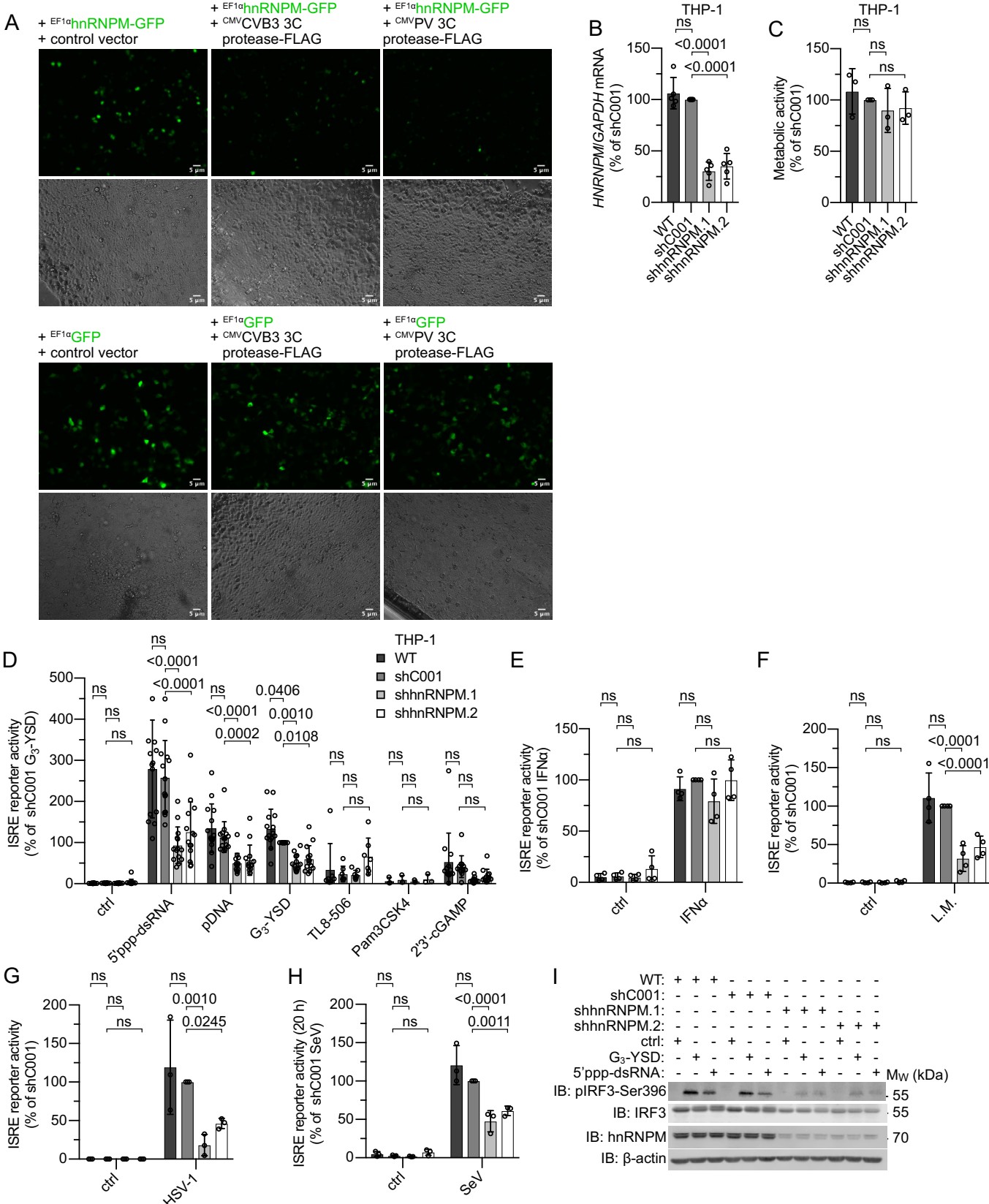

**Figure 1. hnRNPM promotes type I IFN induction downstream of both cGAS and RIG-I.**

(A) Fluorescence microscopy of HEK293FT cells transiently transfected with hnRNPM-GFP (top) or GFP (bottom) together with a control vector or vectors encoding CVB3 3 C protease-FLAG- or PV 3C protease-FLAG (CMV or EF1α promotor). (B) Expression of *HNRNPM* mRNA in THP-1 WT and cells expressing control shRNA (shC001) or hnRNPM-specific shRNAs (shhnRNPM.1, shhnRNPM.2) (from left to right: $n = 5, 5, 5, 5$ independent experiments). (C) MTT assay of the cells depicted in (B) (from left to right: $n = 3, 3, 3, 3$ independent experiments). (D) ISRE reporter activation in the cells depicted in (B) 20 h after stimulation of RIG-I with 5'ppp-dsRNA (0.1 μg/ml), of cGAS with pDNA (0.1 μg/ml) or G$_3$-YSD (0.5 μg/ml), of TLR8 with TL8-506 (1.0 μg/ml), of TLR1/2 with Pam3CSK4 (0.5 μg/ml), or of STING with 2'3'-cGAMP (10 μg/ml). ctrl, non-stimulated (stimuli from left to right: $n = 14, 14, 14, 14, 7, 3, 13$ independent experiments). (E) ISRE reporter activation in the cells depicted in (B) 20 h after stimulation of IFNAR with IFNα (1000 U/ml). ctrl, non-stimulated (stimuli from left to right: $n = 4, 4$ independent experiments). (F) ISRE reporter activation in the cells depicted in (B) 24 h after infection with *L. monocytogenes* (L.M., MOI 1). ctrl, non-stimulated (stimuli from left to right: $n = 4, 4$ independent experiments). (G) ISRE reporter activation in the cells depicted in (B) 24 h after infection with HSV-1 (MOI 5). ctrl, non-stimulated (stimuli from left to right: $n = 3, 3$ independent experiments). (H) ISRE reporter activation in the cells depicted in (B) 24 h after infection with SeV (MOI 1). ctrl, non-stimulated (stimuli from left to right: $n = 3, 3$ independent experiments). (I) Immunoblot analysis of the cells depicted in (B) 3 h after stimulation of cGAS with G$_3$-YSD (0.5 μg/ml) or of RIG-I with 5'ppp-dsRNA (0.1 μg/ml). ctrl, non-stimulated. One representative experiment of two independent experiments is shown. (B, C) Mean ± SD, one-way ANOVA, Dunnett's multiple comparisons test. For (D–H): mean ± SD, two-way ANOVA, Dunnett's multiple comparisons test. ns, $P$ value > 0.05. Source data are available online for this figure.

pDNA, Y-DNA: G$_3$-YSD), or STING (2'3'-cGAMP) (Fig. 1D). Here, pathway inhibition was proportional to the reduction of hnRNPM expression, while ISRE reporter activity induced by stimulation of TLR8 with TL8-506 or IFNα receptor (IFNAR) complex with IFNα was not impaired by KD of hnRNPM (Fig. 1D,E). Similarly, *IFNB1* mRNA expression induced by RIG-I or cGAS was reduced in hnRNPM KD cells compared to control cells (Fig. EV1A). In contrast, *CXCL10* mRNA expression was unchanged between control and hnRNPM KD cells after stimulation of cGAS or RIG-I (Fig. EV1B). Next, we analyzed the role of hnRNPM after infection with *L. monocytogenes*, herpes simplex virus type 1 (HSV-1), and Sendai virus (SeV), which are known to be sensed by cGAS, STING, and/or RIG-I (Abdullah et al, 2012; Hansen et al, 2014; Sun et al, 2013a; Hagmann et al, 2013). Compellingly, hnRNPM KD significantly reduced ISRE reporter induction during infection with *Listeria*, HSV-1, or SeV (Fig. 1F–H). To further investigate the antimicrobial activity of hnRNPM, we asked whether hnRNPM regulates a signaling step downstream of cGAS and RIG-I. We thus analyzed phosphorylation of IRF3 at Ser396 (pIRF3-Ser396) induced by cGAS or RIG-I stimulation. Intriguingly, pIRF3-Ser396 levels elicited by cGAS or RIG-I were markedly reduced in THP-1 hnRNPM KD cells (Figs. 1I and EV1C). By contrast, the IFNα-induced phosphorylation of signal transducer and activator of transcription 1 (STAT1) at Tyr701 (pSTAT1-Tyr701) occurred independently of hnRNPM (Fig. EV1D), further indicating that hnRNPM functions upstream of the IFNAR complex in the interferon-stimulated gene (ISG) induction cascade. In summary, these data suggest that hnRNPM controls both the cGAS- and RIG-I-mediated induction of type I IFNs by promoting IRF3 activation.

## hnRNPM interacts with known innate immunity-associated factors

To further analyze the role of hnRNPM in innate immunity and to identify functionally important interactors that promote signaling downstream of both cGAS-STING and RIG-I-MAVS, we performed AP-MS in combination with an RNAi screen (Fig. 2A). Therefore, we generated THP-1 cells stably expressing C-terminally GFP-tagged hnRNPM or GFP (control) (Fig. 2A). We isolated cell populations with moderate GFP expression levels using fluorescence-activated cell sorting (FACS) to avoid artifacts caused by overexpression. Of note, expression of the hnRNPM-GFP fusion

was weaker compared to endogenous hnRNPM (Fig. EV1E). Additionally, protein expression of the main signaling components of the cGAS and RIG-I pathways was not impaired by KD of hnRNPM in THP-1 cells (Fig. EV1F,G). Next, we affinity-purified GFP from lysates of non-stimulated cells expressing GFP or hnRNPM-GFP and identified bound proteins by liquid chromatography with tandem mass spectrometry (LC-MS/MS) (Fig. 2A). RNA-binding proteins associated to the ribosome (cluster (C) 1) and RNA splicing machinery (C2) were the most abundantly enriched interactors of hnRNPM (Figs. 2B and EV2; Dataset EV1). In addition, hnRNPM interacted with components of the protein folding machinery (C3), mitochondrial transport proteins (C4), and proteins involved in immune system-related processes (C5) or cytokine signaling (C8) (Fig. EV2). Interestingly, we also observed interactions between hnRNPM and proteins that have already been associated with positive regulation of type I IFN expression, including ATP-dependent RNA helicase A (DHX9), pre-mRNA-splicing factor ATP-dependent RNA helicase DHX15 (DHX15), ELAVL1, DNA-dependent protein kinase catalytic subunit (PRKDC; also known as DNA-PKcs), DEAD-box helicase 3 X-linked (DDX3X), and zinc finger CCCH-type antiviral protein 1 (ZC3HAV1) (Fig. 2B) (Burleigh et al, 2020; Ferguson et al, 2012; Hayakawa et al, 2011; Herdy et al, 2015; Kim et al, 2010; Pattabhi et al, 2019; Schröder et al, 2008; Sueyoshi et al, 2018). Notably, we also observed interactions of hnRNPM with IKKβ (IKBKB), which is known to promote type I IFN induction by phosphorylating MAVS in the pLxIS motif (Fig. 2B) (Liu et al, 2015).

## ELAVL1 is a type I IFN-inducing interactor of hnRNPM

To identify binding partners of hnRNPM involved in type I IFN induction, 28 putative interactors, either highly enriched or previously linked to antiviral immunity, were transiently depleted from THP-1 cells by RNAi (targets and pathway annotations are labelled in Fig. 2B) (Fig. EV3A). Next, the generated KD cell lines were stimulated with 5'ppp-dsRNA and pDNA to assess activation of RIG-I and cGAS signaling, respectively (Fig. EV3B). In this study, we focused on the functional evaluation of hits identified in the RNAi screen (Fig. EV3A,B) that positively regulate cGAS and RIG-I signaling in a manner similar to hnRNPM. We selected targets whose depletion by RNAi with two shRNAs reduced both the cGAS- and RIG-I-mediated ISRE reporter activity by at least 50%. Of all hits, only ELAVL1 and HP1BP3 met these criteria

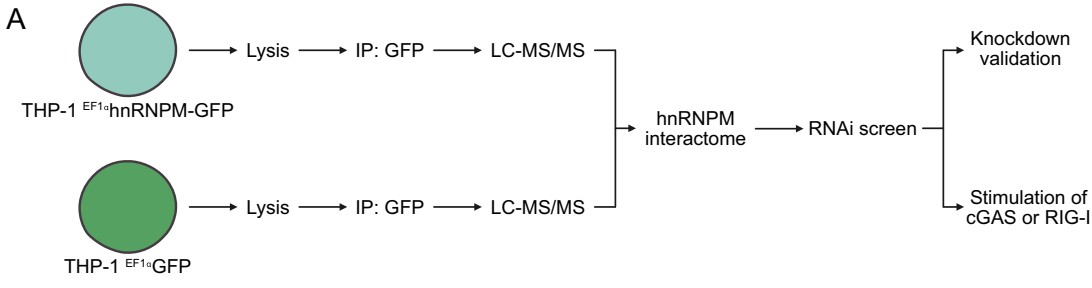

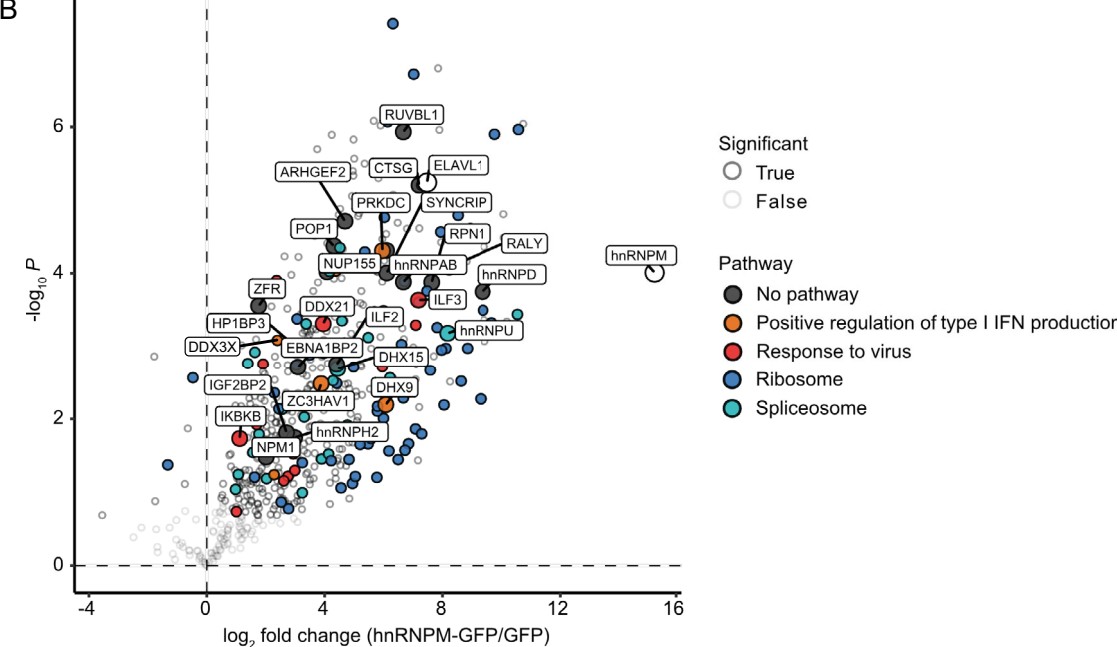

**Figure 2.  Interactome of hnRNPM.**

(A) Schematic workflow of the combined AP-MS-RNAi screening approach to identify immune-relevant interactors of hnRNPM. (B) hnRNPM-GFP and GFP were immunoprecipitated from lysates of non-stimulated THP-1 cells and bound proteins were identified by LC-MS/MS. The Volcano plot shows the $\log_2$ fold change enrichment of proteins detected in the hnRNPM-GFP vs. GFP affinity purification, plotted against the statistical significance ($-\log_{10} P$). Significantly enriched proteins were determined by two-sided Welch's t-tests (S0 = 0.1, permutation-based FDR < 0.05). Colors indicate pathway annotations and the majority of labeled proteins were functionally evaluated by RNAi.

(Figs. 3A and EV3A,B). Since ELAVL1 has already been associated with the expression of type I IFN, we focused our subsequent analyses on this factor (Herdy et al, 2015; Rothamel et al, 2021).

To evaluate the function of hnRNPM and ELAVL1 in primary cells, we performed an shRNA-mediated KD of hnRNPM or ELAVL1 in primary human fibroblasts (Fig. 3B–D). Fibroblasts depleted of hnRNPM or ELAVL1 secreted significantly less CXCL10 after stimulation with 5′ppp-dsRNA or pDNA, indicating that the role of hnRNPM and ELAVL1 in RIG-I and cGAS signaling is not solely restricted to THP-1 but also relevant in primary and non-myeloid human cells.

It has been reported that hnRNPM shuttles from nucleus to cytoplasm after infection with SFV or CHIKV (Varjak et al, 2013). Analysis of the subcellular localization showed that hnRNPM and ELAVL1 are predominantly localized in the nucleus in the non-stimulated condition (Fig. EV3C,D). Stimulation of cGAS or RIG-I

with synthetic ligands did not alter the distribution of hnRNPM or ELAVL1 between nucleus and cytoplasm (Fig. EV3C,D).

We then asked whether hnRNPM and ELAVL1 are components of the same multiprotein complex. Therefore, we used AP-MS to also map the interactome of ELAVL1 (Fig. 3E,F). As expected, hnRNPM was detected in the interactome of ELAVL1 (Fig. 3F). To identify shared interactors of hnRNPM and ELAVL1, we compared the differential interactomes of hnRNPM and ELAVL1 (Fig. 3E,F; Dataset EV1). Among the hnRNPM interactors, 15.7% were also detected in the interactome of ELAVL1, whereas 93.8% of the binding partners of ELAVL1 also interacted with hnRNPM, indicating that hnRNPM and ELAVL1 are part of the same multiprotein complex. We mapped the interactors shared by hnRNPM and ELAVL1 on the human protein-protein interaction network to identify a subnetwork of proteins that is shared by both baits (Fig. 3F). Interestingly, several common interactors have

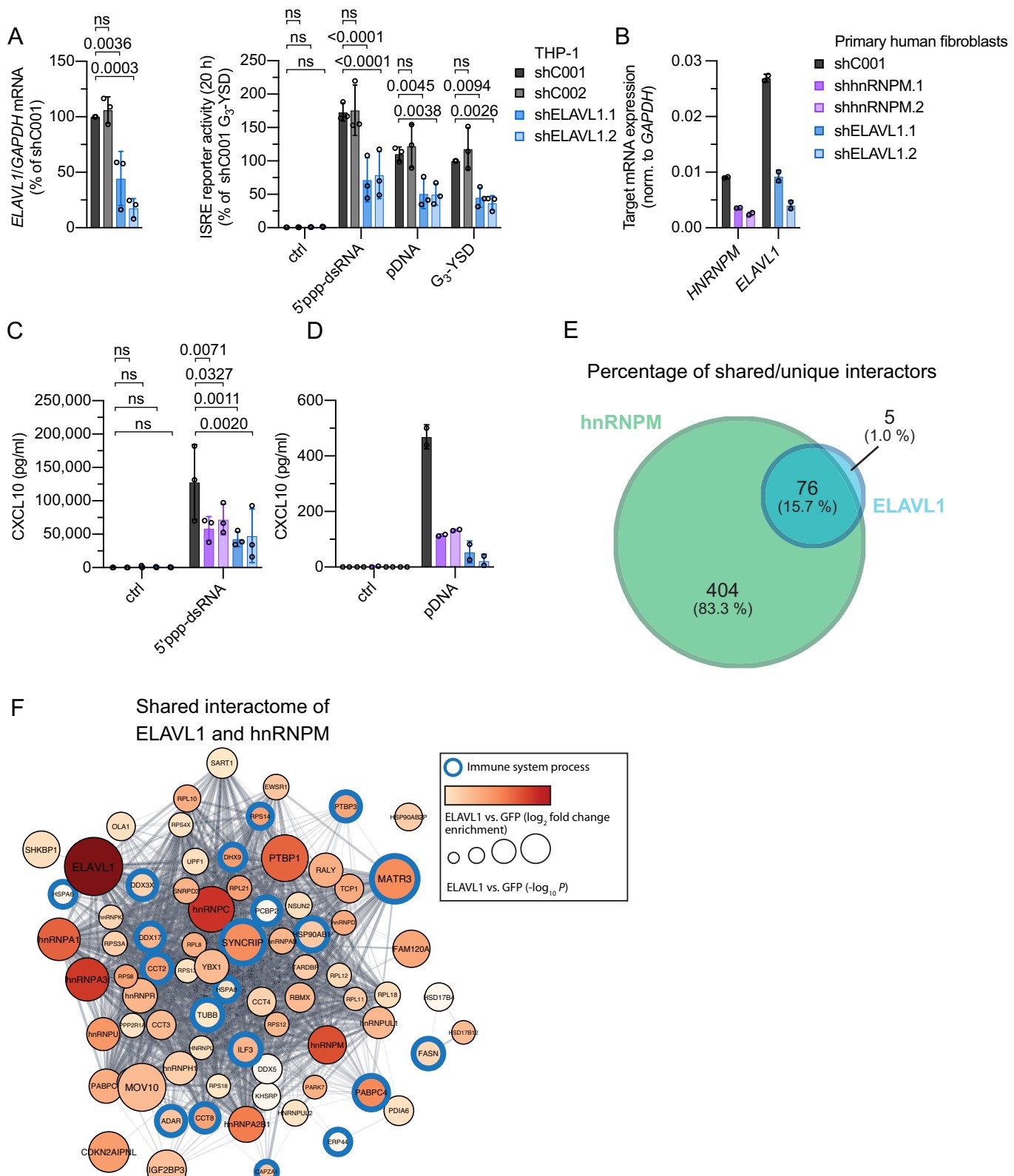

**Figure 3.  ELAVL1 interacts with hnRNPM to regulate type I IFN induction downstream of both cGAS and RIG-I.**

(A) Left panel: Expression of *ELAVL1* mRNA in THP-1 cells expressing control shRNAs (shC001, shC002) or ELAVL1-specific shRNAs (shELAVL1.1, shELAVL1.2) (mean ± SD, one-way ANOVA, Dunnett's multiple comparisons test; from left to right: $n = 3, 3, 3, 3$ independent experiments), right panel: ISRE reporter activation 20 h after stimulation with 5'ppp-dsRNA (0.1 μg/ml), pDNA (0.1 μg/ml), or $G_3$-YSD (0.5 μg/ml) (mean ± SD, two-way ANOVA, Dunnett's multiple comparisons test; stimuli from left to right: $n = 3, 3, 3$ independent experiments). ctrl, non-stimulated. (B) Expression of *HNRNPM* and *ELAVL1* mRNA in primary human fibroblasts stably expressing control shRNA (shC001) or shRNAs against *HNRNPM* or *ELAVL1* (shRNA.1, shRNA.2) (mean ± SD; from left to right: $n = 2, 2, 2, 2, 2, 2$ independent experiments). (C) CXCL10 ELISA with supernatants of the cells depicted in (B) 20 h after stimulation with 5'ppp-dsRNA (0.1 μg/ml). ctrl, non-stimulated (mean ± SD, two-way ANOVA, Dunnett's multiple comparisons test; stimuli from left to right: $n = 3, 3$ independent experiments). (D) CXCL10 ELISA with supernatants of the cells depicted in (B) 20 h after stimulation with pDNA (1 μg/ml). ctrl, non-stimulated (stimuli from left to right: $n = 2, 2$ independent experiments). (E) hnRNPM-GFP, ELAVL1-GFP, and GFP were immunoprecipitated from lysates of non-stimulated THP-1 cells and bound proteins were identified by LC-MS/MS. Proteins detected in hnRNPM and ELAVL1 precipitates were statistically compared to GFP as control using a two-sided Welch's *t* test (S0 = 0.1, permutation-based FDR < 0.05). The Venn diagram shows the absolute and relative proportions of unique and shared interactors of hnRNPM and ELAVL1. (F) Shared interaction network of hnRNPM and ELAVL1. Blue circles indicate proteins annotated with the gene ontology (GO) term immune system process (GO.0002376, FDR = 0.0034). The statistical tests used are described in detail in the methods. ns, *P* value > 0.05. Source data are available online for this figure.

previously been connected to immune system-related processes. Given this high enrichment of proteins related to immune system regulation, we speculate that additional, yet undiscovered regulators of cGAS and RIG-I signaling may exist within this network. However, their functional evaluation is beyond the scope of this manuscript. In summary, we identified hnRNPM and its binding partner ELAVL1 as novel components of both the cGAS and RIG-I signaling pathways, which non-redundantly promote the induction of type I IFNs.

## ELAVL1 non-redundantly promotes type I IFN induction and NF-κB activation elicited by both cGAS- and RIG-I

To further evaluate the role of ELAVL1, we targeted the *ELAVL1* gene locus in THP-1 cells by CRISPR-Cas9 with two different guide RNAs (gRNAs) (clones #1–2, ELAVL1 gRNA AN; clone 3, ELAVL1 gRNA AI) (Fig. 4A). As expected, KO of ELAVL1 severely compromised ISRE reporter activation after stimulation of cGAS or RIG-I (Fig. 4B). In contrast, activation of the ISRE reporter by recombinant IFNα was unchanged between wildtype (WT) and ELAVL1 KO cells, indicating that ELAVL1 does not influence IFNAR signaling (Fig. 4B).

To exclude off-target effects, ELAVL1-FLAG was lentivirally expressed in THP-1 ELAVL1 KO clone #1. As expected, expression of ELAVL1-FLAG but not GFP rescued ISRE reporter activation in cells with ELAVL1 KO genetic background after stimulation of RIG-I, cGAS, or STING, demonstrating that ELAVL1 is a specific positive regulator downstream of these receptors (Fig. 4C–E). Reconstitution of ELAVL1 expression also re-established the RIG-I- and cGAS-dependent activation of the NF-κB reporter, while NF-κB signaling induced by TLR1/2 with Pam3CSK4 was ELAVL1-independent (Fig. 4F). To evaluate the contribution of ELAVL1 to the activation of the respective pathways, we compared THP-1 ELAVL1 KO cells with RIG-I KO, MAVS KO, and cGAS KO cells. As expected, KO of RIG-I or MAVS abolished both 5'ppp-dsRNA-induced ISRE and NF-κB reporter activation, whereas KO of cGAS specifically disrupted the pDNA- or $G_3$-YSD-induced activation of these reporters (Fig. EV4A–C). As shown above, KO of ELAVL1 inhibited the cGAS- or RIG-I-induced ISRE reporter activity by 5–8-fold, suggesting that ELAVL1, while not an essential component, serves as a potent enhancer of these signaling pathways. Although ELAVL1 is involved in different RNA-processing steps, ELAVL1-deficient cells did not display decreased viability

(Fig. EV4D). In fact, THP-1 ELAVL1 KO monocytes were more viable after stimulation of RIG-I or cGAS (Fig. EV4D). In addition, we found that KO of ELAVL1 strongly reduced the expression of *IFNB1* and *CXCL10* mRNA following activation of cGAS or RIG-I (Fig. 4G,H). In line with this observation, secreted C-X-C motif chemokine ligand 10 (CXCL10) was also undetectable in the supernatants of ELAVL1 KO cells after activation of cGAS or RIG-I (Fig. EV4E). Similar to the ISRE reporter gene assay, cGAS- or RIG-I-mediated induction of *IFIT1* mRNA was reduced but not completely abolished in THP-1 ELAVL1 KO cells (Fig. 4I). By contrast, *IFIT1* mRNA expression induced by recombinant IFNα was unchanged in ELAVL1 KO cells compared to WT, further demonstrating that IFNAR signaling is not influenced by ELAVL1 and highlighting the specific role of ELAVL1 in the cGAS and RIG-I signaling cascades (Fig. 4I). In contrast to ISRE and NF-κB reporter activation as well as *IFNB1 and IFIT1* mRNA expression, CXCL10 protein expression induced by cGAS activation with pDNA or $G_3$-YSD was not rescued by overexpression of ELAVL1-FLAG in THP-1 ELAVL1 KO cells (Fig. EV4E), suggesting clonal artifacts for this specific readout.

In addition, we found that the phosphorylation of IRF3, STING, and TBK1 upon activation of RIG-I or cGAS was ELAVL1-dependent (Figs. 4J and EV4F–H). Because the Pam3CSK4-induced NF-κB reporter activity was independent of ELAVL1, we also analyzed the TLR1/2-mediated pTBK1-Ser172 induction in THP-1 ELAVL1 KO cells. Interestingly, the Pam3CSK4-induced phosphorylation of TBK1 at Ser172 was not impaired by KO of ELAVL1 (Figs. 4K and EV4I). These data suggest that ELAVL1 specifically promotes TBK1 phosphorylation induced by MAVS- or STING- but not MyD88-dependent pathways and that ELAVL1 is able to discriminate TBK1 molecules in a PRR-dependent manner. We then analyzed the role of ELAVL1 during infection with DNA and RNA viruses. Similar to hnRNPM, ELAVL1 was required for both HSV-1 and SeV-induced ISRE reporter activation (Fig. 4L,M).

Considering that the relative mRNA levels of *IFNB1*, *CXCL10*, and *IFIT1* were downregulated to different extents in ELAVL1 KO as compared to control cells, we hypothesized that ELAVL1 has a dual function in innate immunity. On the one hand, ELAVL1 modulates TBK1 and IRF3 activation, and, on the other hand, it may also stabilize certain mRNA species as reported previously for *IFNB1* or certain ISGs (Herdy et al, 2015; Rothamel et al, 2021). Stabilization of *IFNB1* mRNA by ELAVL1 could have direct consequences for the detection of cytoplasmic nucleic acids, as type

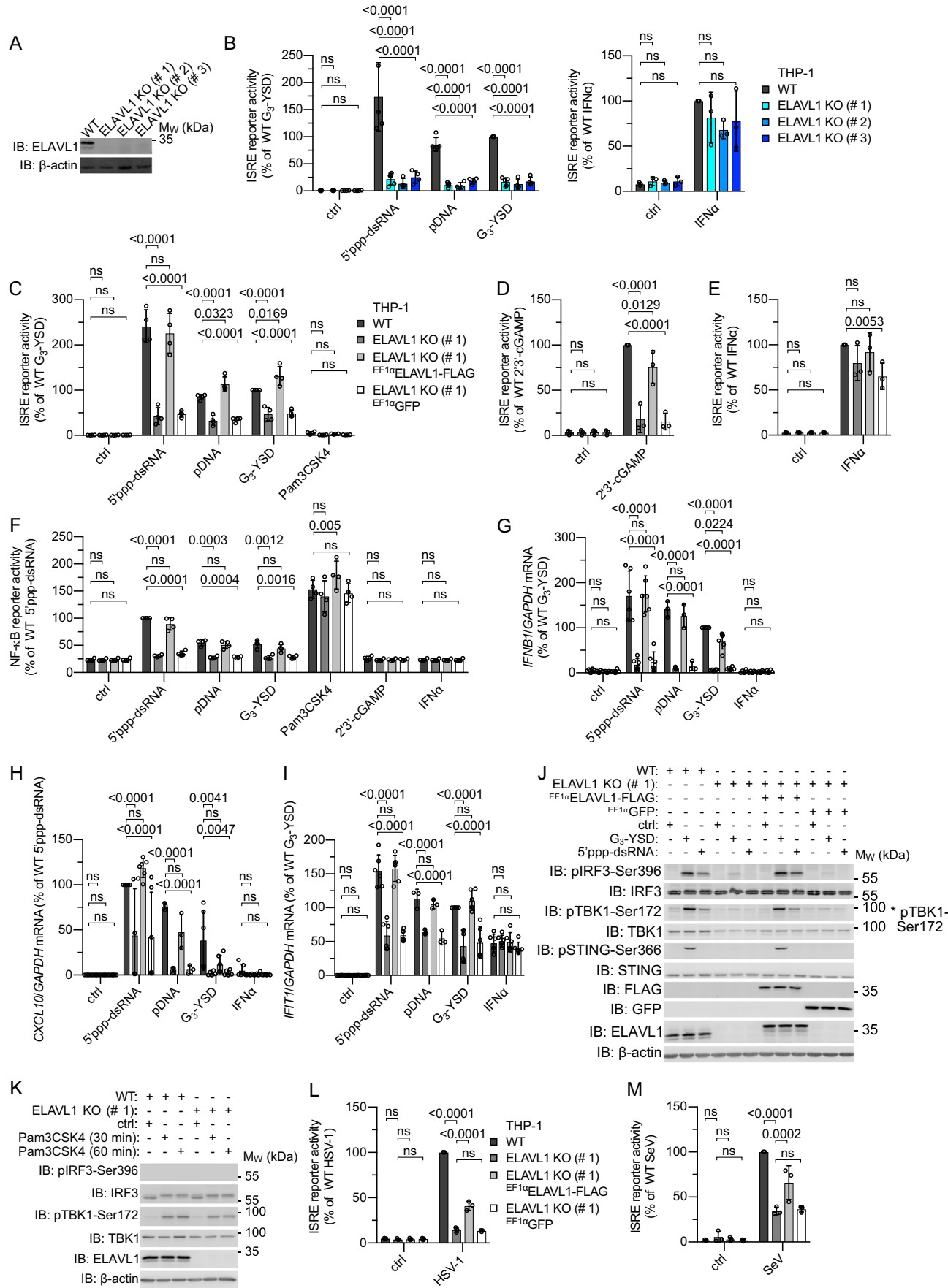

**Figure 4.   ELAVL1 promotes the cGAS- and RIG-I-mediated induction of type I IFNs and activation of NF-κB.**

(A) Immunoblot analysis of THP-1 WT and ELAVL1 KO cells using an ELAVL1-specific antibody. (B) ISRE reporter activation in the cells depicted in (A) 20 h after stimulation with 5′ppp-dsRNA (0.1 μg/ml), pDNA (0.1 μg/ml), G₃-YSD (0.5 μg/ml), or IFNα (1000 U/ml). ctrl, non-stimulated (left panel: stimuli from left to right, $n = 4$, 4, 4, 4 independent experiments; right panel: stimuli from left to right, $n = 3$, 3 independent experiments). (C) ISRE reporter activation in THP-1 WT, ELAVL1 KO (#1), and ELAVL1 KO (#1) expressing ELAVL1-FLAG or GFP 20 h after stimulation with 5′ppp-dsRNA (0.1 μg/ml), pDNA (0.1 μg/ml), G₃-YSD (0.5 μg/ml), or Pam3CSK4 (0.5 μg/ml). ctrl, non-stimulated (stimuli from left to right: $n = 4$, 4, 4, 4, 4 independent experiments). (D) ISRE reporter activation in the cells depicted in (C) 20 h after stimulation with 2′3′-cGAMP (10 μg/ml). ctrl, non-stimulated (stimuli from left to right: $n = 3$, 3 independent experiments). (E) ISRE reporter activation in the cells depicted in (C) 20 h after stimulation with IFNα (1000 U/ml). ctrl, non-stimulated (stimuli from left to right: $n = 3$, 3 independent experiments). (F) NF-κB reporter activation in the cells depicted in (C) 20 h after stimulation with 5′ppp-dsRNA (0.1 μg/ml), pDNA (0.1 μg/ml), G₃-YSD (0.5 μg/ml), Pam3CSK4 (0.5 μg/ml), 2′3′-cGAMP (10 μg/ml), or IFNα (1000 U/ml). ctrl, non-stimulated (stimuli from left to right: $n = 4$, 4, 4, 4, 4, 4 independent experiments). (G) Expression of *IFNB1* mRNA in the cells depicted in (C) 6 h after stimulation with 5′ppp-dsRNA (0.1 μg/ml), pDNA (0.1 μg/ml), G₃-YSD (0.5 μg/ml), or IFNα (1000 U/ml). ctrl, non-stimulated (stimuli from left to right: $n = 6$, 6, 3, 6, 6 independent experiments). (H) Expression of *CXCL10* mRNA in the cells depicted in (C) 6 h after stimulation with 5′ppp-dsRNA (0.1 μg/ml), pDNA (0.1 μg/ml), G₃-YSD (0.5 μg/ml), or IFNα (1000 U/ml). ctrl, non-stimulated (stimuli from left to right: $n = 6$, 6, 3, 6, 6 independent experiments). (I) Expression of *IFIT1* mRNA in the cells depicted in (C) 6 h after stimulation with 5′ppp-dsRNA (0.1 μg/ml), pDNA (0.1 μg/ml), G₃-YSD (0.5 μg/ml), or IFNα (1000 U/ml). ctrl, non-stimulated (stimuli from left to right: $n = 6$, 6, 3, 6, 6 independent experiments). (J) Immunoblot analysis of the cells depicted in (C) 3 h after stimulation with G₃-YSD (0.5 μg/ml) or 5′ppp-dsRNA (0.1 μg/ml). ctrl, non-stimulated. (K) Immunoblot analysis of THP-1 WT and ELAVL1 KO (#1) cells after stimulation with Pam3CSK4 (0.5 μg/ml). ctrl, non-stimulated. (L) ISRE reporter activation in the cells depicted in (C) 24 h after infection with HSV-1 (MOI 5). ctrl, non-stimulated (stimuli from left to right: $n = 3$, 3 independent experiments). (M) ISRE reporter activation in the cells depicted in (C) 24 h after infection with SeV (MOI 1). ctrl, non-stimulated (stimuli from left to right: $n = 3$, 3 independent experiments). For (B–I, L, M): mean ± SD, two-way ANOVA, Dunnett's multiple comparisons test. For (J, K): One representative experiment of at least two independent experiments is shown. ns, $P$ value > 0.05. Source data are available online for this figure.

I IFNs themselves enhance the nucleic acid-sensing capacities of the cell in a positive feedback loop by upregulating the expression of first category nucleic acid receptors such as RIG-I or, to a lesser extent, cGAS. To quantify the role of ELAVL1 beyond *IFNB1*-dependent effects, we generated THP-1 cells lacking both IFNAR2 and ELAVL1. We hypothesized that inhibition of cGAS and RIG-I signaling in THP-1 IFNAR2 ELAVL1 double-KO cells indicates an active role of ELAVL1 in signal transduction. Although, as expected, we observed a reduction in ISRE activation in IFNAR2 KO cells compared to WT, additional deficiency of ELAVL1 substantially inhibited both cGAS and RIG-I signaling compared to IFNAR2 single KO cells (Fig. EV5A). Similar observations were made after KD of hnRNPM in THP-1 IFNAR2 KO cells (Fig. EV5B).

Considering that ELAVL1 is an RNA-binding protein involved in different RNA-processing steps, we performed 3′-mRNA sequencing to determine whether KO of ELAVL1 impairs the overall expression of ISGs induced by IFNα. Here, no significant changes in the upregulation of ISGs were detected between WT and ELAVL1 KO cells, further suggesting that ELAVL1 is uncoupled from the IFNAR pathway and the processing of ISG RNAs (Fig. EV5C; Dataset EV2). Similarly, transcript levels of the main components of the cGAS and RIG-I signaling pathways were unchanged between ELAVL1 KO and THP-1 WT cells (Fig. EV5C; Dataset EV2).

In summary, these data show that ELAVL1 promotes type I IFN expression and NF-κB signaling downstream of RIG-I and cGAS-STING in a non-redundant fashion. We hypothesize that ELAVL1 and hnRNPM couple both pathways by regulating a converging signaling step that is shared by both pathways.

## hnRNPM forms a multiprotein complex with ELAVL1, TBK1, IKKε, IKKβ, and NF-κB p65

Our data suggest that hnRNPM and ELAVL1 are part of a protein complex that regulates the induction of type I IFNs downstream of both cGAS and RIG-I but upstream of IRF3 phosphorylation. Since we identified IKKβ as an interactor of hnRNPM by AP-MS

(Fig. 2B), we tested for interactions with structurally/functionally related antiviral signaling molecules such as IKKα, IKKε, or TBK1 using co-immunoprecipitation (co-IP) (Fig. 5A). As expected, ELAVL1 co-precipitated with hnRNPM. Co-precipitation of ELAVL1 with hnRNPM was detected in non-stimulated cells as well as after activation of cGAS and co-precipitation decreased upon RNase A treatment, suggesting RNA-dependent interactions. In line with our AP-MS data, we observed specific co-precipitation of IKKβ with hnRNPM (Fig. 5A). Intriguingly, the IKK-related kinases TBK1 and IKKε as well as NF-κB p65 also selectively co-precipitated with hnRNPM (Fig. 5A). Interactions of hnRNPM with IKKβ, TBK1, IKKε, and NF-κB p65 were detectable in untreated and cGAS-activated cells. These interactions were independent of RNA since RNase A treatment did not affect precipitation efficacy. pTBK1-Ser172 and NF-κB p65 phosphorylated at Ser536 (NF-κB pp65-Ser536) were also enriched after IP of hnRNPM and interactions decreased upon RNase A treatment. Compared to untreated cells, cGAS activation increased hnRNPM-associated phosphorylation of TBK1 and NF-κB p65. Although IKKα shares 51% sequence identity with IKKβ (Mercurio et al, 1997), we were unable to detect co-precipitation of IKKα with hnRNPM, demonstrating that hnRNPM has a higher specificity for IKKε and IKKβ. Similarly, no interactions of hnRNPM with pIRF3-Ser396, IRF3, cGAS, and β-actin were detected. Although STING phosphorylated at Ser366 (pSTING-Ser366) was not detected in hnRNPM precipitates, low quantities of total STING co-immunoprecipitated with hnRNPM (Fig. 5A). To provide a second layer of specificity, we immunoprecipitated transiently expressed ELAVL1-FLAG from lysates of HEK293FT cells and screened for interactions of ELAVL1 with the newly identified hnRNPM-associated proteins (TBK1, IKKε, IKKβ, NF-κB p65) (Fig. 5B). As expected, endogenous hnRNPM co-immunoprecipitated with ELAVL1, with interactions decreasing slightly upon treatment with RNase A. In line with our hypothesis that hnRNPM and ELAVL1 contribute to the same protein complex, TBK1, IKKβ, and NF-κB p65 also interacted with ELAVL1-FLAG in an RNase A-independent manner. By contrast, interaction of ELAVL1 with IKKε was RNA-dependent. Altogether, these data confirm that

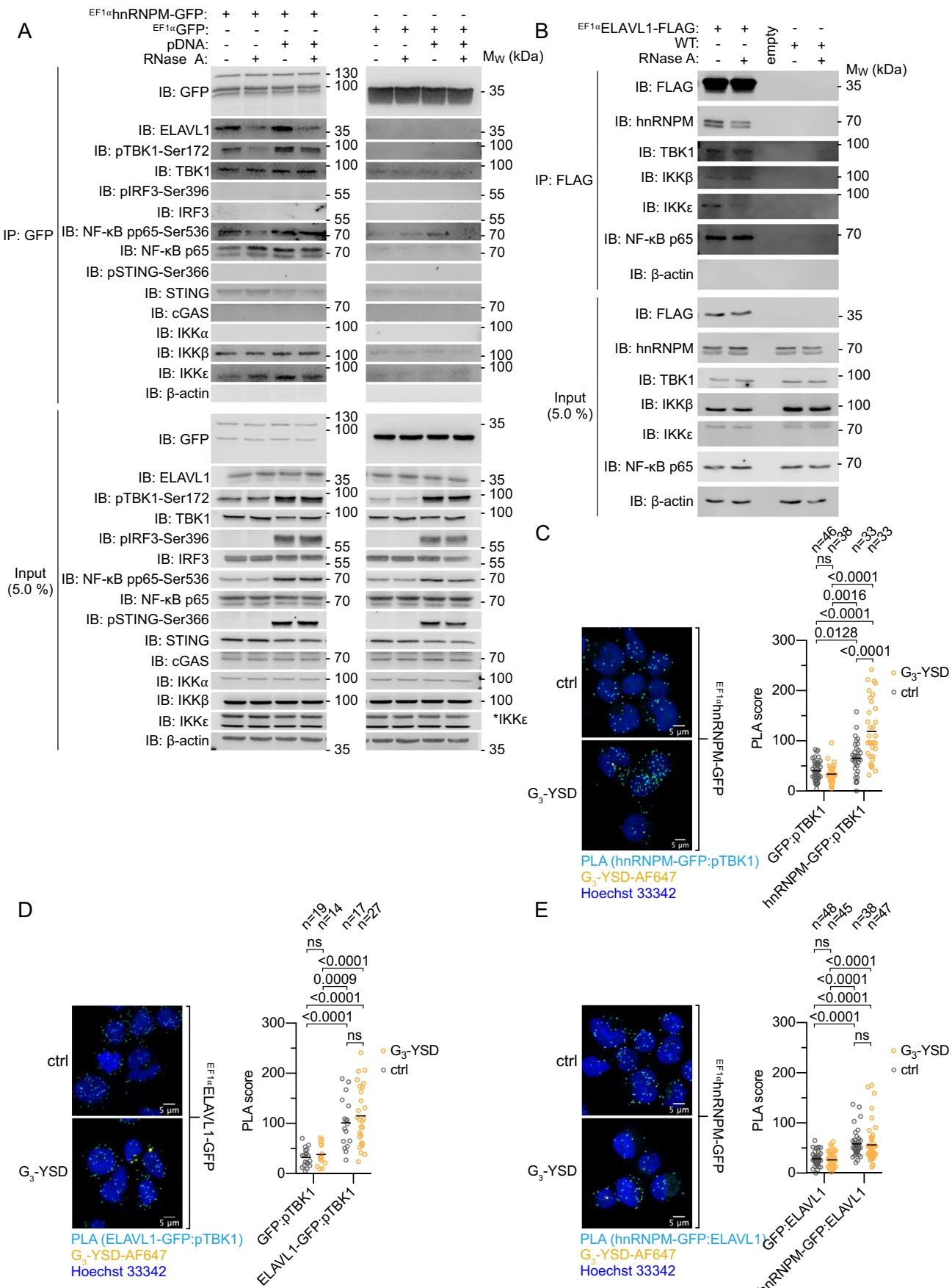

**Figure 5.  hnRNPM and ELAVL1 interact with TBK1, IKKε, IKKβ, and NF-κB p65.**

(A) GFP-specific beads were used to immunoprecipitate hnRNPM-GFP and GFP from lysates of non-stimulated or pDNA-stimulated (0.1 μg/ml, 3 h) THP-1 cells. If indicated, IPs were treated with RNase A (100 μg/ml, 1.5 h). The IPs and 5.0% of the cleared cellular lysate used for IP (input) were analyzed by immunoblotting with the indicated antibodies. IPs and input controls of hnRNPM-GFP and GFP were analyzed on the same membrane, with empty lanes removed. (B) FLAG tag-specific beads were incubated with lysates of WT or ELAVL1-FLAG-expressing HEK293FT cells. If indicated, the IPs were treated with RNase A (100 μg/ml, 1.5 h). IPs and input controls were analyzed by immunoblotting with the indicated antibodies. (C) Differentiated THP-1 cells stably expressing hnRNPM-GFP or GFP were transfected with Alexa Fluor 647-labelled $G_3$-YSD (0.5 μg/ml, 4 h) or left non-stimulated (ctrl). Anti-GFP and anti-pTBK1-Ser172 monoclonal antibodies were used to analyze the interactions between hnRNPM-GFP/GFP and pTBK1-Ser172 *in cellulo* by PLA. Z stack images were recorded on a confocal microscope. Image sequences of hnRNPM-GFP-expressing cells are shown as maximum intensity projections (left panel). The PLA signals per cell in different focal planes were counted and the counted number of puncta per cell was defined as the PLA score (right panel). hnRNPM-GFP:pTBK1, interactions between hnRNPM-GFP and pTBK1-Ser172; GFP:pTBK1, interactions between GFP and pTBK1-Ser172. Blue, Hoechst 33342; yellow, $G_3$-YSD-AF647; cyan, PLA signal. (D) The PLA was performed as described in (C) with differentiated THP-1 cells stably expressing ELAVL1-GFP or GFP. (E) The PLA was performed as described in (C) using anti-GFP and anti-ELAVL1 monoclonal antibodies to analyze the interactions between hnRNPM-GFP/GFP and ELAVL1. For (A, B): One representative experiment of two independent experiments is shown. For (C–E): mean, two-way ANOVA, Tukey's multiple comparisons test; *n* is the number of analyzed cells. ns, *P* value > 0.05. Source data are available online for this figure.

hnRNPM and ELAVL1 form a complex with TBK1, IKKε, IKKβ, and NF-κB p65.

To determine whether the interactions between hnRNPM, ELAVL1, and pTBK1-Ser172 are direct or indirect, and to clarify whether these interactions also occur *in cellulo*, we performed proximity ligation assay (PLA) in differentiated THP-1 cells expressing hnRNPM-GFP, ELAVL1-GFP, or GFP, where positive signals occur only when the GFP-specific antibody is in close vicinity (≤ 40 nm) to the pTBK1-Ser172-specific antibody. It should be noted that a low PLA score was also detectable in GFP-expressing cells (Fig. 5C,D). However, significantly higher PLA scores were observed in cells expressing hnRNPM-GFP or ELAVL1-GFP (hnRNPM-GFP:pTBK1 or ELAVL1-GFP:pTBK1) as compared to GFP (GFP:pTBK1), indicating close interactions of hnRNPM and ELAVL1 with pTBK1-Ser172 in cells (Fig. 5C,D). For hnRNPM-GFP, the PLA score further increased upon cytosolic challenge with $G_3$-YSD (Fig. 5C). Notably, interactions between ELAVL1 and pTBK1-Ser172 were also detectable under non-stimulated conditions (Fig. 5D). We also confirmed close interaction between hnRNPM and ELAVL1 (Fig. 5E). Here, we performed the PLA in hnRNPM-GFP- or GFP-expressing cells using a combination of GFP- and ELAVL1-specific antibodies and observed a significantly higher PLA score for hnRNPM-GFP:E-LAVL1 interaction as compared to GFP:ELAVL1 (Fig. 5E). The interactions between hnRNPM and ELAVL1 were present in both cytoplasm and nucleus. In summary, our data link for the first time hnRNPM and ELAVL1 to active antiviral signaling proteins and provide evidence for the existence of a previously undiscovered protein complex that fuels the induction of type I IFNs upon stimulation of cGAS or RIG-I.

## Inhibition of ELAVL1 with MS-444 ameliorates auto-inflammation in cells from patients with type I interferonopathies and dermatomyositis

Next, we evaluated whether the compound MS-444, a well-characterized homodimerization inhibitor of ELAVL1, can inhibit cGAS and RIG-I signaling to a similar extent as KO of ELAVL1 (Meisner et al, 2007). Of note, MS-444 inhibited both cGAS and RIG-I signaling in THP-1 cells in a dose-dependent manner (Fig. 6A,B). In addition, we found that MS-444 potently inhibited the phosphorylation of IRF3 at Ser396 after stimulation of both cGAS and RIG-I (Fig. 6C). Similar to ELAVL1 KO cells, MS-444

treatment did not impair the IFNα-induced ISRE reporter activity, the TLR1/TLR2-induced NF-κB reporter activity, and the viability the cells (Appendix Fig. S1A–C).

Following, we used primary human fibroblasts to further analyze the inhibitory effect of MS-444 on cGAS and RIG-I signaling. We found that MS-444 inhibited both the cGAS- and RIG-I-mediated secretion of CXCL10 in these cells (Fig. 6D,E). By contrast, MS-444 only modestly impaired TLR1/TLR2-induced secretion of interleukin 6 (IL-6) and did not reduce cell viability (Appendix Fig. S1D,E).

Finally, we wanted to assess whether inhibition of ELAVL1 by MS-444 can reduce the constitutive secretion of pro-inflammatory cytokines in cells from patients suffering from pathological activation of STING- or MAVS-dependent pathways. Therefore, we isolated fibroblasts from patients with Aicardi-Goutières syndrome (AGS) (mutation in adenosine deaminases acting on RNA (ADAR)) and STING-associated vasculopathy with onset in infancy (SAVI)/chilblain lupus (mutation in STING). We also tested fibroblasts of dermatomyositis patients, a type I IFN-associated auto-inflammatory disease of unknown origin. Remarkably, MS-444 treatment potently reduced the constitutive secretion of CXCL10 and tumor necrosis factor α (TNF-α) in these patient cells (Fig. 6F,G) without impairing viability (Appendix Fig. S1F), suggesting that pharmacological inhibition of ELAVL1 may represent a promising strategy to ameliorate auto-inflammation caused by dysregulated activation of STING- or MAVS-dependent pathways.

Overall, our data show that hnRNPM and ELAVL1 form a multiprotein complex with TBK1, IKKε, IKKβ, and NF-κB p65 that interconnects the cGAS and RIG-I signaling pathways by regulating the phosphorylation of IRF3, thus representing a previously undescribed convergence point of both signaling branches that can be targeted with the inhibitor MS-444 (Fig. 6H).

## Discussion

cGAS and RIG-I sense different ligands and signal via distinct adaptor proteins, but both their signaling cascades converge at the level of TBK1/IKK/IRF3/NF-κB p65, ultimately leading to the expression of type I IFNs. In the present study, we report the identification of a novel protein complex that promotes the expression of type I IFNs downstream of both cGAS and RIG-I.

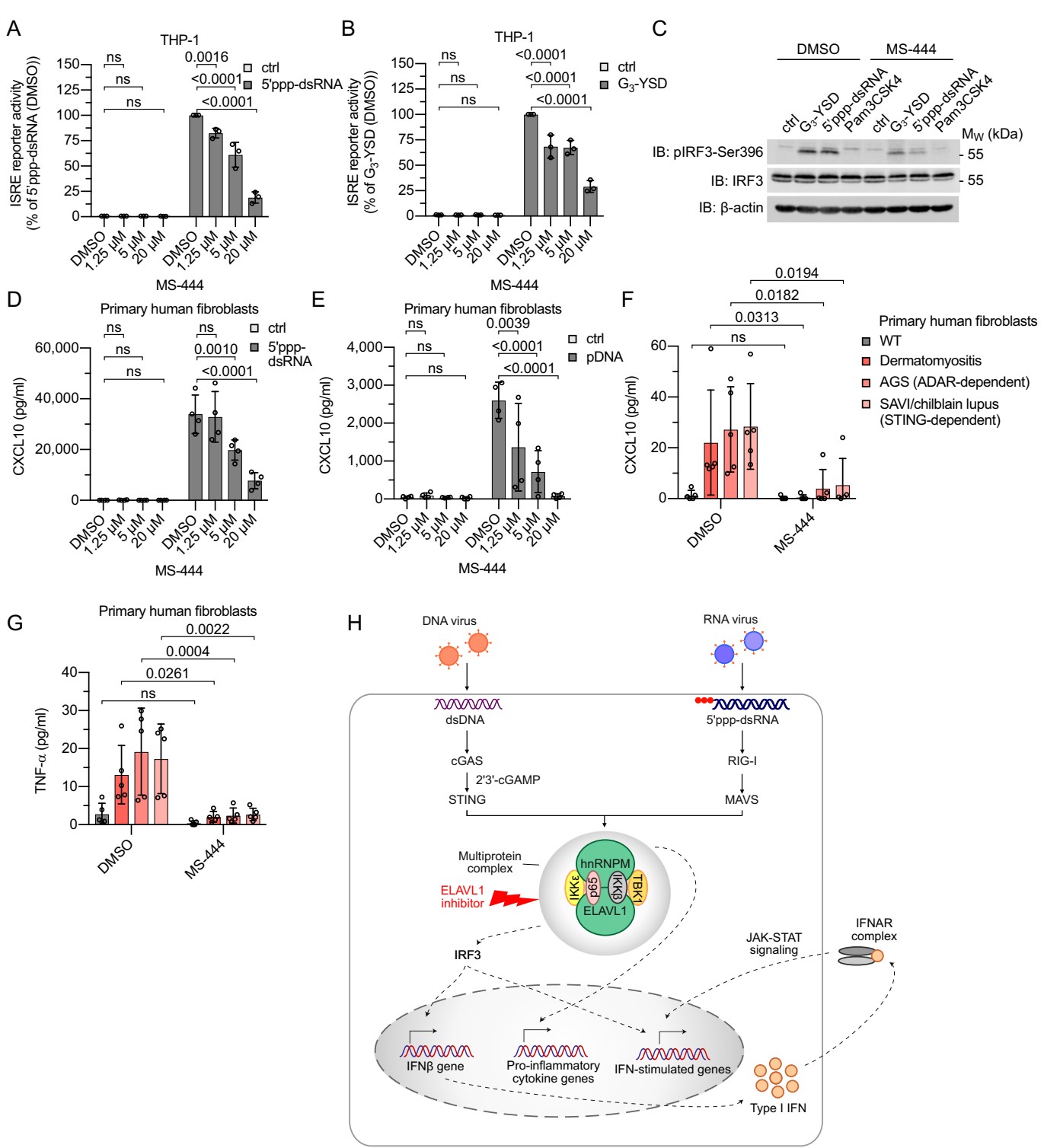

hnRNPM and ELAVL1 constitute two key components of this complex that non-redundantly controls the phosphorylation of IRF3 after cytosolic challenge with ligands for cGAS or RIG-I. We could further show that the protein complex assembled by hnRNPM and ELAVL1 interacts with crucial signaling components downstream of cGAS and RIG-I, such as TBK1, IKKε, IKKβ, and NF-κB p65, thereby merging both signaling pathways. It is

conceivable that hnRNPM and ELAVL1 provide a scaffolding platform that stabilizes interactions and amplifies the phosphorylation of IRF3, thereby promoting type I IFN expression. Considering that hnRNPM and ELAVL1 share several binding partners, our AP-MS analysis strongly suggests that both proteins are part of the same multiprotein complex. This notion is further supported by our PLA data, demonstrating interactions between hnRNPM and

◄

**Figure 6.  MS-444 reduces constitutive secretion of inflammatory cytokines in cells from patients with dermatomyositis, AGS, and SAVI/chilblain lupus.**

(A) THP-1 cells were pre-treated with MS-444/DMSO for 2 h and then stimulated with 5′ppp-dsRNA (0.1 μg/ml) or left non-stimulated (ctrl) in MS-444/DMSO-containing medium. After 16 h, ISRE reporter activation was determined (from left to right: n = 3, 3, 3, 3, 3, 3, 3, 3 independent experiments). (B) THP-1 cells were pre-treated with MS-444/DMSO for 2 h and then stimulated with $G_3$-YSD (0.5 μg/ml) or left non-stimulated (ctrl) in MS-444/DMSO-containing medium. After 16 h, ISRE reporter activation was determined (from left to right: n = 3, 3, 3, 3, 3, 3, 3, 3 independent experiments). (C) Immunoblot analysis of THP-1 cells pre-treated with 20 μM MS-444 or DMSO for 16 h and then stimulated for 1 h with $G_3$-YSD (0.5 μg/ml), 5′ppp-dsRNA (0.1 μg/ml), Pam3CSK4 (0.5 μg/ml), or left non-stimulated (ctrl). One representative experiment of four independent experiments is shown. (D) Primary human fibroblasts were pre-treated with MS-444/DMSO for 2 h and then stimulated with 5′ppp-dsRNA (0.1 μg/ml) or left non-stimulated (ctrl) in MS-444/DMSO-containing medium. After 16 h, CXCL10 secretion was analyzed. ctrl, non-stimulated (from left to right: n = 4, 4, 4, 4, 4, 4, 4, 4 independent experiments). (E) Primary human fibroblasts were pre-treated with MS-444/DMSO for 2 h and then stimulated with pDNA (1 μg/ml) or left non-stimulated (ctrl) in MS-444/DMSO-containing medium. After 16 h, CXCL10 secretion was analyzed. ctrl, non-stimulated (from left to right: n = 4, 4, 4, 4, 4, 4, 4, 4 independent experiments). (F) Primary human fibroblasts from a healthy donor (WT) or from patients with dermatomyositis, AGS, or SAVI/chilblain lupus were treated for 16 h with 20 μM MS-444 or DMSO and then CXCL10 secretion was analyzed (from left to right: n = 5, 5, 5, 5, 5, 5, 5, 5 independent experiments). (G) TNFα ELISA with supernatants of the cells depicted in (F) (from left to right: n = 5, 5, 5, 5, 5, 5, 5, 5 independent experiments). (H) Proposed model of how hnRNPM and ELAVL1 regulate signaling downstream of cGAS and RIG-I. For (A, B, D, E): mean ± SD, two-way ANOVA, Dunnett's multiple comparisons test. For (F, G): mean ± SD, two-way ANOVA, Šídák's multiple comparisons test. ns, P value > 0.05. Source data are available online for this figure.

ELAVL1 in cells. Interestingly, we also identified a number of proteins already connected to type I IFN induction as hnRNPM interactors, such as DDX3X, DHX9, DHX15, PRKDC, and ZC3HAV1. However, KD of these proteins did not significantly inhibit ISRE reporter activation induced by stimulation of cGAS or RIG-I, possibly due to cell type-/stimulus-specific effects or redundant functions. Similarly, other hnRNPs analyzed in our RNAi screen were dispensable for cGAS- or RIG-I-mediated ISRE activation, indicating that hnRNPM may have a unique antiviral activity within the hnRNP family. Since a limited number of interactors were evaluated in the RNAi screen, we suspect that other components of the hnRNPM-ELAVL1 multiprotein complex that promote cGAS and RIG-I signaling remain to be discovered. It has not escaped our attention that ILF2 and hnRNPH2 may represent new negative regulators of the cGAS and RIG-I signaling pathways, as evidenced by the increased ISRE reporter activity elicited by 5′ppp-dsRNA or pDNA after KD of these proteins. It remains elusive why the interactions of hnRNPM with TBK1, IKKε, and NF-κB p65 could not be captured by our AP-MS analysis. Low peptide ionization efficiencies of the prey proteins could provide a possible explanation.

Our biochemical and genetic analyses demonstrate that ELAVL1 specifically promotes the expression of type I IFNs and activation of NF-κB downstream of cGAS-STING and RIG-I and that ELAVL1 is functionally decoupled from IFNAR signaling. We demonstrate that ELAVL1 controls the phosphorylation of TBK1, IRF3, and STING after activation of cGAS or RIG-I. Functional decoupling of ELAVL1 from TLR1/2-dependent NF-κB signaling was reflected by the unaltered Pam3CSK4-induced phosphorylation of TBK1 in THP-1 ELAVL1 KO cells. Considering that ELAVL1 drives the phosphorylation of TBK1 in a PRR-dependent manner, we conclude that TBK1 molecules are recruited to distinct complexes that can be distinguished by ELAVL1. How ELAVL1 discriminates different TBK1 populations will be the subject of future research. In general, pre-formed protein complexes are beneficial for the cell to rapidly orchestrate a directed immune response to exogenous trigger, while providing improved means for spatiotemporal control due to the local accessibility of signaling proteins. Interestingly, although hnRNPM and ELAVL1 are RNA-binding proteins, co-precipitation of TBK1, IKKε, IKKβ, and NF-κB p65 with hnRNPM and co-precipitation of TBK1, IKKβ, and NF-κB p65 with ELAVL1 were RNA-independent. By contrast, interaction of hnRNPM with ELAVL1, pTBK1-Ser172, and NF-κB

pp65-Ser536 as well as interaction of ELAVL1 with hnRNPM and IKKε decreased upon RNase A treatment, suggesting that unknown RNA species may be involved in the regulation of type I IFN induction by cGAS and RIG-I. Based on the PLA data, we conclude that the heterotypic interactions of hnRNPM and ELAVL1 with pTBK1-Ser172 are predominantly localized in the cytoplasm, suggesting that the minor cytoplasmic rather than the predominating nuclear fraction of the hnRNPM-ELAVL1 complex possesses the antiviral activity.

2′3′-cGAMP is well-known to activate NF-κB signaling in human myeloid cells (Balka et al, 2020). However, we were unable to detect a 2′3′-cGAMP-mediated NF-κB reporter activation in both THP-1 WT and ELAVL1 KO cells, presumably due to technical issues such as low sensitivity of the reporter or inefficient cytosolic delivery of 2′3′-cGAMP. Furthermore, unlike the ISRE reporter, which showed activity after 20 h of stimulation with 2′3′-cGAMP, we could not detect IFNB1 mRNA expression in THP-1 cells 6 h after stimulation with 2′3′-cGAMP, further suggesting limited cytosolic availability of the stimulus.

ELAVL1 is a well-studied RNA-binding protein composed of three RRMs and known to stabilize different mRNAs by binding to adenylate-uridylate (AU)-rich elements in the 3′-UTR (Rothamel et al, 2021). Consequently, ELAVL1 has been implicated in several cellular processes ranging from development to angiogenesis and has also been linked to inflammatory diseases and cancer (Dixon et al, 2001; Nabors et al, 2001). Considering the role of ELAVL1 in transcript stabilization, an indirect effect on cGAS and RIG-I signaling by mRNA stabilization is thinkable. However, we could not detect decreased expression of the main components of the cGAS and RIG-I signaling pathways in our 3′-mRNA sequencing analysis. Additionally, the ISRE reporter activity induced by cGAS or RIG-I was also severely reduced in THP-1 cells with ELAVL1/IFNAR2 double-KO genetic background, demonstrating that ELAVL1 promotes signaling upstream of IFNAR. Further suggesting a direct role in cell signaling, ELAVL1 and hnRNPM interact with key signaling proteins such as TBK1, IKKε, IKKβ, and NF-κB p65, which function downstream of both cGAS and RIG-I. Therefore, besides a possible involvement in mRNA stabilization, our data provide evidence for a direct role of ELAVL1 and hnRNPM innate antiviral signaling.

Notably, we find that hnRNPM/ELAVL1 promote an innate immune response against both HSV-1 and SeV. A broad biological relevance of the hnRNPM complex is further highlighted by the

observation that KD of hnRNPM or ELAVL1 also impairs both cGAS and RIG-I signaling in primary human fibroblasts. Furthermore, we could show that pharmacological inhibition of ELAVL1 with the small molecule MS-444 reduces the phosphorylation of IRF3 downstream of cGAS or RIG-I in THP-1. Most importantly, MS-444 treatment also ameliorated the autoinflammatory phenotype in cells from patients diagnosed with AGS, SAVI/chilblain lupus, and dermatomyositis. Therefore, hnRNPM and ELAVL1 may represent attractive new molecular targets for the treatment of auto-inflammatory disorders resulting from the pathological activation of cGAS, STING, or RIG-I. MS-444 has already been tested as an antitumor drug in a mouse glioma model (Blanco et al, 2016). New small molecule inhibitors, such as MS-444, could not only be used for the treatment of monogenetic type I interferonopathies but also hold potential for the treatment of patients with degenerative diseases associated with cGAS-STING activation, such as Huntington disease, Parkinson disease, and amyotrophic lateral sclerosis (Decout et al, 2021; Kato et al, 2017). Ultimately, further studies are needed to gain a more detailed understanding of this newly defined protein complex in innate immunity. Nonetheless, the identification of hnRNPM and ELAVL1 as important components of cGAS-STING and RIG-I-MAVS signaling will provide more insight into the interaction of pathogens with our innate immune system, such as the previously observed interaction between SARS-CoV1 ORF3b and hnRNPM (Stukalov et al, 2021; Konno et al, 2020).

# Methods

**Reagents and tools table**

| Reagent/resource | Reference or source | Identifier or catalog number |
|---|---|---|
| **Experimental models** | | |
| HSV-1 (strain F) | Dr. Maria Hønholt Christensen | Sodeik et al, 1997 |
| Sendai virus | Prof. Hiroki Kato | Kato et al, 2005 |
| *L. monocytogenes* (strain EGD) | Prof. Martin Schlee | Hagmann et al, 2013 |
| HEK293FT | Invitrogen | R70007 |
| Primary human fibroblasts: Dermatomyositis patient (female, 60 years) WT AGS patient (male, 3 years) SAVI/chilblain lupus patient (female, 37 years) | Prof. Claudia Günther (Technische Universität Dresden, Dresden, Germany) Prof. Min Ae Lee-Kirsch (Technische Universität Dresden, Dresden, Germany) | N/A |
| THP-1 dual | InvivoGen | thpd-nfis |
| THP-1 dual cGAS KO | InvivoGen | thpd-kocGAS |
| THP-1 dual IFNAR2 KO | InvivoGen | thpd-koifnar2 |
| THP-1 dual IFNAR2 KO ELAVL1 KO (# 1) | This study | N/A |
| THP-1 dual IFNAR2 KO ELAVL1 KO (# 2) | This study | N/A |
| THP-1 dual ELAVL1 KO (# 1) | This study | N/A |

| Reagent/resource | Reference or source | Identifier or catalog number |
|---|---|---|
| THP-1 dual ELAVL1 KO (# 2) | This study | N/A |
| THP-1 dual ELAVL1 KO (# 3) | This study | N/A |
| THP-1 dual ELAVL1 KO (# 1) EF1αELAVL1-FLAG | This study | N/A |
| THP-1 dual ELAVL1 KO (# 1) EF1αGFP | This study | N/A |
| THP-1 dual RIG-I KO (# 1) | This study | N/A |
| THP-1 dual RIG-I KO (# 2) | This study | N/A |
| THP-1 dual MAVS KO (# 1) | This study | N/A |
| THP-1 dual MAVS KO (# 2) | This study | N/A |
| THP-1 dual EF1αhnRNPM-GFP | This study | N/A |
| THP-1 dual EF1αELAVL1-GFP | This study | N/A |
| THP-1 dual EF1αGFP | This study | N/A |
| **Recombinant DNA** | | |
| Lentiviral and retroviral vectors | This study | Table EV6 |
| Human ELAVL1 cDNA | Sino Biological | hg17509-g |
| Human hnRNPM cDNA | Integrated DNA Technologies (gBlocks gene fragment) | N/A |
| CVB3 3C protease (strain Nancy) | Integrated DNA Technologies (gBlocks gene fragment) | N/A |
| PV 3C protease (strain Mahoney) | Integrated DNA Technologies (gBlocks gene fragment) | N/A |
| **Antibodies** | | |
| Anti-β-actin | Li-COR Biosciences | 926-42212 |
| Anti-β-actin | Li-COR Biosciences | 926-42210 |
| Anti-β-tubulin | Cell Signaling Technology | 9F3 |
| Anti-cGAS | Cell Signaling Technology | D1D3G |
| Anti-ELAVL1 | Cell Signaling Technology | D9W7E |
| Anti-FLAG | Sigma-Aldrich | F1804 |
| Anti-GFP | Cell Signaling Technology | 4B10 |
| Anti-hnRNPM | Santa Cruz | 1D8 |
| Anti-hnRNPM M1-M4 | Abcam | ab177957 |
| Anti-IκBα | Cell Signaling Technology | L35A5 |
| Anti-IGF2BP2 | Abcam | ab124930 |
| Anti-IKKα | Cell Signaling Technology | 3G12 |
| Anti-IKKβ | Cell Signaling Technology | D30C6 |
| Anti-IKKε | Cell Signaling Technology | 2905 |
| Anti-IRF3 | Cell Signaling Technology | D6I4C |
| Anti-lamin A | Abcam | ab26300 |
| Anti-MAVS | Cell Signaling Technology | 3993 |
| Anti-NF-κB p65 | Cell Signaling Technology | D14E12 |
| Anti-phospho-IRF3 (Ser396) | Cell Signaling Technology | 4D4G |
| Anti-phospho-NF-κB p65 (Ser536) | Cell Signaling Technology | 93H1 |
| Anti-phospho-STAT1 (Tyr701) | Cell Signaling Technology | D4A7 |

| Reagent/resource | Reference or source | Identifier or catalog number |
|---|---|---|
| Anti-phospho-STING (Ser366) | Cell Signaling Technology | E9A9K |
| Anti-phospho-TBK1 (Ser172) | Cell Signaling Technology | D52C2 |
| Anti-RIG-I | Cell Signaling Technology | D14G6 |
| Anti-STAT1 | Cell Signaling Technology | 9172 |
| Anti-STING | Cell Signaling Technology | D2P2F |
| Anti-TBK1 | Cell Signaling Technology | D1B4 |
| Anti-rabbit IgG (H + L), F(ab')$_2$ Fragment (Alexa Fluor (AF) 488 Conjugate) | Cell Signaling Technology | 4412 |
| Anti-rabbit IgG-HRP | Santa Cruz | sc-2357 |
| Anti-thiophosphate ester | Abcam | ab133473 |
| IRDye 680RD goat anti-rabbit | Li-COR Biosciences | 926-68071 |
| IRDye 800CW goat anti-rabbit | Li-COR Biosciences | 926-32211 |
| IRDye 680RD goat anti-mouse | Li-COR Biosciences | 926-68070 |
| IRDye 800CW goat anti-mouse | Li-COR Biosciences | 926-32210 |
| **Oligonucleotides and other sequence-based reagents** | | |
| Synthetic DNA oligonucleotides | This study | Table EV1 |
| shRNA-encoding oligonucleotides | This study | Table EV2 |
| Stimulatory oligonucleotides | This study | Table EV3 |
| qPCR primers | This study | Table EV4 |
| gRNAs | This study | Table EV5 |
| **Chemicals, enzymes and other reagents** | | |
| 2'3'-cGAMP | InvivoGen | tlrl-nacga23-1 |
| 4-Nitrophenyl phosphate disodium salt hexahydrate (pNPP) | Carl Roth | 4165.1 |
| Alt-R S.p. Cas9 nuclease V3 | Integrated DNA Technologies | 1081058 |
| Anti-FLAG M2 magnetic beads | Sigma-Aldrich | M8823 |
| Buffer RLT | Qiagen | 79216 |
| Buffer RW1 | Qiagen | 1053394 |
| Coelenterazine native | Synchem UG & Co. KG | S053 |
| cOmplete Mini EDTA-free protease inhibitor cocktail | Roche | 4693159001 |
| DMSO | Carl Roth | A994.1 |
| GFP-Trap magnetic agarose | Chromotek | gtma-100 |
| Gibson assembly master mix | New England Biolabs | E2611 |
| Hexadimethrine bromide (Polybrene) | Sigma-Aldrich | H9268 |
| Hoechst 33342 | Invitrogen | H3570 |
| Human IFN-α2a (IFNα) | Miltenyi Biotec | 130-093-874 |

| Reagent/resource | Reference or source | Identifier or catalog number |
|---|---|---|
| IGEPAL CA-630 | Sigma-Aldrich | I8896 |
| Lipofectamine 2000 | Thermo Fisher Scientific | 11668019 |
| my-Budget 5x EvaGreen qPCR-Mix II | ROX | 80-5801000 |
| MS-444 | MedChemExpress | HY-100685 |
| Odyssey blocking buffer (TBS) | Li-COR Biosciences | 927-50000 |
| Orange G | Carl Roth | 0318.1 |
| p-Nitrobenzyl mesylate (PNBM) | Abcam | ab138910 |
| Pam3CSK4 | InvivoGen | tlrl-pms |
| Paraformaldehyde | Carl Roth | 0335.3 |
| Phorbol 12-myristate 13-acetate (PMA) | Sigma-Aldrich | P8139 |
| PhosSTOP phosphatase inhibitor cocktail | Roche | 04906837001 |
| Puromycin | InvivoGen | ant-pr-1 |
| ReadyMade random hexamers | Integrated DNA Technologies | 51-01-18-26 |
| RNA wash buffer | Zymo Research | R1003-3-24 |
| RNase A | Thermo Fisher Scientific | EN0531 |
| Thiazolyl blue tetrazolium bromide (MTT) | Sigma-Aldrich | M2128 |
| TL8-506 | InvivoGen | tlrl-tl8506 |
| Triton X-100 | Carl Roth | 3051.4 |
| **Software** | | |
| Affinity Designer | Serif Europe | N/A |
| Fiji | National Institutes of Health | N/A |
| Image Studio Lite | LI-COR Biosciences | N/A |
| Mendeley Desktop | Elsevier | N/A |
| Microsoft Office | Microsoft | N/A |
| PRISM 9 | GraphPad | N/A |
| R studio | PBC | N/A |
| **Other** | | |
| Duolink in situ red starter kit mouse/rabbit | Sigma-Aldrich | DUO92101 |
| Human CXCL10 ELISA set | BD Biosciences | 550926 |
| NE-PER™ nuclear and cytoplasmic extraction reagents | Thermo Fisher Scientific | 78835 |
| Neon™ transfection system 10 µL kit | Invitrogen | MPK1096 |
| Pierce BCA kit | Thermo Fisher Scientific | 23227 |
| QuantSeq 3' mRNA-seq library prep kit FWD | Lexogen | 015.24 |
| RevertAid first strand cDNA synthesis kit | Thermo Fisher Scientific | K1621 |

## Methods and protocols

### Ethics statement

The studies involving primary human fibroblasts were approved by the local ethics committee (Ethics committee, Medical Faculty, Technische Universität Dresden, Dresden, Germany (reference number for approval: EK203062013, EK386102017)). Written informed consent was provided by the donors and experiments conformed to the principles set out in the WMA Declaration of Helsinki and the Department of Health and Human Services Belmont Report.

### Cell culture

THP-1 cells were cultured in Roswell Park Memorial Institute (RPMI) 1640 (supplemented with 10% fetal calf serum (FCS), 100 U/ml penicillin, 100 mg/ml streptomycin) at 37 °C. HEK293FT and primary human fibroblasts were cultured in Dulbecco's modified eagle medium (DMEM) (supplemented with 10% FCS, 100 U/ml penicillin, 100 mg/ml streptomycin) at 37 °C.

### Transient expression of 3C proteases of PV and CVB3

$10^5$ HEK293FT cells were seeded in 400 µl DMEM in 24-well plates. On the following day, the medium was refreshed and the cells were transiently transfected with pLenti$^{EF1\alpha}$ ($^{EF1\alpha}$: elongation factor 1-alpha promotor) encoding hnRNPM-GFP or GFP together with pBluescript (control vector)/pRP$^{CMV}$-CVB3 3C-protease-FLAG/ pRP$^{CMV}$-PV 3C-protease-FLAG (200 ng each, $^{CMV}$: cytomegalovirus promotor) using 1.0 µl Lipofectamine 2000 (1 µg/µl) according to the manufacturer's instructions. After 48 h, GFP expression was analyzed using an Axio Vert.A1 fluorescence microscope (Zeiss).

### Generation of cleared cellular lysates

Routinely, $3.6 \times 10^5$ THP-1 cells were washed with PBS and lysed in 25 µl radioimmunoprecipitation assay buffer (RIPA) buffer (150 mM NaCl, 50 mM Tris/HCl, 1.0% (v/v) Triton X-100, 0.5% (w/v) sodium deoxycholate, 0.1% (w/v) sodium dodecyl sulfate (SDS), pH 8.0) supplemented with protease and phosphatase inhibitor cocktail on ice for 20 min. Cellular debris were removed by centrifugation and the supernatant was isolated as the cleared cellular lysate. Following, 25 µl 2× Laemmli buffer (240 mM Tris/ HCl, 8% (w/v) SDS, 40% (v/v) glycerol, 40 mM dithiothreitol (DTT), Orange G, pH 6.8) were added. Then, the samples were incubated at 95 °C for 5 min and subjected to immunoblot analysis.

### Protein quantification

Protein concentrations were determined using the Pierce bicinchoninic acid (BCA) assay kit (Thermo Fisher Scientific) according to the supplier's protocol.

### Cloning techniques

The plasmids listed in Table EV6 were cloned using Gibson assembly or traditional cloning and the primers listed in Tables EV1 and EV2.

### Cloning of pLKO.1 KD vectors

shRNA sequences were from the MISSION shRNA library (Sigma-Aldrich) or designed with the siRNA Wizard Software (InvivoGen). To clone shRNA-expressing vectors, the pLKO.1 shC001 vector (Sigma-Aldrich) was digested with *Age*I and *Eco*RI (Thermo Fisher Scientific) and ligated with the shRNA-encoding, annealed DNA oligonucleotides (Integrated DNA Technologies) (Table EV2). The ligation product was transformed into *E. coli* Stbl3 and positive colonies were identified by Sanger sequencing.

### Lentiviral expression

Lentiviral vectors with pLenti-blasticidin-EF1α and pLVX-puromycin-EF1α backbones (1.6 µg) (Table EV6) were co-transfected into HEK293FT cells with pRSV-Rev (0.4 µg), pMDLg/pRRE (1.1 µg), and pMD2.G (0.6 µg) (6-well format) using standard CaPO$_4$ precipitation. The shRNA-expressing pLKO.1 vectors (1 µg) (Table EV6) were co-transfected into HEK293FT cells with psPAX2 (0.75 µg) and pMD2.G (0.25 µg). After 72 h, the virus-containing supernatants were harvested and passed through 0.45 µm filters. The viral supernatants were spiked with 5 µg/ml polybrene (Sigma-Aldrich) and then the indicated THP-1 cells were spin-infected at 32 °C at 600×*g* for 60 min. Primary human fibroblasts were conventionally infected with polybrene-containing virus solution. After 24 h, the medium was refreshed and transduced cells were selected with 2 µg/ml puromycin (InvivoGen) for 3 days or sorted by FACS to isolate cell populations with comparable GFP expression. Polyclonal cell populations were used for subsequent experiments.

### Quantitative real-time polymerase chain reaction (qPCR)

For isolation of total RNA, cells were resuspended in 350 µl RLT buffer (Qiagen) and then 350 µl 70% (v/v) ethanol were added to the mixture. The samples were loaded onto Zymo Spin IIICG columns (Zymo Research) and washed sequentially with 350 µl buffer RW1 (Qiagen) and 350 µl Zymo RNA wash buffer (Zymo Research). The RNA was eluted in RNase-free H$_2$O. If required, samples were treated with DNase I (Thermo Fisher Scientific) according to the manufacturer's protocol to remove residual genomic DNA. Complementary DNA (cDNA) was synthesized using random hexamers (Integrated DNA Technologies) and the RevertAid first strand cDNA synthesis Kit (Thermo Fisher Scientific) according to the supplier's instructions. qPCR was performed with my-Budget 5x EvaGreen qPCR-Mix II (ROX) on a QuantStudio 5 Real-Time PCR system device (Thermo Fisher Scientific) (qPCR primers listed in Table EV4).

### Stimulatory nucleic acids

Annealed template DNA oligonucleotides (Table EV3) were used to generate 5′ppp-dsRNA by in vitro transcription as described before (Schlee et al, 2009). G$_3$-YSD was prepared by hybridizing the single-stranded DNA oligonucleotides G$_3$-YSD fwd and G$_3$-YSD rev (Table EV3). The oligonucleotides were mixed 1:1 in 1× NEBuffer 2, incubated at 95 °C for 5 min, and annealed by decreasing the temperature by 1 °C/min to 8 °C. pBluescript (pDNA) was isolated from transformed *E. coli* Stbl3 using the PureLink HiPure plasmid midiprep kit (Thermo Fisher Scientific).

### Cell stimulation

Stimulation experiments were routinely performed in 96-well plates in duplicates. Per well, $6 \times 10^4$ cells THP-1 cells or $5 \times 10^3$ primary human fibroblasts were seeded in 150 µl medium. 5′ppp-dsRNA (final concentration: 0.1 µg/ml), pDNA (0.1 µg/ml; 1.0 µg/ml), or G$_3$-YSD (0.5 µg/ml) were transfected using Lipofectamine 2000. Per well, 0.5 µl Lipofectamine 2000 (1 µg/µl) and the RNA or DNA stimuli were diluted in Opti-MEM in a total volume of 25 µl each. Following, both mixtures were combined, incubated for 5 min at room temperature,

and added to the cells. 2′3′-cGAMP (10 µg/ml), TL8-506 (1.0 µg/ml), Pam3CSK4 (0.5 µg/ml), and IFNα (1000 U/ml; 5000 U/ml) were diluted to the desired concentrations in 50 µl Opti-MEM and directly added to the cells. In the non-stimulated condition (ctrl), 50 µl Opti-MEM were added. After the indicated times, the supernatants were harvested and used for downstream assays. If assays required larger formats, the experimental parameters were adjusted proportionally, considering the surface area of the well.

For viral infections, $3 \times 10^5$ THP-1 cells were seeded in 300 µl RPMI and infected with HSV-1 or SeV at the indicated multiplicity of infection (MOI). After 24 h, activation of the ISRE reporter was determined. For infection with *L. monocytogenes*, $3 \times 10^5$ THP-1 cells were seeded in antibiotics-free medium and incubated with *Listeria* at MOI 1 for 2 h. Following, the cells were washed with medium devoid of antibiotics and resuspended in medium containing 50 µg/ml gentamicin. After 24 h, activation of the ISRE reporter was determined.

### Detection of cytokines
Secretion of CXCL10, IL-6, and TNF-α was analyzed with commercial ELISA assay kits (BD Biosciences) according to the manufacturer's instructions.

### Thiazolyl blue tetrazolium bromide (MTT) assay
Metabolic activity was determined by MTT assay and used as a surrogate parameter for cellular viability. $6 \times 10^4$ cells were cultured in 100 µl medium supplemented with the yellow tetrazole substrate compound MTT (1 mg/ml) for 1 h. The reaction was stopped by adding 100 µl SDS solution (10% (w/v)) and the absorbance at 595 nm was measured after dissolution of formazan crystals.

### Quantification of ISRE and NF-κB reporter activation in THP-1 dual cells
THP-1 dual cells stably express a Gaussia luciferase gene under the control of an ISG54 minimal promotor fused to five ISREs, and a secreted alkaline phosphatase (SEAP) gene under the control of an IFNβ minimal promotor in conjunction with five NF-κB consensus transcriptional response elements and three c-Rel-binding sites. To determine ISRE reporter gene activity, 30 µl cell culture supernatant and 30 µl coelenterazine solution (1 µg/ml in $H_2O$) were mixed and luminescence was measured using an EnVision 2104 Multilabel Reader Device (PerkinElmer). For the NF-κB reporter, 40 µl cell culture supernatant were incubated with 40 µl pNPP substrate buffer (100 mM NaCl, 100 mM Tris/HCl, 5 mM $MgCl_2$, 10 mg/ml pNPP, pH 9.5) for 30–60 min and reporter gene activation was determined by measuring the absorbance at 405 nm.

### Generation of nuclear and cytoplasmic extracts
In total, $1.86 \times 10^6$ cells were lipofected with 0.5 µg/ml $G_3$-YSD or 0.1 µg/ml 5′ppp-dsRNA, or left non-stimulated. After 3 h, cells were harvested, washed with PBS, and the NE-PER nuclear and cytoplasmic extraction kit (Thermo Fisher Scientific) was used to isolate nuclear and cytoplasmic fractions according to the supplier's protocol. Protein concentrations were determined by BCA assay (Thermo Fisher Scientific) and 5–10 µg total protein were analyzed by immunoblotting.

### Immunoblotting
Samples were separated by discontinuous, denaturing SDS-polyacrylamide gel electrophoresis (PAGE) using 3% SDS-PAGE stacking gels and 10% SDS-PAGE resolving gels. After separation, proteins were transferred to 0.45 µm nitrocellulose membranes. If multiple targets were analyzed, equal amounts of sample were resolved on different SDS-PAGE gels and membranes were probed sequentially. Ponceaus S staining was used to examine the success of the protein transfer. Membranes were destained with TBS containing 0.1% (v/v) Tween 20, blocked with 5% bovine serum albumin (BSA)-PBS containing 0.1% (v/v) Tween 20 for 1 h, and incubated with the indicated primary (1:500–1:1000) and secondary antibodies (1:12,500). Protein signals were recorded on an Odyssey Fc near-infrared imaging system device (LI-COR Biosciences) and signal intensities were analyzed using Image Studio Lite software (LI-COR Biosciences).

### AP-MS analysis
THP-1 cells expressing hnRNPM-GFP, ELAVL1-GFP, or GFP were lysed in TAP lysis buffer (100 mM NaCl, 50 mM Tris/HCl, 1.5 mM $MgCl_2$, 5% (v/v) glycerol, 0.5% (v/v) IGEPAL CA-630, pH 7.5) supplemented with protease and phosphatase inhibitor cocktail on ice for 20 min, followed by a short sonication burst (15 s, amplitude 90%, cycle 1, VialTweeter Sonication Device) (quadruplicates of each cell line). 25 µl equilibrated GFP-Trap magnetic agarose beads (Chromotek) were added to the clarified lysates (1.5 mg total protein per replicate) and incubated under rotation at 4 °C overnight. The beads were washed with TAP lysis buffer (3×, 700 µl) and TAP wash buffer (100 mM NaCl, 50 mM Tris/HCl, 1.5 mM $MgCl_2$, 5% (v/v) glycerol, pH 7.5) (3×, 700 µl) and stored at −80 °C for further processing. Following, enriched proteins were denatured with 40 µl U/A buffer (8 M urea, 100 mM Tris/HCl, pH 8.5), reduced with 10 mM DTT (30 min, 25 °C), alkylated with 5.5 mM IAA (20 min, 25 °C, in the dark) and digested by subsequent addition of 0.5 µg LysC (3 h, 25 °C; WAKO Chemicals) and 0.5 µg Trypsin (18 h, 25 °C; Promega) in ABC buffer (50 mM $NH_4HCO_3$, pH 8.0). Peptide purification on StageTips with three layers of C18 Empore filter discs (3 M) and subsequent mass spectrometry analysis was performed as described previously (Hubel et al, 2019; Rappsilber et al, 2007). Briefly, purified peptides were loaded on a 50 cm reverse-phase analytical column (75 µm diameter, 60 °C; ReproSil-Pur C18-AQ 1.9 µm resin; Dr. Maisch) and separated using an EASY-nLC 1200 system (Thermo Fisher Scientific). For peptide separation, a 120 min gradient with a flow rate of 300 nl/min and a binary buffer system consisting of buffer A (0.1% formic acid in $H_2O$) and buffer B (80% acetonitrile, 0.1% formic acid in $H_2O$) was used: 5–30% buffer B (95 min), 30–95% buffer B (10 min), wash out at 95% buffer B (5 min), decreased to 5% buffer B (5 min), and kept at 5% buffer B (5 min). Eluting peptides were directly analyzed on a Q-Exactive HF mass spectrometer equipped with a nano-electrospray source (Thermo Fisher Scientific). Spray voltage was set to 2.4 kV, funnel RF level at 60, and heated capillary at 250 °C. Data-dependent acquisition included repeating cycles of one MS1 full scan (300–1650 $m/z$, R = 60,000 at 200 $m/z$) at an ion target of $3 \times 10^6$ with an injection time of 20 ms, followed by 15 MS2 scans of the highest abundant isolated and higher-energy collisional dissociation (HCD) fragmented peptide precursors (R = 15,000 at 200 $m/z$). For MS2 scans, collection of isolated peptide precursors was limited by an ion target of $1 \times 10^5$ and a maximum injection time of 25 ms. Isolation and fragmentation of the same peptide precursor was eliminated by dynamic exclusion for 20 s. The isolation window of the quadrupole

was set to 1.4 $m/z$ and HCD was set to a normalized collision energy of 27%. RAW files were processed with MaxQuant (version 1.6.17.0) using the standard settings and label-free quantification (iBAQ; LFQ, LFQ min ratio count 1, normalization type none) enabled. Spectra were searched against forward and reverse sequences of the reviewed human proteome including isoforms (UniprotKB, release 08.2020) and GFP by the built-in Andromeda search engine (Tyanova et al, 2016a).

### Statistical analysis of AP-MS data

The output of MaxQuant was analyzed with Perseus (version 1.6.14.0), R (version 4.0.2), and RStudio (version 1.3.1073) (Tyanova et al, 2016b). Detected protein groups identified as known contaminants, reverse sequence matches, or only identified by site were excluded from the analysis. Following $\log_2$ transformation, the iBAQ intensity of each protein group in a given sample was normalized to correct for technical variation by subtracting a sample-specific normalization factor ($NF_j$) based on the median iBAQ intensity of all protein groups per sample ($median_j$):

$$median_j = median_{i=1,2,\ldots N}\left(\log_2 iBAQ\ intensity_{i,j}\right)$$

$$NF_j = median_j - median_{j=1,2,\ldots M}\left(median_j\right)$$

$$N = number\ of\ protein\ groups\ per\ sample$$

$$M = number\ of\ samples$$

Following normalization, proteins without quantification in at least three replicates of one condition were removed and missing values were imputed for each replicate individually by sampling values from a normal distribution calculated from the original data distribution (width = 0.3 × s.d., downshift = −1.8 × s.d.). Differentially enriched protein groups between the conditions were identified via two-sided Welch's t tests (S0 = 0.1) corrected for multiple hypothesis testing applying a permutation-based FDR (FDR < 0.05, 250 randomizations). Protein groups that were not quantified in at least three replicates in the ELAVL1 or hnRNPM condition when compared to GFP or when comparing ELAVL1 to hnRNPM were removed for statistical testing. STRING-based analysis of hnRNPM or hnRNPM-ELAVL1 shared interactors was performed in Cytoscape (version 3.8.2) using the stringApp (version 1.6.0) in combination with a confidence cutoff of 0.2 for considering functional connections between interactors, an MCL inflation parameter of 3 for clustering and a Benjamini–Hochberg-adjusted FDR of <0.05 to select significantly enriched pathway annotations.

### Co-immunoprecipitation (co-IP)

Prior to IP, THP-1 cells were lentivirally transduced to express hnRNPM-GFP or GFP and cell populations with comparable GFP expression were isolated by FACS. $7.5 \times 10^6$ cells were seeded in 7.5 ml RPMI and lipofected with 0.1 µg/ml pDNA to activate cGAS or treated with Opti-MEM. After incubation for 3 h, the cells were washed with PBS and lysed in 700 µl TAP lysis buffer supplemented with protease and phosphatase inhibitor cocktail for 20 min on ice. Using a short sonication burst, the nuclei were disrupted and the lysate was cleared from debris by centrifugation. In all, 700–800 µg of total protein were incubated with 12.5 µl equilibrated GFP-Trap magnetic agarose beads (Chromotek) on a rotating wheel at 4 °C

overnight. If RNase A treatment was performed, beads were washed with TAP wash buffer (3×, 700 µl), resuspended in 300 µl TAP wash buffer supplemented with protease inhibitor, phosphatase inhibitor, and 100 µg/ml RNase A (Thermo Fisher Scientific), and incubated under rotation at 4 °C for 90 min. Following, the beads were washed sequentially with TAP lysis buffer (3×, 700 µl) and TAP wash buffer (3×, 700 µl) and bound proteins were eluted by incubating the beads for 10 min at 95 °C with 2× Laemmli buffer. Subsequently, input loading controls (5% of total protein used for IP) and eluates were subjected to immunoblot analysis.

To immunoprecipitate ELAVL1, $2.5 \times 10^6$ HEK293FT cells were seeded per 10-cm plate. On the next day, the medium was replaced and the cells were transfected with 20 µg pLVX$^{EF1\alpha}$-ELAVL1-FLAG using lipofectamine 2000 or left untreated. After 72 h, cleared cellular lysates were generated as described above. Per IP, 2 mg total protein was incubated with 50 µl equilibrated anti-FLAG M2 magnetic beads on a rotating wheel at 4 °C overnight. Unspecific protein binding to the beads incubated with lysate of WT cells was quenched with 4 ng/µl 3× FLAG peptide. Following, the beads were washed with TAP wash buffer (3×, 700 µl), resuspended in 300 µl TAP wash buffer supplemented with protease inhibitor, phosphatase inhibitor and, if indicated, 100 µg/ml RNase A. The samples were incubated under rotation at 4 °C for 90 min and then the beads were washed sequentially with TAP lysis buffer (3×, 700 µl) and TAP wash buffer (3×, 700 µl). Bound proteins were eluted by incubating the beads for 30 min at 4 °C with 100 µl TAP wash buffer supplemented with 150 ng/µl 3× FLAG peptide. After magnetic separation, the eluates were incubated with Laemmli buffer at 95 °C for 5 min and subjected to immunoblot analysis. 5% of the total protein used for IP were analyzed as input loading controls.

### CRISPR-Cas9-mediated KO cell line generation

The Alt-R CRISPR-Cas9 system (Integrated DNA Technologies) was used according to the supplier's instructions to disrupt the genes encoding for ELAVL1, RIG-I, and MAVS in THP-1 cells. gRNAs (Table EV5) were designed using the Alt-R HDR design tool (Integrated DNA Technologies). Briefly, hybrids of CRISPR RNA (crRNA) and trans-activating crRNA (tracrRNA) were formed by mixing 0.5 µl crRNA (200 µM), 0.5 µl tracrRNA (200 µM), and 1.28 µl IDTE buffer, followed by incubating the mixture for 5 min at 95 °C and gradual cooling to room temperature. Afterwards, 1.25 µl crRNA:tracrRNA hybrid was mixed with 1.25 µl diluted Cas9, which was prepared by mixing 0.5 µl resuspension buffer R (Neon transfection system 10 µl kit, Thermo Fisher Scientific) with 0.75 µl Cas9 enzyme, and incubated at room temperature for 20 min. $1.25 \times 10^6$ THP-1 cells were washed with PBS, resuspended in 22.5 µl resuspension buffer R. 18 µl cell suspension were mixed with 4 µl electroporation enhancer (10.8 µM) (Integrated DNA Technologies) and 2 µl crRNA:tracrRNA:Cas9 RNP complex. Electroporations were performed on a Neon electroporation system device (Thermo Fisher Scientific) at 1600 V with 3 pulses and 10 ms pulse width using a 10 µl Neon tip. Cells were cloned by limiting dilution and target KOs were verified by immunoblotting and/or Sanger sequencing.

### 3'-mRNA sequencing

THP-1 WT and ELAVL1 KO (clone #1) cells were stimulated with 1000 U/ml IFNα or left non-stimulated ($3.6 \times 10^5$ cells, triplicates).

After 6 h, the cells were harvested and total RNA was purified as described above. Total RNA was used to generate RNA sequencing libraries using the QuantSeq 3′ mRNA-seq library prep kit FWD (Lexogen). The samples were sequenced on a HiSeq 1500 device (Illumina). For data analysis, the reads were aligned to the human reference genome (Ensembl genome version 96) using STAR aligner, quantified with HTSeq, and differential expression analysis was performed using DESeq2 (Anders et al, 2015; Love et al, 2014; Dobin et al, 2013).

### PLA

The Duolink in situ red kit rabbit/mouse (Sigma-Aldrich) was used according to supplier's instructions to analyze interactions between hnRNPM, ELAVL1, and pTBK1-Ser172 by PLA. Briefly, hnRNPM-GFP-, ELAVL1-GFP-, or GFP-expressing THP-1 cells were differentiated with phorbol-12-myristat-13-acetat (PMA) in CellCarrier-96 ultra imaging microplates (PerkinElmer), stimulated with Alexa Fluor 647 (AF647)-labelled $G_3$-YSD, fixed with 4% (w/v) paraformaldehyde in PBS (pH 6.9) at room temperature for 10 min, and permeabilized with 0.3% (v/v) Triton X-100-PBS at room temperature for 10 min. Following, the samples were washed twice with PBS and then blocked with 40 µl Duolink blocking solution at 37 °C for 60 min and incubated with primary antibodies against the two target proteins on ice for 2 h. The primary antibodies were diluted in Duolink antibody diluent as follows: anti-GFP (1:200, mouse monoclonal antibody (mAb)), anti-phospho-TBK1 (Ser172) (1:100, rabbit mAb), anti-ELAVL1 (1:100, rabbit mAb). The specimens were washed with Duolink wash buffer A (4× 150 µl, 5 min each wash) and subsequently incubated with 40 µl Duolink PLA probes solution, containing secondary antibodies conjugated with PLA probes plus or minus. After 1 h at 37 °C, the samples were washed with Duolink wash buffer A (3× 150 µl, 5 min each wash) and incubated with 40 µl Duolink ligase solution at 37 °C for 30 min. Subsequently, the cells were washed with Duolink wash buffer A (3× 150 µl, 5 min each wash), incubated with 40 µl Duolink polymerase solution at 37 °C for 100 min, followed by washing with Duolink wash buffer B (3× 150 µl, 10 min each wash) and counter-staining of the nuclei with Hoechst 33342-PBS (5 µg/ml) for 10 min. After washing with PBS (3× 150 µl, 5 min each wash), the samples were stored in PBS at 4 °C and imaged on a Leica SP8 confocal microscope (HC PL APO CS2 63x/12.0 water immersion objective) by recording z stacks (0.5 µm z step size) in line sequential scan mode with zoom factor 4 and line average 4. PLA signals were quantified using Fiji. To reduce background signals, a threshold was set (Fiji default threshold: 6–255 or 10–255). Particles (5–∞ pixel units or 10–∞ pixel units) detected in individual cells were counted and the total number of particles for all z stack images was defined as the PLA score, providing a proportional measure to the number of protein interactions. Laser intensities, threshold settings, and the range of the particle size were fixed within one experimental group. Images are shown as maximum intensity projections.

### Experiment study design and statistics

Randomization was not applied in the experiments, as this study is not a randomized control trial, and randomization is typically not done in in vitro studies comparable to this study. All experimental groups were handled under the same protocols and conditions. Samples were not blinded in this study. Data were only excluded from analysis if they were outliers, typically due to experimental errors or hardware malfunction.

GraphPad Prism 9 was used for data analysis and to determine statistical significances. No statistical methods were applied to determine sample sizes and no formal sample size calculations were conducted. Statistical tests and parameters are reported in the figure legends. Exact $P$ values are displayed in the figures and error bars represent the standard deviation (SD). One-way analysis of variance (ANOVA) was used to compare the means of a single independent variable across two or more independent groups. Two-way ANOVA was used to assess the effects of two independent variables, as well as their interaction, on the means across two or more independent groups. When significant main effects or interactions were found in the two-way ANOVA, post-hoc multiple comparison tests were conducted to determine which specific group means differed from each other.

## Data availability

The mass spectrometry proteomics data have been deposited to the ProteomeXchange Consortium via the PRIDE partner repository with the dataset identifier PXD028160 (Perez-Riverol et al, 2019). The 3′-mRNA sequencing data have been deposited to the GEO data repository with the dataset identifier GSE184273. Requests for further information, code, and resources and reagents should be directed to and will be fulfilled by the Lead Contact, Martin Schlee (martin.schlee@uni-bonn.de).

The source data of this paper are collected in the following database record: biostudies:S-SCDT-10_1038-S44318-024-00331-x.

## Peer review information

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

## Acknowledgements

We thank Saskia Schmitz and Laura Mlitzko for technical assistance. Preparatory experiments were performed by the Core Facility Mass Spectrometry, Institute of Biochemistry and Molecular Biology, Medical Faculty, University of Bonn with a mass spectrometer that was funded by the Deutsche Forschungsgemeinschaft (DFG, German Research Foundation)—Projektnummer 174793735. We would like to thank the Microscopy Core Facility of the Medical Faculty at the University of Bonn for providing help, services, and devices funded by the DFG—Projektnummer 388159768. We would like to acknowledge the assistance of the Flow Cytometry Core Facility at the Institute of Experimental Immunology, Medical Faculty at the University of Bonn—Projektnummer 216372545. We would like to acknowledge the assistance of the Next Generation Sequencing Core Facility at the Institute of Human Genetics, Medical Faculty at the University of Bonn. This study was funded by the DFG under Germany's Excellence Strategy—EXC2151—390873048 of which EB, GH, MG, HK, FIS, and MS are members. It was also supported by other grants of the DFG: TRR237 369799452 (EB, GH, MS, MG, BMK, CG, HK, FIS, MAL-K, and A Pichlmair), TRR259 397484323 (EB and GH), SFB670 (EB, GH, and MS), SFB704 (GH), Bonn and Melbourne Research and Graduate School GRK2168 272482170 (EB and MS), Emmy Noether Programme 322568668 (FIS), research grant SCHL1930/1-2 (MS), CRC237 369799452/A11, CRC237 369799452/B21 and CRC369 501752319/C06 (M.L.-K.). Further support was from the Federal Ministry of Education and Research (BMBF) grants 01GM2206C (GAIN, M.L.-K.) and 01GL2405H (DZKJ, M.L.-K.). This work is part of the PhD thesis of AK at the University of Bonn.

## Author contributions

**Alexander Kirchhoff**: Conceptualization; Data curation; Formal analysis; Supervision; Investigation; Visualization; Methodology; Writing—original draft; Writing—review and editing. **Anna-Maria Herzner**: Conceptualization; Investigation; Writing—review and editing. **Christian Urban**: Data curation; Formal analysis; Investigation; Visualization; Writing—review and editing. **Antonio Piras**: Formal analysis; Methodology; Writing—review and editing. **Robert Düster**: Investigation; Writing—review and editing. **Julia Mahlberg**: Investigation; Writing—review and editing. **Agathe Grünewald**: Investigation; Writing—review and editing. **Thais M Schlee-Guimarães**: Data curation; Software; Formal analysis; Visualization. **Katrin Ciupka**: Investigation; Writing—review and editing. **Petro Leka**: Investigation; Writing—review and editing. **Robert J Bootz**: Investigation; Writing—review and editing. **Christina Wallerath**: Investigation; Writing—review and editing. **Charlotte Hunkler**: Investigation; Writing—review and editing. **Ann Kristin De Regt**: Investigation; Writing—review and editing. **Beate M Kümmerer**: Resources; Funding acquisition; Writing—review and editing. **Maria Hønholt Christensen**: Resources; Writing—review and editing. **Florian I Schmidt**: Resources; Funding acquisition; Writing—review and editing. **Min Ae Lee-Kirsch**: Resources; Funding acquisition; Writing—review and editing. **Claudia Günther**: Resources; Funding acquisition; Writing—review and editing. **Hiroki Kato**: Resources; Funding acquisition; Writing—review and editing. **Eva Bartok**: Resources; Funding acquisition; Writing—original draft; Writing—review and editing. **Gunther Hartmann**: Resources; Funding acquisition; Writing—review and editing. **Matthias Geyer**: Resources; Supervision; Funding acquisition; Methodology; Writing—review and editing. **Andreas Pichlmair**: Conceptualization; Resources; Supervision; Funding acquisition; Methodology; Writing—review and editing. **Martin Schlee**: Conceptualization; Resources; Formal analysis; Supervision; Funding acquisition; Methodology; Writing—original draft; Writing—review and editing.

Source data underlying figure panels in this paper may have individual authorship assigned. Where available, figure panel/source data authorship is

listed in the following database record: biostudies:S-SCDT-10_1038-S44318-024-00331-x.

## Funding

## Disclosure and competing interests statement

The authors declare no competing interests.

# Expanded View Figures

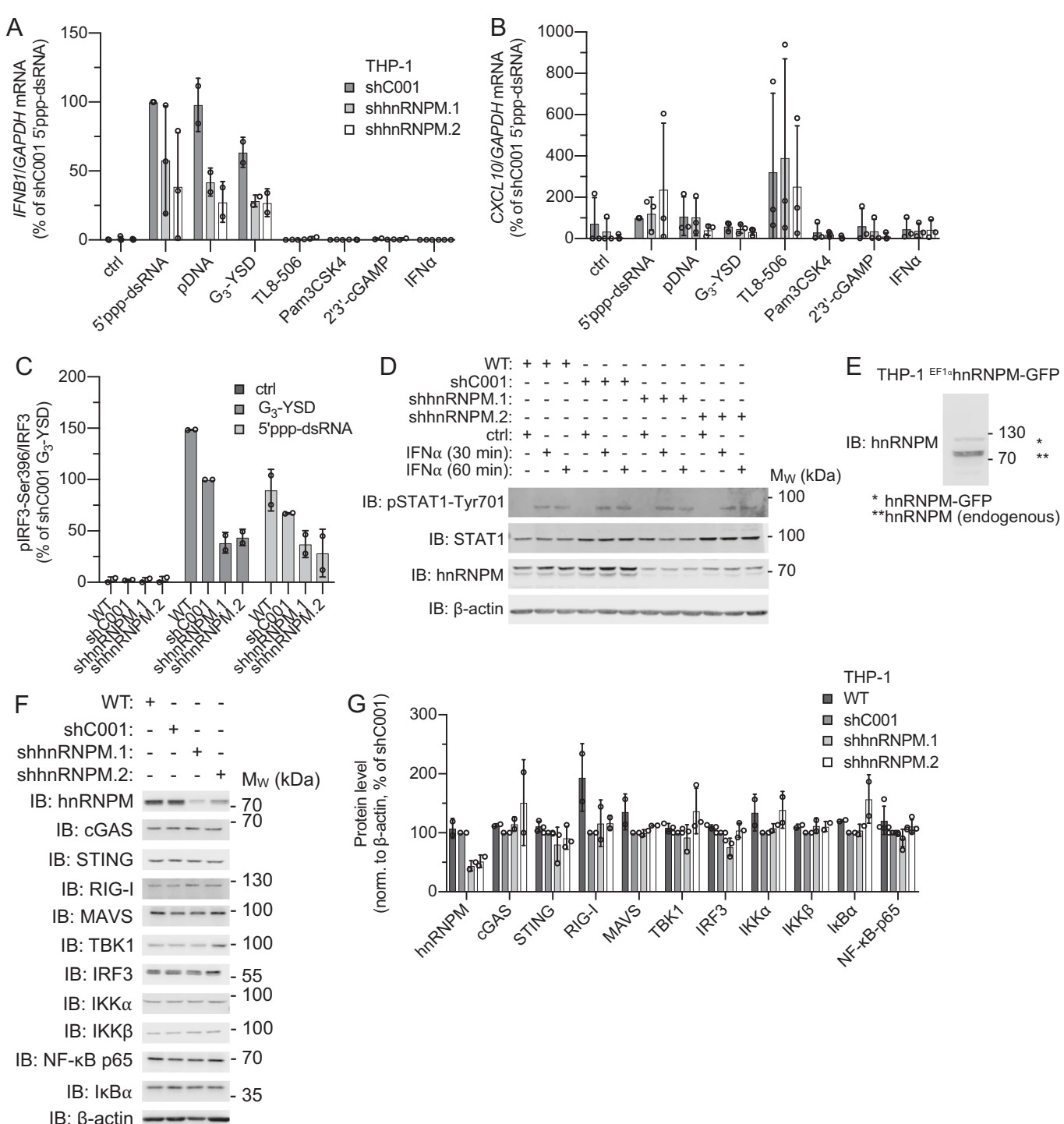

◀ **Figure EV1. hnRNPM functions upstream of IFNAR.**

(A) Expression of *IFNB1* mRNA in THP-1 cells expressing control shRNA (shC001) or hnRNPM-specific shRNAs (shhnRNPM.1, shhnRNPM.2) 6 h after stimulation of RIG-I with 5'ppp-dsRNA (0.1 μg/ml), of cGAS with pDNA (0.1 μg/ml) or $G_3$-YSD (0.5 μg/ml), of TLR8 with TL8-506 (1.0 μg/ml), of TLR1/2 with Pam3CSK4 (0.5 μg/ml), of STING with 2'3'-cGAMP (10 μg/ml), or of IFNAR with IFNα (1000 U/ml). ctrl, non-stimulated (mean ± SD; stimuli from left to right: $n = 3, 3, 2, 2, 2, 2, 2, 2$ independent experiments). (B) Expression of *CXCL10* mRNA in the cells depicted in (A) 6 h after stimulation with the indicated stimuli (mean ± SD; stimuli from left to right: $n = 3, 3, 3, 3, 3, 3, 3, 3$ independent experiments). (C) Quantification of pIRF3-Ser396 levels shown in Fig. 1I (mean ± SD; stimuli from left to right: $n = 2, 2, 2$ independent experiments). (D) Immunoblot analysis of pSTAT1-Tyr701 induction in THP-1 WT and cells expressing control shRNA (shC001) or hnRNPM-specific shRNAs (shhnRNPM.1, shhnRNPM.2) after stimulation with IFNα (1000 U/ml). ctrl, non-stimulated. One representative experiment of two independent experiments is shown. (E) Immunoblot analysis of THP-1 cells expressing hnRNPM-GFP using an hnRNPM-specific antibody. (F) Immunoblot analysis of THP-1 WT and cells expressing control shRNA (shC001) or hnRNPM-specific shRNAs (shhnRNPM.1, shhnRNPM.2) using the indicated antibodies. One representative experiment of two independent experiments is shown. (G) Quantification of immunoblot data shown in Fig. EV1F (mean ± SD; proteins from left to right: $n = 2, 2, 3, 2, 2, 3, 3, 2, 2, 4$ independent experiments).

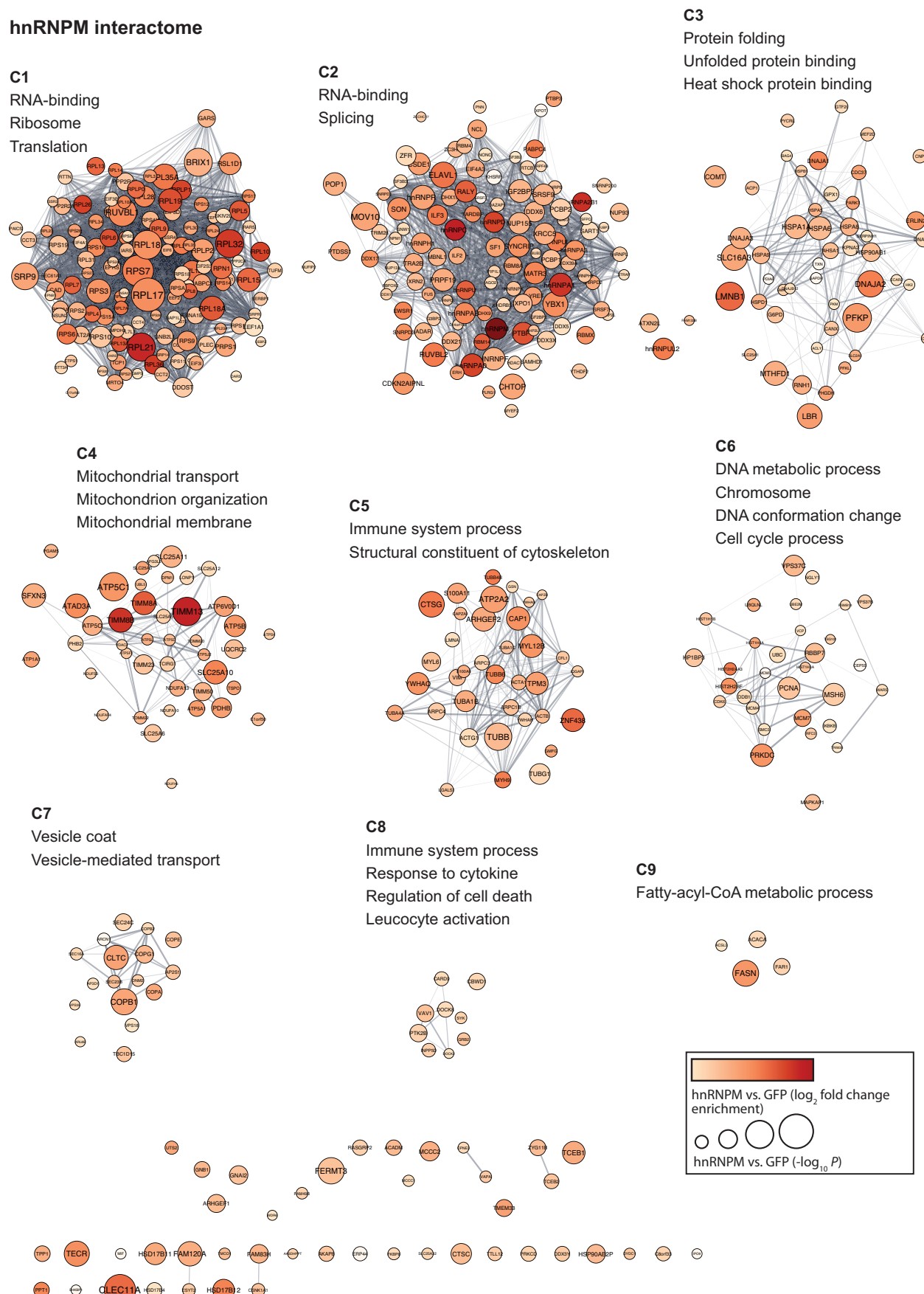

**hnRNPM interactome**

**C1**
RNA-binding
Ribosome
Translation

**C2**
RNA-binding
Splicing

**C3**
Protein folding
Unfolded protein binding
Heat shock protein binding

**C4**
Mitochondrial transport
Mitochondrion organization
Mitochondrial membrane

**C5**
Immune system process
Structural constituent of cytoskeleton

**C6**
DNA metabolic process
Chromosome
DNA conformation change
Cell cycle process

**C7**
Vesicle coat
Vesicle-mediated transport

**C8**
Immune system process
Response to cytokine
Regulation of cell death
Leucocyte activation

**C9**
Fatty-acyl-CoA metabolic process

hnRNPM vs. GFP (log$_2$ fold change enrichment)

hnRNPM vs. GFP (-log$_{10}$ $P$)

**Figure EV2.   hnRNPM interacts with proteins connected to immune system processes.**

hnRNPM-GFP and GFP were immunoprecipitated from lysates of non-stimulated THP-1 cells. Differential interactors of hnRNPM were analyzed by STRING enrichment and annotated with GO terms enriched among hnRNPM interactors. hnRNPM interactors not annotated with these GO terms are shown at the bottom. cluster (C) 1–9. The statistical tests used are described in detail in the methods.

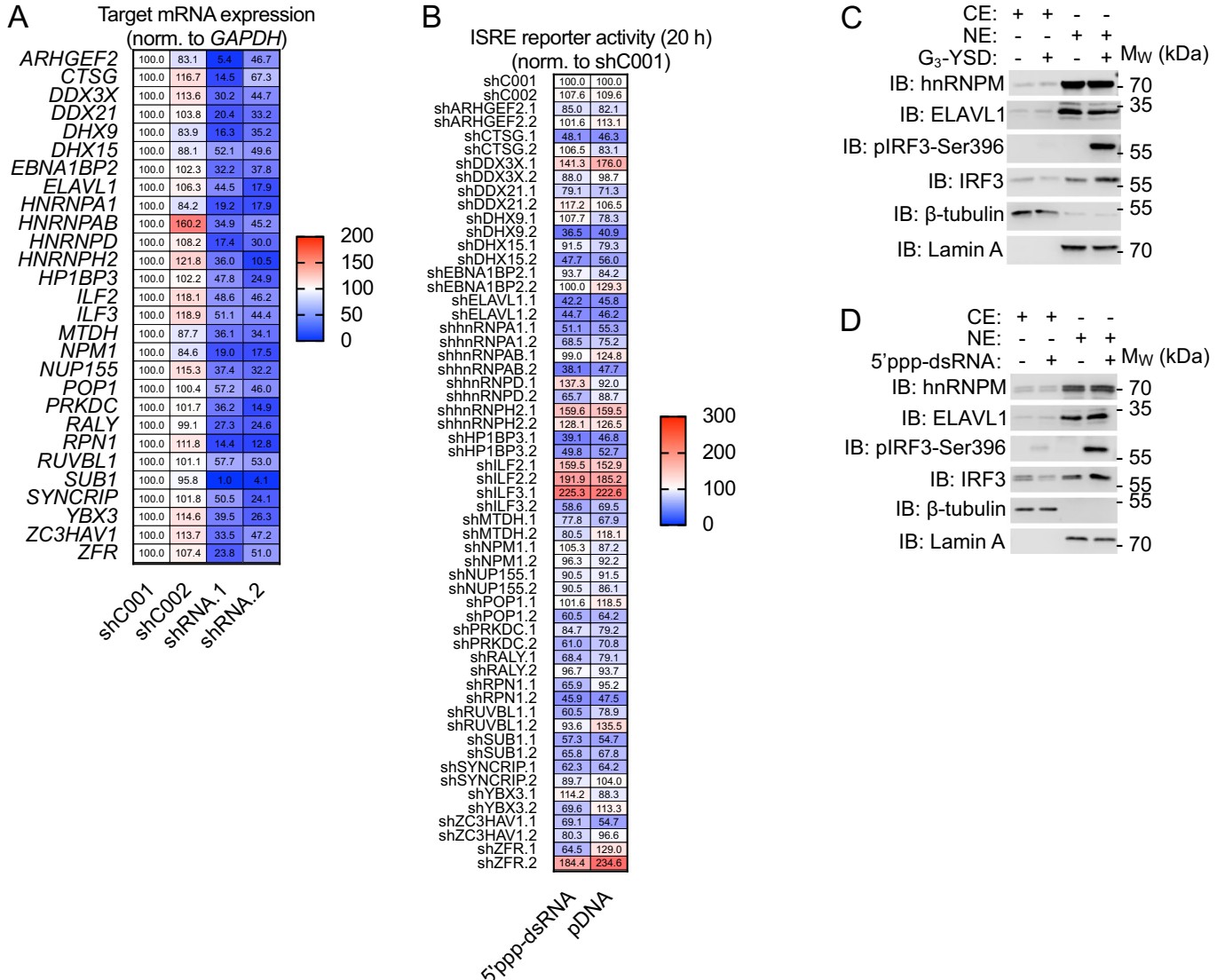

**Figure EV3. Functional RNAi screen of hnRNPM interactors.**

(A) mRNA expression of the indicated targets (y-axis) in THP-1 cells expressing control shRNAs (shC001, shC002) or target-specific shRNAs (shRNA.1, shRNA.2). Target mRNA expression was normalized to *GAPDH* mRNA and then normalized to shC001-expressing cells. Data of two or more independent experiments are shown. (B) ISRE reporter activation in the cells depicted in (A) 20 h after stimulation with 5'ppp-dsRNA (0.1 μg/ml) or pDNA (0.1 μg/ml). Luciferase signals were normalized to shC001-expressing cells of the respective condition. Data of two or more independent experiments are shown. (C) Nuclear extracts (NE) and cytoplasmic extracts (CE) were prepared from non-stimulated THP-1 cells or from cells stimulated with $G_3$-YSD (0.5 μg/ml) for 3 h and analyzed by immunoblotting with the indicated antibodies. One representative experiment of two independent experiments is shown. (D) NE and CE were prepared from non-stimulated THP-1 cells or from cells stimulated with 5'ppp-dsRNA (0.1 μg/ml) for 3 h and analyzed by immunoblotting with the indicated antibodies. One representative experiment of two independent experiments is shown.

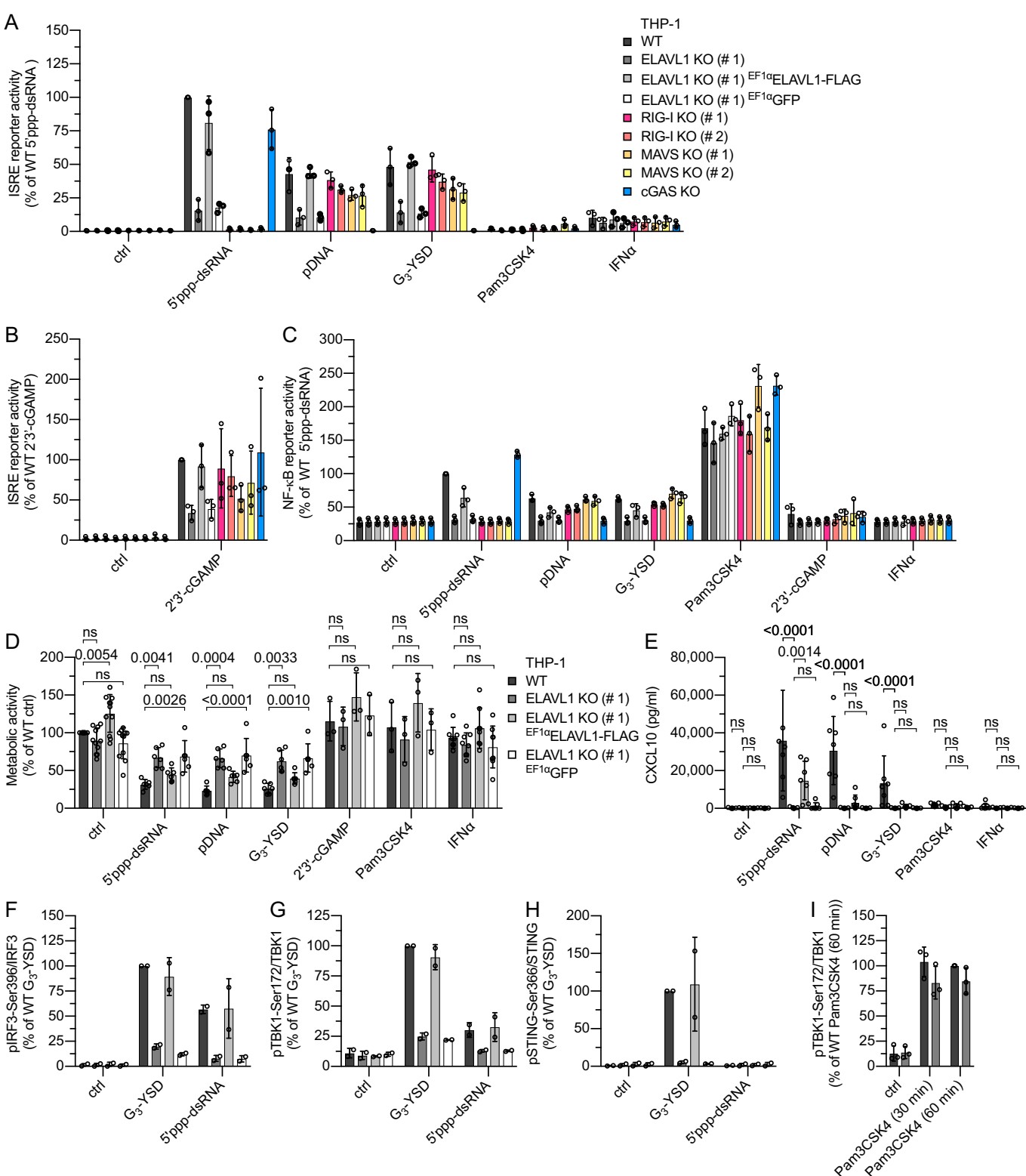

◀ **Figure EV4.** **ELAVL1 is a potent positive regulator of cGAS and RIG-I signaling.**

(A) ISRE reporter activation in THP-1 WT, ELAVL1 KO (#1), ELAVL1 KO (#1) expressing ELAVL1-FLAG or GFP, RIG-I KO (clones #1–2), MAVS KO (clones #1–2), and cGAS KO cells 20 h after stimulation with 5′ppp-dsRNA (0.1 µg/ml), pDNA (0.1 µg/ml), $G_3$-YSD (0.5 µg/ml), Pam3CSK4 (0.5 µg/ml), or IFNα (5000 U/ml) (mean ± SD; stimuli from left to right: $n = 3, 3, 3, 3, 3$ independent experiments). ctrl, non-stimulated. (B) ISRE reporter activation in the cells depicted in (A) 20 h after challenge with 2′3′-cGAMP (10 µg/ml) (mean ± SD; stimuli from left to right: $n = 3, 3$ independent experiments). ctrl, non-stimulated. (C) NF-κB reporter activation in the cells depicted in (A) 20 h after challenge with the indicated stimuli (mean ± SD; stimuli from left to right: $n = 3, 3, 3, 3, 3, 3$ independent experiments). (D) MTT assay of THP-1 WT, ELAVL1 KO (#1), and ELAVL1 KO (#1) expressing ELAVL1-FLAG or GFP 20 h after stimulation with 5′ppp-dsRNA (0.1 µg/ml), pDNA (0.1 µg/ml), $G_3$-YSD (0.5 µg/ml), 2′3′-cGAMP (10 µg/ml), Pam3CSK4 (0.5 µg/ml), or IFNα (1000 U/ml). ctrl, non-stimulated (stimuli from left to right: $n = 11, 6, 6, 6, 3, 3, 8$ independent experiments). (E) CXCL10 ELISA with supernatants of the cells depicted in (D) collected 20 h after stimulation with 5′ppp-dsRNA (0.1 µg/ml), pDNA (0.1 µg/ml), $G_3$-YSD (0.5 µg/ml), Pam3CSK4 (0.5 µg/ml), or IFNα (1000 U/ml). ctrl, non-stimulated (stimuli from left to right: $n = 7, 7, 7, 7, 7, 7$ independent experiments). (F) Quantification of pIRF3-Ser396 levels shown in Fig. 4J (mean ± SD; stimuli from left to right: $n = 2, 2, 2$ independent experiments). (G) Quantification of pTBK1-Ser172 levels shown in Fig. 4J (mean ± SD; stimuli from left to right: $n = 2, 2, 2$ independent experiments). (H) Quantification of pSTING-Ser366 levels shown in Fig. 4J (mean ± SD; stimuli from left to right: $n = 2, 2, 2$ independent experiments). (I) Quantification of pTBK1-Ser172 levels shown in Fig. 4K (mean ± SD; stimuli from left to right: $n = 3, 3, 3$ independent experiments). For (D, E): mean ± SD, two-way ANOVA, Dunnett's multiple comparisons test. ns, $P$ value > 0.05.

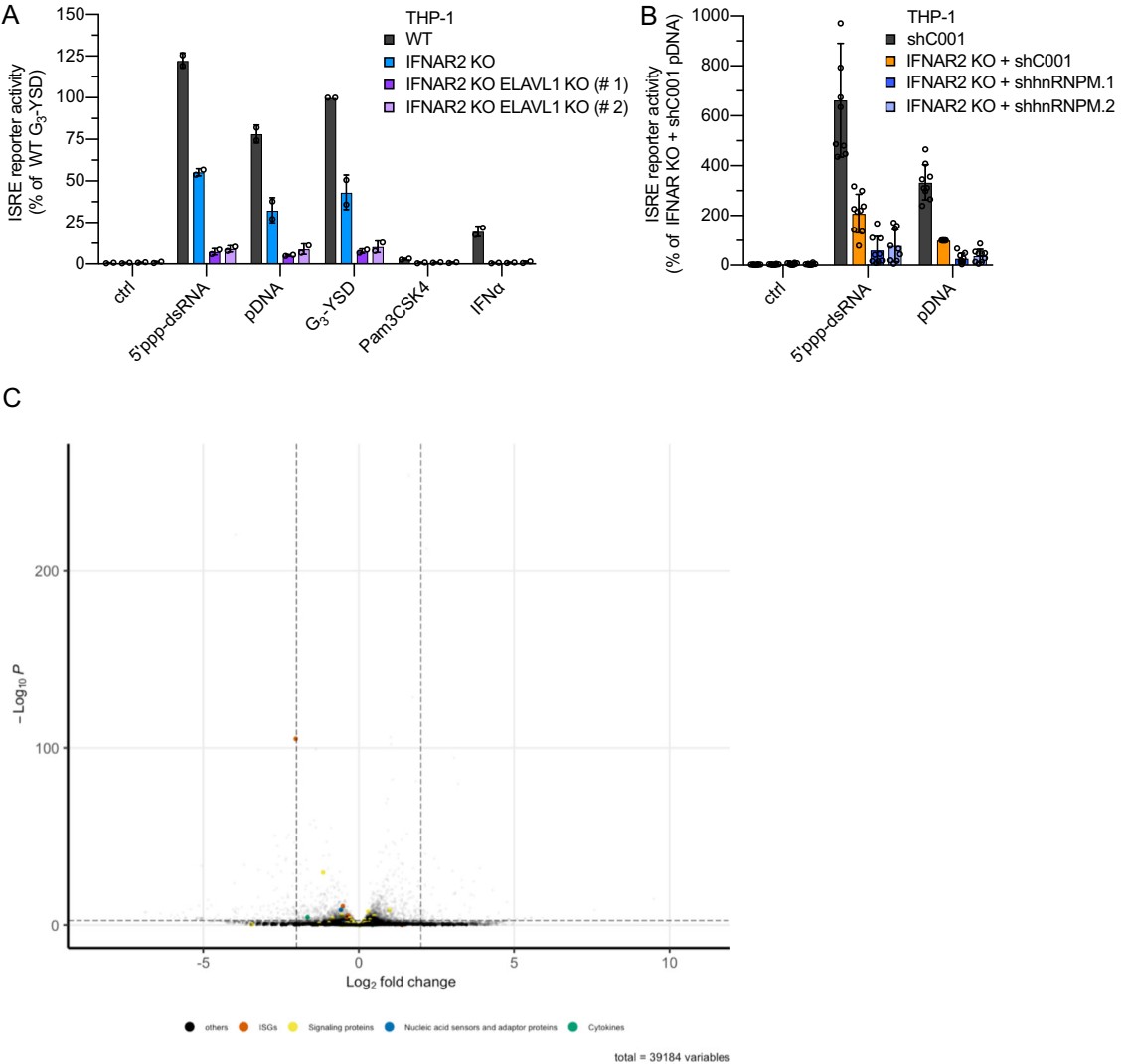

**Figure EV5. ELAVL1 regulates signal transduction downstream of cGAS and RIG-I.**

(A) ISRE reporter activation in THP-1 WT, IFNAR2 KO and IFNAR2/ELAVL1 double-KO cells (clones #1–2, ELAVL1 gRNA AN) 20 h after stimulation with 5'ppp-dsRNA (0.1 μg/ml), pDNA (0.1 μg/ml), G₃-YSD (0.5 μg/ml), Pam3CSK4 (0.5 μg/ml), or IFNα (5000 U/ml) (mean ± SD: stimuli from left to right: $n = 2, 2, 2, 2, 2$ independent experiments). ctrl, non-stimulated. (B) ISRE reporter activation in shC001-expressing THP-1 cells and IFNAR2 KO cells expressing shC001, shhnRNPM.1, or shhnRNPM.2 20 h after stimulation with 5'ppp-dsRNA (0.1 μg/ml) or pDNA (0.1 μg/ml) (mean ± SD; stimuli from left to right: $n = 10, 9, 9$ independent experiments).). ctrl, non-stimulated. (C) 3'-mRNA sequencing of total RNA from IFNα-stimulated (1000 U/ml, 6 h) THP-1 ELAVL1 KO (#1) and WT cells. Wald test was used to identify differentially expressed genes, with $P$ values adjusted for multiple testing using the Benjamini–Hochberg procedure. The Volcano plot correlates the gene expression (log₂ fold change of ELAVL1 KO vs. WT cells) with the -log₁₀ adjusted $P$ value ($P_{\text{adjusted}}$). Significantly regulated genes were defined as $P_{\text{adjusted}} < 0.05$ and log2 fold change >2 or log2 fold change > -2 (nucleic acid sensors and adaptor proteins (blue): cGAS, DDX58, IFIH1, MAVS, TMEM173; signaling proteins (yellow): IKBIP, TBK1, CHUK, IKBKB, IKBKE, IRF1, IRF2, IRF3, IRF4, IRF5, IRF7, IRF9, TICAM1, MYD88, TRAF1, TRAF2, TRAF3, TRAF5, TRAF6, TRAF7, TRIM25, RNF135, HMGB1, TFAM, ZCCHC3, G3BP1, NONO, IFI16, TTLL4, TTLL6, IFI16, DDX60, DHX58, IFNAR1, IFNAR2; ISGs (red): IFIT1, IFIT2, IFIT3, MX1, IL6, TNFA, IFI44L, IFI16, OASL, OAS1, OAS2, OAS3; cytokines (green): IFNB1, IFNL1, CXCL10).

