## [Peer Review File · The EMBO Journal]

RNA-binding proteins hnRNPM and ELAVL1 promote type-I interferon induction downstream of the nucleic acid sensors cGAS and RIG-I

Martin Schlee, Alexander Kirchhoff, Anna Herzner, Christian Urban, Antonio Piras, Robert Düster, Julia Mahlberg, Agathe Grünewald, Thais Schlee-Guimarães, Katrin Ciupka, Petro Leka, Robert Bootz, Christina Wallerath, Charlotte Hunkler, Ann De Regt, Beate Kümmerer, Maria Christensen, Florian Schmidt, Min Ae Lee-Kirsch, Claudia Günther, Hiroki Kato, Eva Bartok, Gunther Hartmann, Matthias Geyer, and Andreas Pichlmair

Corresponding author(s): Martin Schlee (martin.schlee@uni-bonn.de) , Alexander Kirchhoff (alkirchhoff@web.de)

Review Timeline:

Submission Date:	20th Mar 24
Editorial Decision:	8th May 24
Revision Received:	12th Sep 24
Editorial Decision:	4th Nov 24
Revision Received:	14th Nov 24
Accepted:	15th Nov 24

Editor: Ioannis Papaioannou

Transaction Report:

Dear Prof. Schlee,

Thank you for submitting your manuscript EMBOJ-2024-117355 for consideration by The EMBO Journal, and for your patience during peer review. It has now been seen by two experts in the field, and we have received the full set of their comments, which you can find below.

As you will see, both referees -who have provided detailed and well-informed reports- are largely supportive of the study. They recognize the novelty and significance of the findings, and they acknowledge the amount of the presented data as well as the fact that the work is for the most part well-performed and described. However, they also identify some limitations in the study and the manuscript, and they provide a number of constructive comments and suggestions for the improvement of the manuscript and further strengthening of its conclusions.

Given the referees' comments and recommendations, I would like to invite you to submit a revised version of the manuscript along with a detailed point-by-point response addressing all referees' comments. I should add that it is EMBO Journal policy to allow only a single round of major revision, and acceptance of your manuscript will therefore depend on the completeness of your responses in this revised version. Please let me know if you have any questions or comments that you would like to discuss with me.

We generally allow three months as standard revision time (August 7, 2024). As a matter of policy, competing manuscripts published during this period will not negatively impact our assessment of the conceptual advance presented by your study. However, we request that you contact us as soon as possible upon publication of any related work, to discuss how to proceed. Should you foresee a problem in meeting this three-month deadline, please let us know in advance and we may be able to grant an extension.

Thank you for the opportunity to consider your work for publication in The EMBO Journal. I look forward to your revision.

Best regards,

Ioannis

Instructions for preparing your revised manuscript

1. When you are ready to submit the revision, please upload:

- A Word file of the manuscript text (including legends of main Figures, EV Figures and Tables). Please make sure that changes are highlighted (or "tracked") to be clearly visible.

- Individual production-quality figure files (one file per figure). When assembling your figures, please refer to our figure preparation guidelines in order to ensure proper formatting and readability in print as well as on screen:

If the data shown in a figure are obtained from n {less than or equal to} 2, please use scatter plots showing the individual data points.

- i. the name of the statistical test used to generate error bars and P values
- ii. the number (n) of independent experiments (please specify technical or biological replicates) underlying each data point (discussion of statistical methodology can be reported in the Materials and Methods section, but figure legends should contain a basic description of n , P, and the test applied)
- iii. the nature of the bars and error bars (s.d., s.e.m.).

- A point-by-point response to the referees' comments, with a detailed description of the changes made (as a word file). All referees' concerns must be fully addressed and their suggestions taken on board. When preparing your letter of response to the

referees' comments, please bear in mind that this will form part of the Review Process File and will therefore be available online to the community. Please note that you have the possibility to opt out of the transparent process at any stage prior to publication by letting the editorial office know (contact@embojournal.org); if you do opt out, the Review Process File link will point to the following statement: "No Review Process File is available with this article, as the authors have chosen not to make the review process public in this case.". For more details on our Transparent Editorial Process, please visit our website: <https://www.embopress.org/page/journal/14602075/authorguide#transparentprocess>

- Expanded View (EV) files (replacing Supplementary Information) that are collapsible/expandable online. A maximum of 5 EV Figures can be typeset. EV Figures should be cited as "Figure EV1, Figure EV2" etc. in the text, and their respective legends should be included in the manuscript file after the legends of regular figures. See detailed instructions regarding Expanded View files here:

- For the figures that you do NOT wish to display as Expanded View figures, they should be bundled together with their legends in a single PDF file called "Appendix", which should start with a short Table of Contents (including page numbers). Appendix figures should be referred to in the main text as: "Appendix Figure S1, Appendix Figure S2" etc. Please see detailed instructions here: <https://www.embopress.org/page/journal/14602075/authorguide#expandedview>

- A complete author checklist, which you can download from our author guidelines (<https://www.embopress.org/page/journal/14602075/authorguide>). Please note that the checklist will also be part of the Review Process File.

2. Please note that no statistics should be calculated and shown in Figures if $n=2$. Please also note that each p value should be reported as an exact value.

3. Before submitting your revision, primary datasets (and computer code, where appropriate) produced in this study need to be deposited in appropriate public databases (see <https://www.embopress.org/page/journal/14602075/authorguide#dataavailability>). In particular, you are kindly requested to deposit all RNA sequencing and mass spectrometry data produced in your study. The accession numbers and databases should be listed in a formal "Data availability" section (placed after Materials and Methods) that follows the model below (see also <https://www.embopress.org/page/journal/14602075/authorguide#dataavailability>):

Data availability

- RNA-seq data: Gene Expression Omnibus GSE46843 (<https://www.ncbi.nlm.nih.gov/geo/query/acc.cgi?acc=GSE46843>)
- [data type]: [name of the resource] [accession number/identifier/doi] ([URL or identifiers.org/DATABASE:ACCESSION])

*** All links should resolve to a page where the data can be accessed. ***

*** Please remember to provide in the Data availability section of your revised manuscript reviewer passwords if the datasets are not yet public. ***

*** The Data Availability Section is restricted to new primary data that are part of this study. In case you have no data that require deposition in a public database, please state so instead of referring to the database: "Our study includes no data deposited in public repositories." under the heading "Data availability". ***

4. Please check that the title and the abstract of the manuscript are brief, yet explicit, even to non-specialists. The length of the title should not exceed 100 characters, and the abstract should be a single paragraph not exceeding 175 words.

5. Please also note our reference format: <https://www.embopress.org/page/journal/14602075/authorguide#referencesformat>.

7. Please remember: digital image enhancement is acceptable practice, as long as it accurately represents the original data and conforms to community standards. If a figure has been subjected to significant electronic manipulation, this must be noted in the figure legend or in the "Materials and Methods" section. The editors reserve the right to request original versions of figures and the original images that were used to assemble the figure.

8. Our journal encourages inclusion of data citations in the reference list to directly cite datasets that were obtained from public databases. Data citations in the article text are distinct from normal bibliographical citations and should directly link to the database records from which the data can be accessed. In the main text, data citations are formatted as follows: "Data ref:

Smith et al, 2001" or "Data ref: NCBI Sequence Read Archive PRJNA342805, 2017". In the Reference list, data citations must be labeled with "[DATASET]". A data reference must provide the database name, accession number/identifiers, and a resolvable link to the landing page from which the data can be accessed at the end of the reference. Further instructions are available at: <https://www.embopress.org/page/journal/14602075/authorguide#referencesformat>.

9. We request authors to consider both actual and perceived competing interests. Please review our policy (<https://www.embopress.org/page/journal/14602075/authorguide#conflictsofinterest>) and update your competing interests statement if necessary. Please name this section 'Disclosure and competing interests statement' and place it after the Acknowledgements section.

10. Please note that all corresponding authors are required to provide an ORCID ID upon submission of a revised manuscript (<https://orcid.org/>). Please find instructions on how to link your ORCID ID to your account in our manuscript tracking system in our Author guidelines (<https://www.embopress.org/page/journal/14602075/authorguide#authorshipguidelines>).

11. We use CRediT to specify the contributions of each author in the journal submission system. CRediT replaces the author contribution section, which should be removed from the manuscript. Please use the free text box to provide more detailed descriptions. See also guide to authors: <https://www.embopress.org/page/journal/14602075/authorguide#authorshipguidelines>.

13. We would also welcome the submission of cover suggestions or motifs to be used by our Graphics Illustrator in designing a cover.

14. Please use the link below to submit your revision:
<https://emboj.msubmit.net/cgi-bin/main.plex>

Referee #1:

RIG-I and cGAS are cytosolic nucleic acid sensors for RNA and DNA detection, respectively. The author discovered that hnRNPM is a positive regulator of type I interferon through ISRE reporter, interactome analysis, genome editing, confocal microscopy and other technologies. They reported that hnRNPM can interact with ELAVL1 and TBK1. A large body of data support the conclusion that hnRNPM and ELAVL1 were the first non-redundant signal elements that integrate the cGAS-STING pathway and RIG-I-MAVS pathway. This study provides a wealth of information that hnRNPM and ELAVL1 are new potential targets for antiviral defense and autoinflammatory diseases. This review has some comments for the authors to consider.

1. It is generally believed that hnRNPM is localized in the nucleus, but hnRNPM can also be localized in the cytoplasm in this paper, which indicates that hnRNPM can shuttle between the nucleus and cytoplasm. Have the authors screened related proteins that assist hnRNPM in shuttling between these two compartments in FIG2? Is there NLS (Nuclear Localization Signal) or NES (Nuclear Export Signal) in the protein sequence of hnRNPM?
2. Can the hnRNPM-ELAVL1-P65-IKK β / ξ complex enter the nucleus together to promote the expression of inflammation-related factors?
3. For better control, the author should add the number of cells under the white light view or DAPI staining.
4. The author should add the mRNA expression of IFN β and cxcl10 by RT-qPCR in order to further confirm the phenotype of Fig. 1D.
5. In Fig. 2, the proteins pulled down by hnRNPM include ELAVL1 and NF-kB. But, whether the hnRNPM-ELAVL1-NF-kb complex contains these proteins, such as p65, TBK1, IKK β / ξ , is not clear.
6. In Fig. S2, (figure C5 (immune system process structural constituent of cytoskeleton) and figure C8 (immune system process response to cytokine regulation of cell death leucocyte activation)), among all the screened proteins, are proteins related to the cGAS-STING pathway and RIGI-MAVS pathway included?
7. In Fig. 3E, the author should add the TBK1 phosphorylation, protein levels of TBK1 and IRF3 in the input and IP for better

controls.

8. In Fig. 5A, hnRNPM can interact with IKK β and IKK ξ . Why can't ELAVL1 pull down IKK β and IKK ξ in Fig. 5B? If the interaction between ELAVL1 and IKK β , IKK ξ is weak or the signal is weak, whether the signal could be detected by increasing the loading sample?

Referee #2:

In this manuscript, Kirchhoff et al. demonstrated that hnRNPM and ELAVL1 are critical for the phosphorylation of IRF3 and the induction of type I interferon (IFN), functioning downstream of the cGAS-STING and RIG-I-MAVS pathways. Through interactome analysis, they identified ELAVL1 as a key hnRNPM-associated protein that impacts type I IFN responses. The depletion of either hnRNPM or ELAVL1 reduced type I IFN production in response to HSV-1 and SeV infections. Through co-immunoprecipitation and PLA assays, they also found that both proteins were found to interact with TBK1 and NF- κ B, and their interactions occur in cytosolic and perinuclear regions. Furthermore, pharmacological inhibition of ELAVL1 significantly decreased cytokine release in fibroblasts from patients with type-I interferonopathy, such as AGS and SAVI.

Overall, their finding that hnRNPM and ELAVL1 form complexes downstream of RIG-I and cGAS, integrating these signaling pathways to connect with the antiviral response, is novel and of significant importance. Furthermore, throughout the manuscript, the experiments are well designed and well done. However, adequate datasets to support their conclusions are not provided in some sections. Furthermore, objective interpretation and description are necessary due to the occasional arbitrary interpretation of the data. To strengthen their conclusions, the following list of concerns should be addressed.

Comments

1. In figure 1I, they suggest that IRF3 phosphorylation was decreased upon hnRNPM depletion. However, in the actual data, it appears that hnRNPM depletion also reduces IRF3 protein expression by approximately 50%. Furthermore, in figure 1B-H, the impact of hnRNPM depletion on hnRNPM mRNA and ISRE activity also appears to be approximately 50%. Taking these into consideration, the possibility that hnRNPM may simply positively regulate the IRF3 protein expression cannot be ruled out. Furthermore, a complicating factor is that in the Western blot analyses using lysates from cells stably expressing hnRNPM (Figures 3E and 5A), the results from hnRNPM and control samples are presented separately. This separation makes it impossible to determine whether stable expression of hnRNPM affects the protein expression of various signaling molecules downstream of RIG-I/cGAS. For these reasons, they should carefully assess the protein expression of each signaling molecule on the same membrane for both hnRNPM-stably expressing cells and control cells. Additionally, similar experiments should be done using cells with hnRNPM depletion.
2. They selected ELAVL1 as an interacting partner of hnRNPM and as the subject of subsequent analysis, based on the information in lines 224-227, "Although KD of the targeted...". However, the results in figure S3B also show a decrease in luciferase activity for SUB1, HP1BP3, and hnRNPA1, as well as an increase in luciferase activity for hnRNPH2 and ILF2, besides ELAVL1. This selection appears highly arbitrary. Moreover, in this experiment, knockdown of the involved genes does not necessarily lead to a decrease in ISRE activity; an increase is also possible. Given these considerations, it is odd to solely focus on genes with decreased luciferase activity, particularly on ELAVL1 alone. Therefore, they need to provide clear explanations or use data to demonstrate the basis for their selection.
3. In figure 4F, they showed NF- κ B reporter activity upon various stimuli. It is well known that the activation of the cGAS-STING pathway can induce NF- κ B activation in a STING-TAK1-IKKs axis-dependent manner. Nevertheless, cGAMP did not increase NF- κ B reporter activity, and they did not mention this at all. They should clearly explain about this point.
4. In figure 4C, ISRE reporter activity in ELAVL1 KO cells stably expressing ELAVL1-FLAG, is restored to levels comparable to control cells. However, the expression of IFNB1 and CXCL10 mRNA in figure 4G and 4H, as well as the production of CXCL10 protein in figure S4D, were not restored in several samples. They should explain about these discrepancies. Furthermore, to strengthen their conclusion, they should include the data showing mRNA expression level upon pDNA stimulation in figure 4G-I, and the data showing CXCL10 protein expression upon G3-YSD in figure S4D.
5. In figure 5A and 5B, they describe that the interaction of hnRNPM with IKK β , TBK1, IKK ξ , and NF- κ B p65 were independent of RNA, and that the interaction between hnRNPM and ELAVL1 was slightly decreased upon RNase A treatment. While these points are indeed factual, in Figure 5A, the interaction between hnRNPM and pTBK1-Ser172 or NF- κ B p65-Ser536 is significantly reduced by RNase A treatment. These findings may suggest critical points in the regulation of type I IFN production via the RIG-I/cGAS pathway by hnRNPM, thus warranting further evaluation.
6. At line 367-370, they describe that "Although, we cannot exclude that ...". However, without providing evidence for this, there is a potential for arbitrary interpretation. To strengthen their conclusion, they should evaluate the interaction between ELAVL1 and IKKs also using PLA assays.
7. At lines 170-173, they express 'unexpectedly strong inhibition,' but this expression feels inappropriate. This is because both the knockdown efficiency (Figure 1B) and the ISRE inhibition efficiency (Figure 1D) are around 50-70%, and there is no significant difference.
8. The manuscript lacks concise explanations or full names for some specialized terms, resulting in the sudden appearance of terms throughout the text. For readability purposes, specialized terms should include their full names or brief explanations upon their first occurrence. For example, CVB3 and PV 3C.
9. At lines 249-250, they state that 99.0% of the ELAVL1 interactors interact with hnRNPM, but in reality, it is 93.8% (76/81).

10. In Figure 2B, DDX3X is missing, resulting in 27 squared words.
11. For readability purposes, it might be better to place the controls on the left in all figures.
12. In figure S1A, they should use STAT1 instead of b-actin as a control for phosphorylated STAT1.
13. In figure S4A and B, to strengthen their conclusion, it is better to add cGAMP stimulation.
14. At lines 512-515, references should be provided.
15. In figure S5B, the label on the Y-axis appears to be incorrect.
16. All of tables are missing.

Point-by-point response Kirchhoff et al.

Foremost, we would like to thank both reviewers and the editor for their valuable time, during which they evaluated our manuscript in great detail and provided positive and constructive feedback. We now present a thoroughly revised manuscript in which most of the critical points raised by the reviewers have been addressed. We believe that the manuscript has been substantially improved by incorporating these comments and suggestions.

Referee #1:

RIG-I and cGAS are cytosolic nucleic acid sensors for RNA and DNA detection, respectively. The author discovered that hnRNPM is a positive regulator of type I interferon through ISRE reporter, interactome analysis, genome editing, confocal microscopy and other technologies. They reported that hnRNPM can interact with ELAVL1 and TBK1. A large body of data support the conclusion that hnRNPM and ELAVL1 were the first non-redundant signal elements that integrate the cGAS-STING pathway and RIG-I-MAVS pathway. This study provides a wealth of information that hnRNPM and ELAVL1 are new potential targets for antiviral defense and autoinflammatory diseases. This review has some comments for the authors to consider.

We thank the reviewer for reading our work and providing positive and supportive comments.

1. It is generally believed that hnRNPM is localized in the nucleus, but hnRNPM can also be localized in the cytoplasm in this paper, which indicates that hnRNPM can shuttle between the nucleus and cytoplasm. Have the authors screened related proteins that assist hnRNPM in shuttling between these two compartments in FIG2? Is there NLS (Nuclear Localization Signal) or NES (Nuclear Export Signal) in the protein sequence of hnRNPM?

Our nuclear and cytoplasmic extraction experiments show that hnRNPM is predominantly localized in the nucleus of THP-1 monocytes, while a portion of hnRNPM is also detectable in the cytoplasm (Fig. EV3C-EV3D). It has been shown that hnRNPM depletion by RNAi enhances the replication of certain alphaviruses (i.e., Semliki Forest virus, Sindbis virus, and Chikungunya virus) and that infection with these viruses triggers the translocation of hnRNPM from nucleus to cytoplasm (Varjak *et al.*, 2013). In contrast to Varjak *et al.*, we used synthetic ligands for RIG-I (5'ppp-dsRNA) and cGAS (G₃-YSD) to mimic viral infection. Upon treatment of THP-1 monocytes with these ligands, we did not observe an altered distribution of hnRNPM between nucleus and cytosol (Fig. EV3C-EV3D). Since these synthetic ligands do not fully mimic a viral infection and are used specifically to activate cGAS or RIG-I, it is conceivable that active cytosolic translocation of hnRNPM could be observed only during viral infection. Unfortunately, the viruses tested by Varjak *et al.* are currently not available to us. Similar to hnRNPM, ELAVL1 was predominantly localized to the nucleus of THP-1 cells and stimulation of cGAS or RIG-I did not alter its distribution between cytosol and nucleus (Fig. EV3C-EV3D). Since we were particularly interested in analyzing the roles of hnRNPM and ELAVL1 in cGAS and RIG-I signaling, we did not systematically investigate potential nucleocytoplasmic shuttling using more sophisticated methods in this study. However, in order to further analyze in which compartment the hnRNPM-ELAVL1 complex regulates cGAS and RIG-I signaling, we have now separated cytosol and nucleus of THP-1 dual wildtype cells, THP-1 dual ELAVL1 KO cells, and THP-1 dual ELAVL1 KO ELAVL1-FLAG-expressing cells 3 h after stimulation with 0.5 μ g/ml G₃-YSD or 0.1 μ g/ml 5'ppp-dsRNA (Revision Fig. 1A-1B).

Revision Fig. 1: KO of ELAVL1 reduces cytosolic and nuclear levels of pIRF3-Ser396.

THP-1 dual cells (WT), THP-1 dual ELAVL1 KO cells, and THP-1 dual ELAVL1 KO ELAVL1-FLAG-expressing cells were stimulated with for 3 h with 0.5 μ g/ml G₃-YSD (A) or 0.1 μ g/ml 5'ppp-dsRNA (B). Following, nuclear

extracts (NE) and cytoplasmic extracts (CE) were prepared and analyzed by immunoblotting with the indicated antibodies.

It is well-established that activation of cGAS or RIG-I by cytosolic ligands activates downstream kinases such as TBK1 and IKK ϵ (Liu *et al.*, 2015). These kinases then phosphorylate the downstream transcription factor IRF3, leading to IRF3 dimerization and translocation of dimeric IRF3 to the nucleus. In the nucleus, IRF3 induces the expression of type I IFN. We hypothesized that a decrease in IRF3 phosphorylation only in the nucleus, and not in the cytoplasm, of ELAVL1 knockout (KO) cells would indicate that the hnRNPM-ELAVL1 complex functions within the nucleus. By contrast, a decrease in IRF3 phosphorylation in both cytoplasm and nucleus would rather suggest that the hnRNPM-ELAVL1 complex regulates signaling in the cytosol, since cytosolic phosphorylation of IRF3 occurs before its nuclear translocation. Of note, we observed that IRF3 phosphorylation triggered by activation of cGAS or RIG-I is decreased in both cytosol and nucleus of ELAVL1 KO cells (Revision Fig. 1A-1B), indicating that the hnRNPM-ELAVL1 complex regulates IRF3 phosphorylation in the cytoplasm. In further support of this observation, we also detected reduced levels of phosphorylated TBK1 in the cytosol of ELAVL1 KO cells upon challenge with cytosolic G₃-YSD or 5'ppp-dsRNA as well as reduced levels of phosphorylated STING after stimulation of cGAS (Revision Fig. 1A-1B). Interestingly, we also detected strong signals of phosphorylated TBK1 in the nucleus of THP-1 WT cells, which warrants further investigation. Nuclear phosphorylated TBK1 has already been observed by Du *et al.* in L929 and RAW 264.7 cells upon stimulation with the murine STING ligand DMXAA (Du *et al.*, 2015). In summary, these data suggest that the hnRNPM-ELAVL1 complex promotes cGAS and RIG-I mediated phosphorylation of kinases and transcription factors in the cytoplasm and that translocation from nucleus to cytoplasm is not key for its function.

hnRNPM contains a non-classical basic proline-tyrosine (bPY)-NLS in the N-terminal region between amino acids 41-70 which was shown to interact with Transportin-1 (TNPO1, Karyopherin β 2) (Cansizoglu *et al.*, 2007; Lee *et al.*, 2006). To analyze whether the reported N-terminal bPY-NLS indeed controls nuclear localization of hnRNPM in THP-1 monocytes, we generated N-terminally truncated versions of hnRNPM either lacking the first 70 N-terminal amino acids (hnRNPM 71-730) or the first 195 N-terminal amino acids (hnRNPM 196-730) and stably expressed them in THP-1 cells. Following, we analyzed the distribution of these hnRNPM variants in cytosol and nucleus. We found that deletion of the amino acids 1-70 of hnRNPM indeed increased its cytosolic localization compared to full-length hnRNPM (hnRNPM 1-730) (Revision Fig. 2). However, only when additional N-terminal amino acids were deleted, the localization of hnRNPM completely shifted to the cytosol (hnRNPM 196-730), suggesting that residues C-terminal of the reported bPY-NLS also control nuclear retention of hnRNPM (Revision Fig. 2).

Revision Fig. 2: Subcellular localization of N-terminally truncated hnRNPM variants.

Nuclear extracts and cytoplasmic extracts of non-stimulated THP-1 cells stably expressing full-length hnRNPM (hnRNPM 1-730) or N-terminally truncated hnRNPM variants (hnRNPM 71-730, hnRNPM 196-730) were analyzed by immunoblotting with the indicated antibodies.

Although hnRNPM has been reported to shuttle between cytosol and nucleus upon infection with certain alphaviruses (Varjak *et al.*, 2013), no NES has been described so far. Thus, we have run an *in silico* analysis using the tool LocNES to predict whether hnRNPM contains a classical NES for exportin-1 (XPO1) (Revision Fig. 3) (Xu *et al.*, 2015). As shown in Revision Fig. 3, multiple potential NESs with high probability score > 0.3 (column 4) were predicted to be present in hnRNPM. Similar to the approach confirming the NLS of hnRNPM described above (Revision Fig. 2), hnRNPM variants lacking the predicted NES could be expressed in THP-1 cells to assess whether they control the distribution of hnRNPM between cytosol and nucleus. Although these experiments would certainly provide further insight regarding the subcellular localization of hnRNPM, we hope that the reviewer agrees that we consider it to be beyond the scope of the current manuscript.

Protein Name	Position	Sequence	Score
>LocNES499260326_0	4-18	GVEAAAEVAATEIKM	0.019
>LocNES499260326_0	66-80	NPTKRYRAFIINIPF	0.005
>LocNES499260326_0	89-103	DLVKEKVGVEVYVEL	0.031
>LocNES499260326_0	91-105	VKEKVGVEVYVELLM	0.022
>LocNES499260326_0	155-169	RAMQKVMATTGGMG	0.003
>LocNES499260326_0	194-208	IHALQAGRLGSTVFV	0.000
>LocNES499260326_0	217-231	WKKLKEVFSMAGVVV	0.004
>LocNES499260326_0	244-258	RGIGTVTFEQSIEAV	0.061
>LocNES499260326_0	249-263	VTFEQSI EAVQAISM	0.320
>LocNES499260326_0	256-270	EAVQAISM FNGLLF	0.219
>LocNES499260326_0	292-306	ERPQQLPHGLGGIGM	0.124
>LocNES499260326_0	312-326	GQPIDANHLNKGIGM	0.028
>LocNES499260326_0	325-339	GMGNIGPAGMGMEGI	0.068
>LocNES499260326_0	327-341	GNIGPAGMGMEGIGF	0.047
>LocNES499260326_0	332-346	AGMGMEGIGFGINKM	0.095
>LocNES499260326_0	355-369	GGMENMGRFSGSMNM	0.063
>LocNES499260326_0	397-411	SVPGIERMGPPIGIDL	0.126
>LocNES499260326_0	423-437	LGHGMDRVGSEIERM	0.251
>LocNES499260326_0	425-439	HGMDRVGSEIERMGL	0.310
>LocNES499260326_0	430-444	VGSEIERMGLVMDRM	0.230
>LocNES499260326_0	433-447	EIERMGLVMDRMGSV	0.252
>LocNES499260326_0	443-457	RMGSVERMGSGERM	0.394
>LocNES499260326_0	448-462	ERMGSGERMGLPLGL	0.307
>LocNES499260326_0	458-472	GPLGLDHMASSIERM	0.048
>LocNES499260326_0	465-479	MASSIERMGQTMERI	0.033
>LocNES499260326_0	472-486	MGQTMERIGSGVERM	0.076
>LocNES499260326_0	478-492	RIGSGVERMGAGMGF	0.011
>LocNES499260326_0	490-504	MGFGLERMAAPIDRV	0.102
>LocNES499260326_0	497-511	MAAPIDRVGQTIERM	0.026
>LocNES499260326_0	504-518	VGQTIERMGSVERM	0.100
>LocNES499260326_0	511-525	MGSVERMGPPIERM	0.581
>LocNES499260326_0	513-527	SGVERMGPPIERMGL	0.226
>LocNES499260326_0	518-532	MGPAIERMGLSMERM	0.586
>LocNES499260326_0	537-551	MGAGLERMGPVMDRM	0.164
>LocNES499260326_0	544-558	MGVMDRMATGLERM	0.097
>LocNES499260326_0	554-568	GLERMGANLERMGL	0.165
>LocNES499260326_0	567-581	GLERMGANSLERMGL	0.359
>LocNES499260326_0	595-609	AMGPALGAGIERMGL	0.058
>LocNES499260326_0	643-657	GHAPGVARKACQIFV	0.006
>LocNES499260326_0	648-662	VARKACQIFVRNLPF	0.013
>LocNES499260326_0	703-717	EVAERACRMMNGMKL	0.034

Revision Fig. 3: Output of *in silico* prediction of classical NES in human full-length hnRNPM protein sequence using LocNES (Xu *et al*, 2015).

2. Can the hnRNPM-ELAVL1-P65-IKK β / ξ complex enter the nucleus together to promote the expression of inflammation-related factors?

We have not specifically addressed this question experimentally in this manuscript. However, as shown in Revision Fig. 1, our data shows that pIRF3-Ser396 induced by stimulation of cGAS or RIG-I are reduced in both cytosol and nucleus of THP-1 ELAVL1 KO cells. As discussed in response to comment 1 of reviewer 1, we interpret these data that the hnRNPM-ELAVL1 complex regulates IRF3 phosphorylation in the cytoplasm. This is supported by our PLA analysis, showing mostly cytoplasmic and, to a smaller extent, also peri-nuclear interactions of hnRNPM/ELAVL1 with activated TBK1 (Fig. 5C-5D). Collectively, our data suggests that the hnRNPM-ELAVL1 complex represents a platform for kinases such as TBK1, IKK ϵ , and IKK β that facilitates cytosolic phosphorylation of IRF3.

3. For better control, the author should add the number of cells under the white light view or DAPI staining.

We thank the reviewer for this valuable suggestion. We have made the following modifications to improve data interpretation:

Fig. 1A:

We have added the brightfield images below the corresponding fluorescence images in Fig. 1A. Due to the high cell density, we were unable to count the cells but hope that the displayed images allow a better qualitative evaluation of the data.

Fig. 5C-5E:

We have included the number of cells above each sample group in the dot plots. Since the microscopy images on the left side of each plot are representative samples of all images acquired, we have added the cell counts above each sample group in the plots rather than in the microscopy images.

4. The author should add the mRNA expression of IFN β and cxcl10 by RT-qPCR in order to further confirm the phenotype of Fig. 1D.

We thank the reviewer for her/his insight and included the newly generated RT-qPCR data of IFN β and CXCL10 mRNA expression in THP-1 hnRNPM KD cells into the revised manuscript (Fig. EV1A-EV1B) (lines 223-226). As expected, we detected reduced levels of IFN β mRNA in hnRNPM KD cells after stimulation with 5'ppp-dsRNA, pDNA, or G $_3$ -YSD. As described in the response to comment 3 of referee 2, absence of IFN β mRNA induction by 2'3'-cGAMP may be a technical issue caused by insufficient cytosolic delivery of the stimulus (lines 749-775). In contrast to 5'ppp-dsRNA, pDNA or G $_3$ -YSD, 2'3'-cGAMP cannot be lipofected and is added to the cells directly. Consequently, a substantial amount of 2'3'-cGAMP may not cross the plasma membrane, preventing it from activating STING in the cytosol. Note that 2'3'-cGAMP was already used at 10 μ g/ml, compared to 5'ppp-dsRNA, pDNA and G $_3$ -YSD, which were used at 0.1 μ g/ml, 0.1 μ g/ml and 0.5 μ g/ml, respectively. It is conceivable that extending the incubation time with 2'3'-cGAMP from 6 hours to 16 hours could enable the detection of IFN β mRNA, which we did not further investigate during the revision.

Interestingly, in contrast to IFN β mRNA, CXCL10 mRNA levels induced by stimulation of cGAS or RIG-I were unchanged between hnRNPM KD and control cells, suggesting different regulation mechanisms of IFN β and CXCL10 mRNA expression.

Since IFN β expression is a hallmark of cGAS or RIG-I activation and induction of CXCL10, which can be induced by various cytokines, including type I and type III IFNs (Goel *et al*, 2020), is only an indirect consequence of cGAS or RIG-I stimulation, we hope the reviewer agrees that these data further confirm the phenotype shown in Fig. 1D. Nonetheless, it requires further investigation why KO of ELAVL1 but not KD of hnRNPM reduces CXCL10 expression in response to stimulation of cGAS or RIG-I. Hypothetically, absence of ELAVL1 could have broader impact on transcript levels since ELAVL1 is known to bind AU-rich elements located on the 3'UTR of transcripts (Rothamel *et al*, 2021). It is also conceivable that downregulated CXCL10 mRNA levels induced by cGAS or RIG-I can only be observed in hnRNPM knockout cells, which we (and others) were unable to generate despite multiple attempts (lines 199-202).

5. In Fig. 2, the proteins pulled down by hnRNPM include ELAVL1 and NF- κ B. But, whether the hnRNPM-ELAVL1-NF- κ B complex contains these proteins, such as p65, TBK1, IKK β / ξ , is not clear.

In this study, we have mapped the interactome of hnRNPM in THP-1 monocytes using AP-MS and detected interactions of hnRNPM with ELAVL1 and IKK β with this method (Fig. 2B). By contrast, we observed co-precipitation of hnRNPM with ELAVL1 and IKK β as well as TBK1, IKK ϵ , and NF- κ B p65 by co-IP (Fig. 5A). It is possible that due to low peptide ionization of TBK1, IKK ϵ , and NF- κ B p65, interaction with hnRNPM could not be captured by AP-MS. It is well-known that peptide ionization can be a limiting factor in proteomics, as not all peptides ionize equally well (Aebersold & Mann, 2003). This phenomenon may explain why interactions of hnRNPM with TBK1, IKK ϵ , and NF- κ B p65 were only captured by co-IP. To provide a second layer of specificity, we also assessed whether ELAVL1 can interact with these signaling proteins. In the course of the revision, we were able to confirm co-precipitation of hnRNPM, TBK1, IKK ϵ , IKK β , and NF- κ B p65 with ELAVL1 (see details in the response to comment 8 of reviewer 1) (Fig. 5B). As discussed below, we think that these new data substantially strengthen our conclusion that hnRNPM, ELAVL1, TBK1, IKK ϵ , IKK β , and NF- κ B p65 form a signaling complex that regulates both cGAS and RIG-I signaling. In addition, we used proximity ligation assay (PLA) as an orthogonal method to confirmed interactions of hnRNPM with pTBK1-Ser172. Note that the signal amplification provided by the detection antibodies used for immunoblotting and the rolling-circle amplification of the PLA reaction likely increased signal strength and thus may enable detection of the mentioned signaling proteins with hnRNPM by co-IP/PLA compared to AP-MS.

6. In Fig. S2, (figure C5 (immune system process structural constituent of cytoskeleton) and figure C8 (immune system process response to cytokine regulation of cell death leucocyte activation)), among all the screened proteins, are proteins related to the cGAS-STING pathway and RIGI-MAVS pathway included?

In Fig. EV2 (formerly Fig. S2), only the strongest association with the displayed GO terms is shown for each identified hnRNPM interactor. In this representation, the two largest clusters represent RNA-binding proteins involved in translation (C1) or splicing (C2). Compared to clusters C1 and C2, hnRNPM interactors mapped to cluster C5 (GO terms "immune system process" and "structural constituent of cytoskeleton") or cluster C8 (GO terms "immune system process", "response to cytokine", "regulation of cell death", and "leukocyte activation") are less enriched. However, this does not mean that hnRNPM interactors not mapped to clusters C5 and C8 are generally unrelated to immune system processes, as only the strongest association with the displayed GO terms is shown.

By searching the literature and examining the multiple GO terms associated with each hnRNPM interactor (see Table S1), we identified DHX9, DHX15, ELAVL1, PRKDC, DDX3X, and ZC3HAV1 as potential hnRNPM interactors

associated with positive regulation of type I IFN production. However, DHX9, DHX15, ELAVL1, DDX3X, and ZC3HAV1 were mapped to cluster C2, while PRKDC was mapped to cluster C6 (Fig. EV2). These proteins were not mapped to clusters C5 or C8 (immune system processes) because they had a stronger association with clusters C2 or C6. Nonetheless, given their association with type I IFN expression, we analyzed whether DHX9, DHX15, ELAVL1, PRKDC, DDX3X, and ZC3HAV1 are involved in cGAS or RIG-I signaling in our model system. Our functional RNAi screen demonstrated that only KD of ELAVL1 reduced the cGAS or RIG-I mediated ISRE reporter activation (Fig. EV3A-EV3B). As discussed in the manuscript (lines 703-707), it is conceivable that DHX9, DHX15, PRKDC, DDX3X, and ZC3HAV1 may have cell type- or stimulus-specific effects in nucleic acid sensing by the innate immune system.

Among the members of clusters C5 and C8 (Fig. EV2), only ARHGEF2 and CARD9 have been previously linked to nucleic acid immunity. In mice, GEF-H1, encoded by *ARHGEF2*, was reported to be involved in recognition of non-self RNA by RIG-I or MDA5, leading to the phosphorylation of IRF3 and subsequent secretion of IFN β (Chiang *et al*, 2014). Consequently, human ARHGEF2 was one of the most interesting interactors of hnRNPM that we functionally evaluated. However, shRNA-mediated KD of ARHGEF2 did not impair ISRE reporter induction following cGAS or RIG-I stimulation in our model system (Fig. EV3A-EV3B); therefore, we did not investigate ARHGEF2 further. Additionally, the innate immune adaptor protein CARD9 has been reported to form a complex with Rad50 that senses dsDNA following DNA virus infection, thereby mediating the activation of NF- κ B and the secretion of IL-1 β (Roth *et al*, 2014). Since we did not target CARD9 in the RNAi screen presented in this manuscript, we performed a KD of CARD9 in THP-1 dual cells (Revision Fig. 4). Although only one of the two tested shRNAs significantly reduced CARD9 mRNA expression (only shCARD.2 shown), we did not observe reduced ISRE reporter activity upon stimulation of cGAS or RIG-I. However, since the reduction in CARD9 mRNA expression was observed with only one shRNA, this finding should be validated using another CARD9-specific shRNA.

Revision Fig. 4: KD of CARD9 does not impair cGAS and RIG-I signaling in THP-1 monocytes.

(A) qPCR of CARD9 in THP-1 cells expressing control shRNA (shC001) or CARD9-specific shRNA (shCARD9.2) (mean \pm SD)

(B) ISRE reporter activation 20 h after stimulation with 5'ppp-dsRNA (0.1 μ g/ml) or pDNA (0.1 μ g/ml). ctrl, non-stimulated.

7. In Fig. 3E, the author should add the TBK1 phosphorylation, protein levels of TBK1 and IRF3 in the input and IP for better controls.

We greatly appreciate the reviewer's suggestion. In the originally submitted manuscript, Fig. 3E verified the co-immunoprecipitation of hnRNPM with ELAVL1. We included pIRF3-Ser396 in the input loading control solely to demonstrate the technical success of cGAS stimulation with pDNA. Interactions with pTBK1/TBK1 or lack of interaction with pIRF3/IRF3 were not included in this figure, as this aspect is specifically addressed in Fig. 5A. Experimentally, Fig. 3E and Fig. 5A were conducted using the same methodology, with Fig. 3E focusing on the interaction of hnRNPM with ELAVL1, while Fig. 5A highlights hnRNPM interactions with key signaling proteins involved in cytosolic nucleic acid sensing pathways.

Since our initial intention to separate Fig. 3E and Fig. 5A may be disruptive to the reader, as interaction data of hnRNPM is split across two figures, we decided to merge Fig. 3E and Fig. 5A. In the updated version of the manuscript, former Fig. 3E is deleted and the co-IP of ELAVL1 with hnRNPM is instead shown in Fig. 5A (lines 517-520). We now report the identification of ELAVL1 as an hnRNPM interactor by AP-MS in Fig. 3, continue with a detailed functional characterization of ELAVL1 in Fig. 4, and finally demonstrate in Fig. 5 that hnRNPM and ELAVL1 form a complex with essential signaling proteins of the cGAS and RIG-I signaling pathways. We believe this data consolidation reduces complexity and enhances the readability of the manuscript.

8. In Fig. 5A, hnRNPM can interact with IKK β and IKK ξ . Why can't ELAVL pull down IKK β and IKK ξ in Fig. 5B? If the interaction between ELAVL and IKK β , IKK ξ is weak or the signal is weak, whether the signal could be detected by increasing the loading sample?

We thank the reviewer for this suggestion. We have now repeated the IP of ELAVL1 using twice the amount of cell lysate, which indeed has allowed us to detect co-precipitation of IKK β and IKK ξ with ELAVL1 (Fig. 5B). Unlike IKK β , co-precipitation of IKK ξ with ELAVL1 was RNA-dependent, suggesting that yet unknown RNA species may bind to the hnRNPM-ELAVL1 complex and regulate type I IFN expression induced by cGAS or RIG-I (this point is specifically discussed in response to comment 5 of reviewer 2). In our opinion, these new data substantially increase the quality of this manuscript since we are now able to show that hnRNPM and ELAVL1 interact with the same signaling components activated downstream of both cGAS and RIG-I (TBK1, IKK ϵ , IKK β , NF- κ B p65). This further strengthens our conclusion that hnRNPM and ELAVL1 form an important signaling node promoting the expression of type I IFN induced by activation of cGAS or RIG-I. We have changed the text in the revised manuscript accordingly (lines 546-556).

Referee #2:

In this manuscript, Kirchhoff et al. demonstrated that hnRNPM and ELAVL1 are critical for the phosphorylation of IRF3 and the induction of type I interferon (IFN), functioning downstream of the cGAS-STING and RIG-I-MAVS pathways. Through interactome analysis, they identified ELAVL1 as a key hnRNPM-associated protein that impacts type I IFN responses. The depletion of either hnRNPM or ELAVL1 reduced type I IFN production in response to HSV-1 and SeV infections. Through co-immunoprecipitation and PLA assays, they also found that both proteins were found to interact with TBK1 and NF- κ B, and their interactions occur in cytosolic and perinuclear regions. Furthermore, pharmacological inhibition of ELAVL1 significantly decreased cytokine release in fibroblasts from patients with type-I interferonopathy, such as AGS and SAVI.

Overall, their finding that hnRNPM and ELAVL1 form complexes downstream of RIG-I and cGAS, integrating these signaling pathways to connect with the antiviral response, is novel and of significant importance. Furthermore, throughout the manuscript, the experiments are well designed and well done. However, adequate datasets to support their conclusions are not provided in some sections. Furthermore, objective interpretation and description are necessary due to the occasional arbitrary interpretation of the data. To strengthen their conclusions, the following list of concerns should be addressed.

We thank the reviewer for the supportive comments and helpful feedback.

Comments

1. In figure 1I, they suggest that IRF3 phosphorylation was decreased upon hnRNPM depletion. However, in the actual data, it appears that hnRNPM depletion also reduces IRF3 protein expression by approximately 50%. Furthermore, in figure 1B-H, the impact of hnRNPM depletion on hnRNPM mRNA and ISRE activity also appears to be approximately 50%. Taking these into consideration, the possibility that hnRNPM may simply positively regulate the IRF3 protein expression cannot be ruled out. Furthermore, a complicating factor is that in the Western blot analyses using lysates from cells stably expressing hnRNPM (Figures 3E and 5A), the results from hnRNPM and control samples are presented separately. This separation makes it impossible to determine whether stable expression of hnRNPM affects the protein expression of various signaling molecules downstream of RIG-I/cGAS. For these reasons, they should carefully assess the protein expression of each signaling molecule on the same membrane for both hnRNPM-stably expressing cells and control cells. Additionally, similar experiments should be done using cells with hnRNPM depletion.

To the first part of comment 1:

We appreciate the reviewer's concern that the total IRF3 protein levels appear to decrease similarly to pIRF3-Ser396 in THP-1 hnRNPM KD cells upon stimulation of cGAS or RIG-I. Signal quantification using ImageStudio Lite software (LI-COR Biosciences, version 5.2.5) showed that the induction of pIRF3-Ser396 relative to total IRF3 after stimulation of cGAS or RIG-I is decreased in THP-1 hnRNPM KD cells compared to shC001-expressing cells (Fig. EV1C). The relative decrease in pIRF3-Ser396 concentration is approximately 50%, which is consistent with the signal reduction observed in Figures 1D and 1F-1H of this manuscript. To be consistent throughout the manuscript, we have now also included signal quantification of the Western blots shown in Fig. 4J and 4K (phosphorylation analysis of THP-1 ELAVL1 KO cells) (see Fig. EV4F-EV4I). We hope that the newly added data strengthen our conclusion.

To the second part of comment 1:

We thank the reviewer for raising this important point and we realized that we should have included more detailed legends for the figures showing co-IPs. The samples from hnRNPM-GFP- and GFP-expressing cells shown in former Fig. 3E and Fig. 5A (former Fig. 3E was removed from revised manuscript and ELAVL1 blots shown in former Fig. 3E were moved to Fig. 5A; see response to comment 7 of reviewer 1 for details) were indeed probed on the same membrane. To improve readability, empty lanes or wells loaded with markers between the two sample groups were removed and thus the respective sample groups were displayed separately (please note that this was described in the legend of Fig. 5A in the initial submission). In the revised version of the manuscript, we have added a more detailed legend for Fig. 5A (lines 1674-1675). As suggested, we also quantified the signals of the input loading controls for all tested protein shown in Fig. 5A and did not observe strong differences in protein expression between hnRNPM-GFP- and GFP-expressing THP-1 monocytes (Revision Fig. 5). In addition, we analyzed whether hnRNPM KD affects the expression of the main signaling proteins of the cGAS and RIG-I pathways (Fig. EV1F-EV1G). As expected, KD of hnRNPM did not affect expression of the tested proteins. We hope these additional explanations and new data adequately address the reviewer's concern regarding protein expression.

Revision Fig. 5: Quantification of Western blot data of Fig. 5A.
mean \pm SD

2. They selected ELAVL1 as an interacting partner of hnRNPM and as the subject of subsequent analysis, based on the information in lines 224-227, "Although KD of the targeted...". However, the results in figure S3B also show a decrease in luciferase activity for SUB1, HP1BP3, and hnRNPA1, as well as an increase in luciferase activity for hnRNPH2 and ILF2, besides ELAVL1. This selection appears highly arbitrary. Moreover, in this experiment, knockdown of the involved genes does not necessarily lead to a decrease in ISRE activity; an increase is also possible. Given these considerations, it is odd to solely focus on genes with decreased luciferase activity, particularly on ELAVL1 alone. Therefore, they need to provide clear explanations or use data to demonstrate the basis for their selection.

We thank the reviewer for the thorough evaluation of our manuscript and completely agree that we should have provided more information regarding target selection. In this manuscript, we focused on ELAVL1 for the following reasons:

(a) Given that KD of hnRNPM leads to a reduced activation of the cGAS and RIG-I pathways, we were primarily interested to identify hnRNPM binding partners that also act as positive regulators of these pathways. Therefore, we did not further investigate the potential negative regulatory roles of ILF2 or hnRNPH2 in cGAS and RIG-I signaling. We agree with the reviewer that ILF2 and hnRNPH2 may have negative regulatory roles in these pathways, which warrants further investigation; however, we believe this is beyond the scope of the current manuscript.

(b) We set an arbitrary threshold of at least 50% ISRE reporter inhibition for both used shRNAs as a criterion to further evaluate hits from the RNAi screen using CRISPR-Cas9. Among all hits, only ELAVL1 and HP1BP3 met this criterion. KD of hnRNPA1, SUB1, or PRKDC did not reduce ISRE reporter activation induced by cGAS or RIG-I stimulation by 50%. In follow-up studies, a knockout (KO) of HP1BP3 did not efficiently reduce cGAS- and RIG-I-mediated ISRE activity, suggesting RNAi off-target effects (Revision Fig. 6). Consequently, HP1BP3 was excluded from deeper analysis.

Revision Fig. 6: KO of HP1BP3 does not impair cGAS- or RIG-I-mediated ISRE reporter activation.

We hope these explanations clarify why we focused on ELAVL1 in this manuscript. We agree with the reviewer that additional yet undiscovered positive (e.g., SUB1, hnRNPA1) or negative regulatory proteins (e.g., ILF2,

hnRNPH2) involved in both cGAS and RIG-I signaling may be present in the interactomes of hnRNPM and ELAVL1. We also addressed this point in lines 255-258 of the first submission (revised manuscript: lines 328-331). To explicitly emphasize potential negative regulatory functions of ILF2 and hnRNPH2 in cGAS and RIG-I signaling to the reader, we have added a paragraph to the discussion in the revised version of this manuscript (lines 710-722). In addition, we included a paragraph explaining target selection in more detail (lines 288-304). Furthermore, we have added the corresponding data values to each cell in the heatmap in Fig. EV3A/EV3B (formerly Appendix Fig. S3A/S3B), helping the reader to assess which knockdowns reduced ISRE reporter activity by more than 50%. We hope that these explanations and modifications sufficiently clarify our decision to focus on ELAVL1 in this manuscript.

3. In figure 4F, they showed NF-κB reporter activity upon various stimuli. It is well known that the activation of the cGAS-STING pathway can induce NF-κB activation in a STING-TAK1-IKKs axis-dependent manner. Nevertheless, cGAMP did not increase NF-κB reporter activity, and they did not mention this at all. They should clearly explain about this point.

We thank the reviewer for raising this point. Throughout our experiments, we were unable to detect NF-κB reporter activation induced by 2'3'-cGAMP (Fig. 4F). By contrast, we were able to monitor STING activation by 2'3'-cGAMP using the ISRE reporter of THP-1 dual cells (Fig. 4D), suggesting that the ISRE reporter is more sensitive compared to the NF-κB reporter. Nonetheless, we consistently observed relatively low ISRE reporter induction by 2'3'-cGAMP compared to lipofected stimuli such as 5'ppp-dsRNA, pDNA, or G₃-YSD. To illustrate this, we plotted the absolute relative light units (RLUs) of the stimulations shown in Fig. 4C-4D below (Revision Fig. 7, please also see Fig. 1D).

Revision Fig. 7: Low absolute RLU signals after stimulation with 2'3'-cGAMP compared to lipofected cGAS or RIG-I ligands.

ISRE reporter activation in THP-1 WT, ELAVL1 KO (#1), and ELAVL1 KO cells expressing ELAVL1-FLAG or GFP 20 h after stimulation with 5'ppp-dsRNA (0.1 μ g/ml), pDNA (0.1 μ g/ml), G₃-YSD (0.5 μ g/ml), 2'3'-cGAMP (10 μ g/ml), or IFN α (1000 U/ml). ctrl, non-stimulated; RLU, relative light units.

The lower luciferase activity induced by 2'3'-cGAMP compared to 5'ppp-dsRNA, pDNA, and G₃-YSD is most likely caused by the different availabilities of the stimuli in the cytosol of the cell. While 5'ppp-dsRNA, pDNA, and G₃-YSD are lipofected, 2'3'-cGAMP cannot be transfected with lipofectamine and is added to the cell culture supernatant in the absence of any transfection reagent. It is conceivable that a portion of 2'3'-cGAMP does not reach the cytosol, where it can activate STING. In addition, we had to use a high 2'3'-cGAMP concentration (10 μ g/ml) to induce a detectable ISRE reporter activity, while 5'ppp-dsRNA, pDNA, and G₃-YSD were used at 0.1 μ g/ml, 0.1 μ g/ml, and 0.5 μ g/ml, respectively. In summary, the ISRE luciferase reporter of THP-1 dual cells is highly sensitive and can thus be used to monitor STING activation by 2'3'-cGAMP. By contrast, 5'ppp-dsRNA, pDNA, and G₃-YSD, which are very potent inducers of the ISRE reporter, only modestly activate the secreted alkaline phosphatase (SEAP)-induced NF-κB reporter (Fig. 4F), further suggesting lower sensitivity of the NF-κB reporter compared to the ISRE reporter. In addition, we observed that the cGAS-mediated NF-κB reporter activity induced by pDNA and G₃-YSD is weaker compared to 5'ppp-dsRNA (Fig. 4F), indicating that the detection window of cGAS-STING-induced NF-κB reporter activation is relatively narrow compared to the RIG-I pathway. In our hands, the only potent inducer of the NF-κB reporter in THP-1 cells is Pam3CSK4 (Fig. 4F). Given the lower sensitivity of the NF-κB reporter compared to the ISRE reporter and the presumably less efficient cytosolic delivery of 2'3'-cGAMP compared to lipofected stimuli, we were not able to detect the NF-κB reporter induced by 2'3'-cGAMP. Therefore, the lack of NF-κB reporter activity induced by 2'3'-cGAMP is likely due to a technical problem that cannot be easily resolved.

Considering that both cytosolic dsDNA and 2'3'-cGAMP activate the same pathway and that STING is the only known receptor for 2'3'-cGAMP in human cells, we are convinced that the reduced NF- κ B reporter activity induced by cGAS activation with pDNA or G₃-YSD in ELAVL1 KO cells demonstrates that ELAVL1 is a positive regulator of cGAS-mediated NF- κ B signaling. Since it was reported by others that 2'3'-cGAMP activates NF- κ B signaling (Balka *et al.*, 2020), we have added a paragraph to the discussion in the revised manuscript to emphasize the lack of NF- κ B reporter activation upon 2'3'-cGAMP challenge in our model system (lines 749-775). We hope that our explanations and text modifications adequately address the reviewer's concern regarding the absence of the 2'3'-cGAMP-mediated NF- κ B reporter activity in Fig. 4F.

4. In figure 4C, ISRE reporter activity in ELAVL1 KO cells stably expressing ELAVL1-FLAG, is restored to levels comparable to control cells. However, the expression of IFNB1 and CXCL10 mRNA in figure 4G and 4H, as well as the production of CXCL10 protein in figure S4D, were not restored in several samples. They should explain about these discrepancies. Furthermore, to strengthen their conclusion, they should include the data showing mRNA expression level upon pDNA stimulation in figure 4G-I, and the data showing CXCL10 protein expression upon G3-YSD in figure S4D.

We have now tested the respective stimuli in the readouts shown in Fig. 4G-4I as well as the CXCL10 ELISA shown in Fig. EV4E (formerly Fig. S4D) of the revised manuscript. As expected, mRNA expression of IFNB1 and IFIT1 induced by stimulation of cGAS with pDNA were strongly reduced in ELAVL1 KO cells and rescued in THP-1 ELAVL1 KO cells expressing ELAVL1-FLAG but not GFP (Fig. 4G, 4I). Similarly, the pDNA-induced mRNA expression of CXCL10 was blunted in ELAVL1 KO cells and largely rescued in ELAVL1-FLAG expression KO cells (Fig. 4H). Since the number of data points in the other conditions of Fig. 4G-4I increased in the course of the revision, we now demonstrate that IFNB1 mRNA expression induced by G₃-YSD stimulation is almost completely rescued by ELAVL1 reconstitution. This suggests that the partial rescue observed in the initial submission in Fig. 4G was of a technical nature and could be improved by increasing the number of biological replicates.

In contrast to the 5'ppp-dsRNA condition, we observed only partial rescue of CXCL10 mRNA expression induced by pDNA and G₃-YSD in ELAVL1-FLAG expressing ELAVL1 KO cells (Fig. 4H) (however, no significant difference between WT and rescued ELAVL1 KO in these conditions). Similarly, CXCL10 protein secretion induced by 5'ppp-dsRNA was only partially rescued by ELAVL1-FLAG overexpression in ELAVL1 KO cells and CXCL10 secretion barely increased upon stimulation with pDNA or G₃-YSD (slight but insignificant increase observed in pDNA condition) (Fig. EV4E). Please note that the data of the pDNA condition shown in Fig. 4G-4I and Fig. EV4E come from the same experiment. Because we observed full rescue of the pDNA-mediated *IFNB1* and *IFIT1* mRNA expression by ELAVL1-FLAG expression in ELAVL1 KO cells (Fig. 4G, 4I), it is likely that the partial rescues in the respective condition of the CXCL10 readouts (mRNA expression in Fig. 4H and protein secretion in Fig. EV4E) are biological or clonal phenomena. Note that the rescued ELAVL1 KO is a single clone cell line derived from the WT pool. Since the responsiveness of a cell pool to different stimuli follows a normal distribution, it is conceivable that relative to the average of the parental cell pool the selected KO clone exhibits a lower level of responsiveness to the cGAS pathway compared to RIG-I. Considering that G₃-YSD and pDNA are weaker stimuli than 5'ppp-dsRNA and that different (IFN-induced) target genes of cGAS and RIG-I have different promoters and therefore different sensitivity, it is thinkable that the incomplete rescue is a clonal artifact only observed for CXCL10 readouts. In addition, the validated KO of ELAVL1 may also have downregulated or upregulated other factors that impair the cGAS pathway or promotor methylation of selected ISGs, leading to lower expression of selected target genes despite of equal cGAS activation. Those methylation patterns could also change upon culture time and freeze-thaw cycles of the cells. Therefore, we interpret a clear signal in our readouts induced by ELAVL1 expression –even if only partial– as a proof that ELAVL1 plays an important role in the activation process. To explicitly highlight the lack of rescue of the cGAS-induced CXCL10 protein secretion in ELAVL1-FLAG expressing KO cells in Fig. EV4E, we have changed the text accordingly (lines 457-461).

5. In figure 5A and 5B, they describe that the interaction of hnRNPM with IKKb, TBK1, IKKe, and NF- κ B p65 were independent of RNA, and that the interaction between hnRNPM and ELAVL1 was slightly decreased upon RNase A treatment. While these points are indeed factual, in Figure 5A, the interaction between hnRNPM and pTBK1-Ser172 or NF- κ B p65-Ser536 is significantly reduced by RNase A treatment. These findings may suggest critical points in the regulation of type I IFN production via the RIG-I/cGAS pathway by hnRNPM, thus warranting further evaluation.

We thank the reviewer for her/his insightful comment regarding the RNA-dependent interactions of hnRNPM with pTBK1-Ser172, NF- κ B p65-Ser536, and ELAVL1. Similar to the reviewer, we have also noted that RNA bound by hnRNPM or ELAVL1 may have important regulatory roles in cGAS or RIG-I signaling.

To identify these RNAs, one could immunoprecipitate hnRNPM followed by perform RNA sequencing. A similar approach was employed by Morchikh *et al.*, who discovered that Hexamethylene bis-acetamide-inducible protein

1 (HEXIM1) binds to the long non-coding RNA NEAT1 to promote cGAS signaling (Morchikh *et al*, 2017). While we fully agree that this is a highly interesting experiment, we consider it beyond the scope of the current manuscript. We appreciate the reviewer's suggestion and are actively considering it for future investigations. To emphasize the partial RNA-dependent interaction of hnRNPM with pTBK1-Ser172, NF- κ B pp65-Ser536, and ELAVL1 to the reader, we have modified some sections of the main text accordingly (lines 517-556; 739-745).

6. At line 367-370, they describe that "Although, we cannot exclude that ... ". However, without providing evidence for this, there is a potential for arbitrary interpretation. To strengthen their conclusion, they should evaluate the interaction between ELAVL1 and IKKs also using PLA assays.

We thank the reviewer for raising this point and agree that the phrasing of this sentence was too imprecise. Since PLA is a very sophisticated assay and difficult to establish for each antibody pair, we instead repeated the IP of ELAVL1 using twice the amount of cell lysate as suggested by reviewer 1 in comment 8. As discussed above, we are now able to show RNA-independent co-precipitation of IKK β and RNA-dependent co-precipitation of IKK ϵ with ELAVL1 in addition to the previously confirmed interaction of ELAVL1 with hnRNPM and NF- κ B p65 (Fig. 5B) (please see response to comment 8 of reviewer 1). These new results show that hnRNPM and ELAVL1 interact with the same set of shared signaling components (TBK1, IKK ϵ , IKK β , NF- κ B p65) involved in both cGAS and RIG-I signaling. In our opinion, these new results considerably improve the overall quality of the revised manuscript and we hope that we have adequately addressed the reviewer's concern.

7. At lines 170-173, they express 'unexpectedly strong inhibition,' but this expression feels inappropriate. This is because both the knockdown efficiency (Figure 1B) and the ISRE inhibition efficiency (Figure 1D) are around 50-70%, and there is no significant difference.

We agree with the reviewer that our phrasing was too imprecise. We therefore edited the text accordingly (lines 204-219).

8. The manuscript lacks concise explanations or full names for some specialized terms, resulting in the sudden appearance of terms throughout the text. For readability purposes, specialized terms should include their full names or brief explanations upon their first occurrence. For example, CVB3 and PV 3C.

We thank the reviewer for noticing missing full names. CVB3 and PV were defined as Coxsackievirus B3 and Poliovirus in lines 126-128 of the original manuscript (revised manuscript lines 156-157) and we would leave the definition unchanged. As correctly pointed out by the reviewer, we have found several instances lacking full names and have provided missing full in the revised manuscript.

9. At lines 249-250, they state that 99.0% of the ELAVL1 interactors interact with hnRNPM, but in reality, it is 93.8% (76/81).

We thank the reviewer for noticing this mistake and have edited the text as suggested (line 323).

10. In Figure 2B, DDX3X is missing, resulting in 27 squared words.

We thank the reviewer for noticing this missing label and have modified Fig. 2B accordingly.

11. For readability purposes, it might be better to place the controls on the left in all figures.

We have changed all figures as suggested.

12. In figure S1A, they should use STAT1 instead of b-actin as a control for phosphorylated STAT1.

We agree with the reviewer that total STAT1 is the appropriate control for phosphorylated STAT1 and have repeated the experiment as suggested (Fig. EV1D).

13. In figure S4A and B, to strengthen their conclusion, it is better to add cGAMP stimulation.

We agree with the reviewer that inclusion of this data further strengthens our conclusion and have added the requested data to the revised manuscript (Fig. EV4A-EV4C). As already observed in Fig. 4D, KO of ELAVL1 reduced the 2'3'-cGAMP-mediated ISRE reporter activation. As expected, cGAMP-induced ISRE reporter activity was unchanged in RIG-I, MAVS, and cGAS KO cells compared to control. Similar to Fig. 4F and as discussed in the response to comment 3 of reviewer 2, we were unable to detect NF- κ B reporter activation upon treatment with 2'3'-cGAMP.

14. At lines 512-515, references should be provided.

We have now added the respective reference to this sentence and thank the reviewer for noticing this missing citation (lines 831).

15. In figure S5B, the label on the Y-axis appears to be incorrect.

We thank the reviewer for noticing this mistake and have changed the y axis label accordingly (Fig. EV5B).

16. All of tables are missing.

We apologize for not uploading the tables during submission and have now uploaded them.

Additional modifications

- As recommended in the instructions for preparing a revised manuscript, the Materials table of the initially submitted manuscript was converted to the format of Reagents and Tools Table provided by the journal.

Overview of figure modifications

- As recommended in the instructions for preparing a revised manuscript, the supplementary figures of the initially submitted manuscript (formerly Appendix Figure S1-S5) were renamed to Figure EV1-EV5. Former Appendix Figure S6 is now named Appendix Figure S1.

Figure	Change in the revised manuscript
1A	Added brightfield images
1B	Reported p values as exact numbers; otherwise unchanged
1C	Unchanged
1D	Reported p values as exact numbers; ctrl moved to left; data set was revised (conclusion unchanged)
1E	ctrl moved to left; otherwise unchanged
1F	Reported p values as exact numbers; ctrl moved to left; otherwise unchanged
1G	Reported p values as exact numbers; ctrl moved to left; otherwise unchanged
1H	Reported p values as exact numbers; ctrl moved to left; otherwise unchanged
1I	Unchanged
2A	Unchanged
2B	DDX3X label was added
3A	Reported p values as exact numbers; ctrl moved to left; otherwise unchanged
3B	Reported p values as exact numbers, otherwise unchanged
3C	Reported p values as exact numbers; ctrl moved to left; otherwise unchanged
3D	Reported p values as exact numbers; ctrl moved to left; otherwise unchanged
3E	Formerly Fig. 3F
3F	Formerly Fig. 3G
former 3E	Removed former Fig. 3E; co-IP of ELAVL1 now shown in Fig. 5A
4A	Unchanged
4B	Reported p values as exact numbers; ctrl moved to left; otherwise unchanged
4C	Reported p values as exact numbers; ctrl moved to left; data set was revised (conclusion unchanged)
4D	Reported p values as exact numbers; ctrl moved to left; otherwise unchanged
4E	ctrl moved to left; data set was revised (conclusion unchanged)
4F	Reported p values as exact numbers; ctrl moved to left; otherwise unchanged
4G	pDNA stimulus was added; reported p values as exact numbers; ctrl moved to left
4H	pDNA stimulus was added; reported p values as exact numbers; ctrl moved to left
4I	pDNA stimulus was added; reported p values as exact numbers; ctrl moved to left

4J	Unchanged
4K	Unchanged
4L	Reported p values as exact numbers; ctrl moved to left; otherwise unchanged
4M	Reported p values as exact numbers; ctrl moved to left; otherwise unchanged
5A	Added ELAVL1 from former Fig. 3E
5B	New Western blots of the repeated ELAVL1 IP with more lysate
5C	Reported p values as exact numbers; ctrl moved to left; added number of cells; otherwise unchanged
5D	Reported p values as exact numbers; ctrl moved to left; added number of cells; otherwise unchanged
5E	Reported p values as exact numbers; ctrl moved to left; added number of cells; otherwise unchanged
6A	Reported p values as exact numbers; ctrl moved to left; otherwise unchanged
6B	Reported p values as exact numbers; ctrl moved to left; otherwise unchanged
6C	Unchanged
6D	Reported p values as exact numbers; ctrl moved to left; otherwise unchanged
6E	Reported p values as exact numbers; ctrl moved to left; otherwise unchanged
6F	Reported p values as exact numbers; otherwise unchanged
6G	Reported p values as exact numbers; otherwise unchanged
6H	Moved the components of the hnRNPM-ELAVL1 complex closer together
EV1A	New figure: RT-qPCR of IFNB1 mRNA in hnRNPM KD cells
EV1B	New figure: RT-qPCR of CXCL10 mRNA in hnRNPM KD cells
EV1C	New figure: Quantification of Western blots shown in Fig. 11
EV1D	Total STAT1 blot was added (formerly Fig. S1A)
EV1E	Unchanged (formerly Fig. S1B)
EV1F	New figure: basal expression of key proteins involved cGAS and RIG-I signaling in hnRNPM KD cells
EV1G	New figure: Quantification of Western blots shown in Fig. EV1F
EV2	Unchanged (formerly Fig. S2)
EV3A	Data values were added to each cell (formerly Fig. S3A)
EV3B	Data values were added to each cell (formerly Fig. S3B)
EV3C	Unchanged (formerly Fig. S3C)
EV3D	Unchanged (formerly Fig. S3D)
EV4A	Updated figure containing new round of stimulations (formerly Fig. S4A)
EV4B	New Figure containing cGAMP stimulus
EV4C	Updated Figure containing new round of stimulations (formerly Fig. S4B)
EV4D	Reported p values as exact numbers; ctrl moved to left; otherwise unchanged (formerly Fig. S4C)

EV4E	G ₃ -YSD stimulus was added; reported p values as exact numbers; ctrl moved to left; otherwise unchanged (formerly Fig. S4D)
EV4F	New figure: Quantification of Western blots shown in Fig. 4J
EV4G	New figure: Quantification of Western blots shown in Fig. 4J
EV4H	New figure: Quantification of Western blots shown in Fig. 4J
EV4I	New figure: Quantification of Western blots shown in Fig. 4K
EV5A	ctrl moved to left; otherwise unchanged (formerly Fig. S5A)
EV5B	ctrl moved to left; otherwise unchanged; y axis label was correct (formerly Fig. S5B)
EV5C	Unchanged (formerly Fig. S5C)
Appendix Figure S1A	ctrl moved to left; otherwise unchanged (formerly Fig. S6A)
Appendix Figure S1B	ctrl moved to left; otherwise unchanged (formerly Fig. S6B)
Appendix Figure S1C	Unchanged (formerly Fig. S6C)
Appendix Figure S1D	Reported p values as exact numbers; ctrl moved to left; otherwise unchanged (formerly Fig. S6D)
Appendix Figure S1E	Unchanged (formerly Fig. S6E)
Appendix Figure S1F	Unchanged (formerly Fig. S6F)

Revision References

- Aebersold R & Mann M (2003) Mass spectrometry-based proteomics. *Nature* 422: 198–207
- Balka KR, Louis C, Saunders TL, Smith AM, Calleja DJ, D'Silva DB, Moghaddas F, Tailler M, Lawlor KE, Zhan Y, *et al* (2020) TBK1 and IKK ϵ Act Redundantly to Mediate STING-Induced NF- κ B Responses in Myeloid Cells. *Cell Rep* 31: 107492
- Cansizoglu AE, Lee BJ, Zhang ZC, Fontoura BMA & Chook YM (2007) Structure-based design of a pathway-specific nuclear import inhibitor. *Nat Struct Mol Biol* 14: 452–454
- Chiang H-S, Zhao Y, Song J-H, Liu S, Wang N, Terhorst C, Sharpe AH, Basavappa M, Jeffrey KL & Reinecker H-C (2014) GEF-H1 controls microtubule-dependent sensing of nucleic acids for antiviral host defenses. *Nat Immunol* 15: 63–71
- Du M, Liu J, Chen X, Xie Y, Yuan C, Xiang Y, Sun B, Lan K, Chen M, James SJ, *et al* (2015) Casein Kinase II Controls TBK1/IRF3 Activation in IFN Response against Viral Infection. *J Immunol* 194: 4477–4488
- Goel RR, Wang X, O'Neil LJ, Nakabo S, Hasneen K, Gupta S, Wigerblad G, Blanco LP, Kopp JB, Morasso MI, *et al* (2020) Interferon lambda promotes immune dysregulation and tissue inflammation in TLR7-induced lupus. *Proc Natl Acad Sci* 117: 5409–5419
- Lee BJ, Cansizoglu AE, Süel KE, Louis TH, Zhang Z & Chook YM (2006) Rules for Nuclear Localization Sequence Recognition by Karyopherin β 2. *Cell* 126: 543–558
- Liu S, Cai X, Wu J, Cong Q, Chen X, Li T, Du F, Ren J, Wu YT, Grishin N V., *et al* (2015) Phosphorylation of innate immune adaptor proteins MAVS, STING, and TRIF induces IRF3 activation. *Science* 347: aaa2630
- Morchikh M, Cribier A, Raffel R, Amraoui S, Cau J, Severac D, Dubois E, Schwartz O, Bennasser Y & Benkirane M (2017) HEXIM1 and NEAT1 Long Non-coding RNA Form a Multi-subunit Complex that Regulates DNA-Mediated Innate Immune Response. *Mol Cell* 67: 387-399.e5
- Roth S, Rottach A, Lotz-Havla AS, Laux V, Muschaweckh A, Gersting SW, Muntau AC, Hopfner K-P, Jin L, Vanness K, *et al* (2014) Rad50-CARD9 interactions link cytosolic DNA sensing to IL-1 β production. *Nat Immunol* 15: 538–545
- Rothamel K, Arcos S, Kim B, Reasoner C, Lisy S, Mukherjee N & Ascano M (2021) ELAVL1 primarily couples mRNA stability with the 3' UTRs of interferon-stimulated genes. *Cell Rep* 35: 109178
- Varjak M, Saul S, Arike L, Lulla A, Peil L & Merits A (2013) Magnetic Fractionation and Proteomic Dissection of Cellular Organelles Occupied by the Late Replication Complexes of Semliki Forest Virus. *J Virol* 87: 10295–10312
- Xu D, Marquis K, Pei J, Fu S-C, Cağatay T, Grishin N V & Chook YM (2015) LocNES: a computational tool for locating classical NESs in CRM1 cargo proteins. *Bioinformatics* 31: 1357–1365

Dear Martin,

Thank you again for the submission of your revised manuscript to The EMBO Journal. I sincerely apologize for the rather protracted review process on this occasion -which was due to the unavailability of the referees for a few weeks and then a backlog in our editorial office- and thank you very much for your understanding and patience.

We have now received the comments of both referees (included below for your information), and I am glad to say that they are both satisfied with the revision and acknowledge that the revised manuscript has been significantly improved and their previous concerns sufficiently addressed. I am thus happy to say that your manuscript has been in principle accepted for publication in The EMBO Journal. Congratulations on an excellent manuscript!

From the editorial side, there are a few minor changes and corrections that we need from you before we can proceed with formal acceptance of the manuscript and its publication:

- Please note that the full funding information should be entered in our manuscript handling system (eJP) as well as listed in the Acknowledgements section of the manuscript itself. Information related to "Bo&MeRanG GRK 2168 and BONFOR (University of Bonn)" is currently missing from eJP.

- Please provide a list of up to 5 relevant keywords after the Abstract of your revised manuscript.

- Thank you for providing access to your deposited datasets. Now that review is complete, the confidential reviewer access credentials (usernames and passwords) can be removed from the Data availability statement, and should be replaced by the specific URLs to the deposited mass spectrometry and 3'-mRNA sequencing datasets. Please make sure that all datasets will be publicly available at the time of publication.

- The author contributions statement should be removed from the manuscript file. Instead, we use CRediT to specify the contributions of each author in the journal submission system. Please feel free to use the free text box to provide more detailed descriptions during submission. See also our guide to authors for more information:
<https://www.embopress.org/page/journal/14602075/authorguide#authorshipguidelines>.

- As per our journal's policy, "data not shown" (stated on pages 8, 12, 42 -3 times- and 45 of your manuscript) is not permitted. All data referred to in the paper should be displayed in the main or Expanded View figures, or in the Appendix. Please add these data or change the text accordingly if these data are not central to the study and its conclusions, or properly cite the respective published sources if these data can be found elsewhere.

- Please rename Tables S1 and S2 to "Dataset EV1" and "Dataset EV2", respectively. Their legends should be uploaded as a separate tab in each Excel file. Please also update accordingly their callouts throughout the manuscript.

- Tables S3-S8 should be renamed to "Table EV1-EV6" with the corresponding callouts and labels updated accordingly.

- The Appendix Figure S1 should only be contained in the Appendix, please do not upload it individually as well.

- Please remove your Reagents and Tools table from the main manuscript file; it should only remain uploaded as an individual file ("Reagent Table").

- Please note that EMBO press papers are accompanied online by:

A) a short (2 sentences) summary of the findings and their significance,

B) 2-5 short bullet points highlighting the key results, and

C) a synopsis image in .jpg or .png format that is exactly 550 pixels wide and 300-600 pixels high (the height is variable). Please note that the text needs to be legible at the final size. Please upload this information along with your revised manuscript (the text for A and B should be provided in a separate Word file).

- Please correct the section order of the manuscript as follows: title page with complete author information, abstract, keywords, introduction, results, discussion, methods, data availability section, acknowledgements, disclosure and competing interests statement, references, main figure legends, tables, expanded figure legends.

- During our routine pre-acceptance checks, our data editors have raised the following queries regarding figures, data, and legends. You are kindly requested to address them all completely in your revised manuscript:

1. Please note that Figure 4e does not contain p-value, kindly rectify the statistical test related information in the figure legend appropriately.

2. Please note that the exact p values are not provided in the legends of Figures 1b, d, f, h; 3a, d; 4b-d, f-i, l-m; 5c-e; 6a-b, d-e; EV 4d-e.
3. Please indicate the statistical test used for data analysis in the legends of Figures 3f; EV 2; EV 5c.
4. Please note that information related to "n" is missing in the legends of Figures 1b-h; 3a-d; 4b-i, l-m; 6a-b, d-g; EV 1a-c, g; EV 4a-i; EV 5a-b.
5. Although "n" is provided, please describe the nature of entity for "n" in the legends of Figures 5c-e.
6. Please note that scale bar and its definition are missing for Figure 1a.

Please also note that we have failed to reach the co-author Maria Hønholt Christensen at the e-mail address listed in the author's profile (christen@uni-bonn.de), as this address has bounced. Could you please make sure that the profile of this co-author is properly updated with the current contact information so that we can keep all co-authors informed on the progress of the manuscript?

I would also like to note that as part of the EMBO publications' Transparent Editorial Process, The EMBO Journal publishes online a Peer Review File along with each accepted manuscript. This File will be published in conjunction with your paper and will include the referee reports, your point-by-point response and all pertinent correspondence relating to the manuscript. You can opt out of this by letting the editorial office know (contact@embojournal.org). If you do opt out, the Peer Review File link will point to the following statement: "No Peer Review File is available with this article, as the authors have chosen not to make the review process public in this case."

We look forward to seeing a final version of your manuscript as soon as possible. Please let us know if you have any questions and use this link to submit your revision: <https://emboj.msubmit.net/cgi-bin/main.plex>

Best wishes,

Ioannis

Referee #1:

The authors have done a great job to address my questions. I have no more concerns. I appreciate the response from the authors.

Referee #2:

Overall, their response seems to adequately address this reviewer's comments. However, the line numbers provided in the point-by-point response do not match the line numbers in the manuscript, making it difficult to thoroughly evaluate the revised version. This reviewer cannot determine whether the discrepancy is due to an error on the authors' part or a system issue, but requests that they resubmit the manuscript.

All editorial and formatting issues were resolved by the authors.

Dear Martin,

Congratulations on an excellent manuscript! I am very pleased to inform you that it has been accepted for publication in The EMBO Journal. Thank you for your thorough responses to the initially raised referees' concerns, and for addressing all editorial and formatting requests.

If you have any questions, please do not hesitate to contact the Editorial Office. Thank you for your contribution to The EMBO Journal. Working with you has been a pleasure!

Best regards,

Ioannis
